# Quantifying the intra- and inter-species community interactions in microbiomes by dynamic covariance mapping

Melis Gencel[1,2], Gisela Marrero Cofino[3], Cang Hui [4,5], Zahra Sahaf[1,2], Louis Gauthier[1,2], Chloé Matta[1,2], David Gagné-Leroux[1,2], Derek K. L. Tsang[6], Dana P. Philpott [6], Sheela Ramathan[7], Alfredo Menendez[3], Shimon Bershtein [8] & Adrian W. R. Serohijos [1,2] ✉

A microbiome's composition, stability, and response to perturbations are governed by its community interaction matrix, typically quantified through pairwise competition. However, in natural environments, microbes encounter multispecies interactions, complex conditions, and unculturable members. Moreover, evolutionary and ecological processes occur on overlapping time-scales, making intra-species clonal diversity a critical but poorly understood factor influencing community interactions. Here, we present Dynamic Covariance Mapping (DCM), a general approach to infer microbiome interaction matrices from abundance time-series data. By combining DCM with high-resolution chromosomal barcoding, we quantify inter- and intra-species interactions during *E. coli* colonization in the mouse gut under three contexts: germ-free, antibiotic-perturbed, and innate microbiota. We identify distinct temporal phases in susceptible communities: (1) destabilization upon *E. coli* invasion, (2) partial recolonization of native bacteria, and (3) a quasi-steady state where *E. coli* sub-lineages coexist with resident microbes. These phases are shaped by specific interactions between *E. coli* clones and community members, emphasizing the dynamic and lineage-specific nature of microbial networks. Our results reveal how ecological and evolutionary dynamics jointly shape microbiome structure over time. The DCM framework provides a scalable method to dissect complex community interactions and is broadly applicable to bacterial ecosystems both in vitro and in situ.

The microbiome's dynamic composition, stability, and response to perturbations are crucial to their function in the environment and to the health of their host[1,2]. These characteristics can often be explained using the community matrix, which quantifies pairwise effects of one species' abundance on another's population growth[3,4]. The community matrix, a core concept of ecology, has been scrutinized in diverse systems of plants, animals, and microbes[3–17]. The gold standard method to measure the community interactions is to perform pairwise co-culture competition experiments in vitro and in vivo. Indeed, measurements of such interactions, a form of "bottom-up" approach, have been shown to predict simple assembly rules of the community[18,19]. In their natural environment, however, microbes concurrently experience multiple species, face conditions that may be difficult to mimic in vitro, and some of these species may be challenging to culture or practically isolate. A complementary "top-down" approach is to estimate the community interaction matrix from the

time series of the abundance of community members in situ[20–23]. However, prior approaches involve parameterizing ecological models (e.g., generalized Lotka-Volterra (gLV))[24–26], which are difficult to verify in natural environments. Most non-parametric models of the community matrix do not incorporate potential evolutionary forces acting on the population[27]. Due to limited experimental resolution of 16S rRNA profiling, the community matrix is also typically described at the level of species or higher-level taxonomic groupings[5–12,28,29].

Importantly, because of the overlap of evolutionary and ecological timescales within microbiomes, the community matrix is in principle also influenced by intra-species diversity, but to what extent it matters remains poorly understood[30–34]. A species rarely exists as a homogenous population due to spatial partitioning and genomic variation from pre-existing or de novo mutations[35]. Despite the prevalence of intra-species variation (ISV) in nature, how ISV affects the community matrix is not fully known. Indeed, the role of ISV on community composition and stability has rarely been tested experimentally[29,31–33,36–39]. Some theoretical studies have also yielded contradictory results regarding the role of intraspecific variation on species coexistence[34].

Depending on host phenotype and genetics, a typical human gut microbiome, for example, consists of ~$10^{13}$ individual bacterial cells, which can be partitioned into ~$10^3$ species[40,41]. The relatively high mutation rate, large population size, and frequency-dependent selection could lead to ~$10^5$ clones co-existing in a population of a single bacterial species[42]. Some of these clones, albeit present at very low frequencies, could nonetheless provide rapid adaptation[43–46]. Altogether, there is a critical need for quantitative methods and experimental approaches to assess the inter- and intra-species community matrix within the microbiome, and thus, understand its stability and dynamics.

Here, we develop a general approach, called Dynamic Covariance Mapping (DCM), to estimate the community matrix from high-resolution community abundance time-series data, which we apply during colonization of the mouse gut microbiome. We expand the definition of community matrix interactions, traditionally defined between species or families, to include both intra-species and between-species interactions. To this end, we combine DCM with high-resolution lineage tracking by chromosomal barcoding. Deciphering the effects of intra-species clonal variation on community dynamics requires the ability to track intraspecific clonal lineages at very high resolutions. In our previous work[43], we utilized Tn7 transposon machinery to integrate ~500,000 distinct chromosomal DNA barcodes into a population of roughly $10^8$ E. coli cells. This technique allows for the tracking of clonal lineage dynamics at high resolution. This method has proven particularly useful in quantifying at high resolution how E. coli populations adapt to antibiotic resistance during laboratory evolution[43]. High-resolution chromosomal barcoding techniques have been applied in studies involving single species, such as yeast[45,46], bacteria[43], or mammalian cell lines[47,48].

First, we provide a theoretical foundation for the DCM analysis and demonstrate that the pairwise covariance between the abundance time series of one member and the time derivative of the abundance of another ("growth rate") is an accurate estimate of the inter-, intra-, and inter-/intra-species interactions in the community. The eigenvalue decomposition of the time-dependent community matrix identifies distinct temporal domains based on community stability. Second, by applying the DCM to the gut invasion data, we were able to identify distinct temporal phases during colonization. Third, we demonstrate that these temporal phases uniquely arise due to the specific interaction between E. coli clonal clusters and certain families of the resident microbiota. Notably, such interactions were reproducible across mice colonization replicates that were susceptible to E. coli invasion. Lastly, we performed whole-genome sequencing to uncover the genetic underpinnings behind the

dynamics of the clonal clusters. Through our analysis of high-resolution barcoded populations in complex environments, we demonstrated that DCM could identify the dynamical stability changes while describing the interplay between ecology and evolution in microbial communities.

## Results

### Dynamic covariance mapping (DCM)

**Theory.** We first provide a formal introduction to the ecological and mathematical theory behind DCM and its biological intuition. Without loss of generality, the microbiome as an ecological community can be described as a system of nonlinear ordinary differential equations[2,9,17]:

$$\dot{z}_i = f_i = z_i \phi_i \tag{1}$$

where $z_i$ represents the time-varying abundance of community member $i$ (=1,...,$n$), and $\dot{z}_i$ the time derivative of its abundance; $f_i$ is the population growth rate, and because the autocatalytic nature of population growth it is further expressed as the product of abundance and the *per-capita* population growth rate ($\phi_i$) of member $i$. Both rates ($f_i$ and $\phi_i$) are functions of the abundances of all community members ($\mathbf{z} = [z_1, \ldots, z_n]^T$). Here, 'member' can refer to species, family, or operational taxonomic unit (OTU) by which the community is described. In our application of DCM for the gut microbiome, this vector of abundance time series describe the community at the inter- and intra-species level (Fig. 1a).

Let the abundance vector and the population growth rate vector at time $t_*$ be denoted by $\mathbf{z}_* = [z_1(t_*), \ldots, z_n(t_*)]^T$ and $\mathbf{f}_* = [f_i(\mathbf{z}_*), \ldots, f_i(\mathbf{z}_*)]^T$ respectively. The community dynamics can be approximated by Taylor expansion at time $t_*$ as $\dot{\mathbf{z}} \approx \mathbf{f}_* + \mathbf{J}_*(\mathbf{z} - \mathbf{z}_*)$, where $J_*$ is the system Jacobian matrix evaluated at $\mathbf{z}_*$, with its non-diagonal entry $J_{ij}(\mathbf{z}_*) = \partial f_i / \partial z_j |_{z=z_*} = z_i(t_*) \partial \phi_i / \partial z_j |_{z=z_*}$ for $i \neq j$. The non-diagonal entry $J_{ij}(\mathbf{z}_*)$ of the Jacobian matrix can be used to indicate the overall impact of the variation of member $j$'s abundance on the population growth rate of member $i$ at point $\mathbf{z}_*$, while the per-capita impact of $j$ on $i$ at point $\mathbf{z}_*$ is thus $J_{ij}(z_*) / z_i(t_*) = \partial \phi_i / \partial z_j |_{z=z_*} \equiv a_{ij}(z_*)$. This allows us to approximate Eq. 1 as the following:

$$\frac{\dot{z}_i}{z_i} = \phi_i \approx \phi_i(\mathbf{z}_*) + \propto \sum_{j=1}^{n} a_{ij}(\mathbf{z}_*)\left(z_j(t) - z_j(t_*)\right). \tag{2}$$

Both $J_{ij}$ and $a_{ij}$ have been used to indicate the interaction strength of member $j$ on $i$[17], and they are not necessarily time invariant. Only in some simple models of second-order differential equations (e.g., the generalized Lotka-Volterra model (gLV)), $a_{ij}$ is assumed to be time-invariant parameters. The correspondence between Eq. 2 and gLV model with constant interaction matrix is shown in the Supplementary Information section B.1.

In dynamical systems theory, the Jacobian not only provides a way to estimate the community interaction strengths, but also the stability of the community[2,9,17,49]. Let the matrix of $a_{ij}(\mathbf{z}_*)$ be named $A_*$. Since $J = \mathrm{diag}(\mathbf{z})A$, where $\mathrm{diag}(\mathbf{z})$ is a diagonal matrix with $z_i$ as the diagonal elements; it can be shown that for an eigenvalue $\mu_i$ of $A$, there is a corresponding eigenvalue $\lambda_i$ of $J$ such that $\lambda_i = z_i \mu_i$ (Methods). If $\mathbf{z}_*$ is a feasible equilibrium of the system (i.e., $\mathbf{f}_* = \mathbf{0}$), the stability of this equilibrium can be assessed by the eigenvalues of $J$ (and thus also the eigenvalues of $A$), with a positive eigenvalue signaling system instability and the system state $z$ moving away from this equilibrium. If $\mathbf{z}_*$ is not an equilibrium, a positive eigenvalue would indicate the system state $\mathbf{z}$ departures from the reference interaction-free path $\dot{\mathbf{z}} = \mathbf{f}_*$. Indeed, the abundance time series of the community members can be expressed as a linear combination of periodic functions with varying amplitudes, where the intuition of instability refers to the presence of

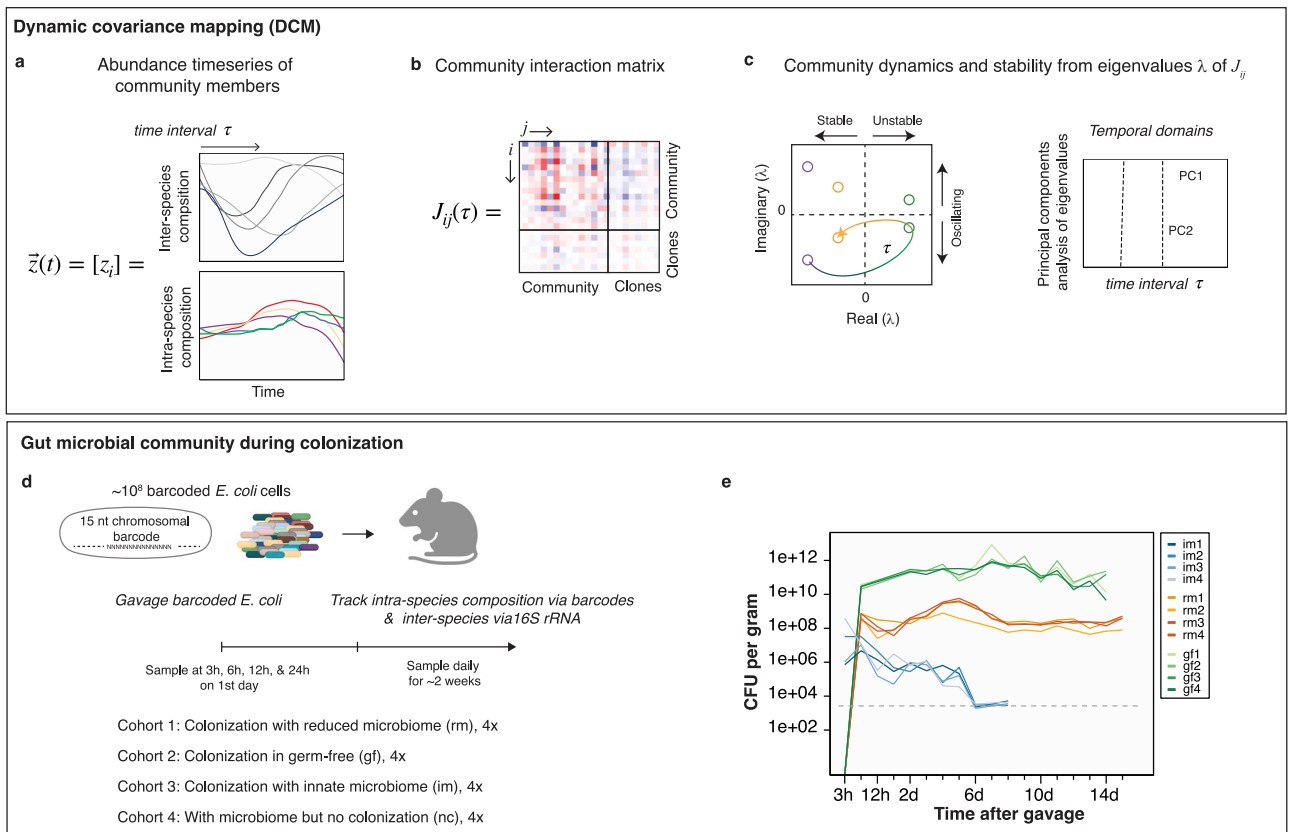

**Fig. 1 | Dynamic covariance mapping (DCM) to estimate the community interaction matrix during colonization of the gut microbiota. a** Microbial community is described by abundance time series $[z_i(t)]$ of member $i$ that could include inter- or intra-species composition. **b** The community matrix $J_{ij}(\tau)$ estimated over the time interval $\tau$ reflects the influence of the abundance of a member $j$ on the rate of increase of another member $i$. It is quantified by the covariance between $z_j(t)$ and $f_i$. When the time-series data includes inter- and intra-species compositions, the covariance matrix describes the community matrix within-community, within-clones, and between community and clones. **c** The stability and dynamics of the community can be inferred from the eigenvalues of the community matrix $J_{ij}(\tau)$. Specifically, the abundance timeseries is a linear combination of the periodic functions with frequency defined by $\text{Im}(\lambda_k)$ and amplitude by $\exp(\text{Re}(\lambda_k))$ (Main text). Consequently, the eigenvalues define stability and oscillatory features of the community. Principal component analysis (PCA) of the time-dependent

eigenvalues can reveal distinct temporal domains in community dynamics. In DCM, we look at the first two components PC1 and PC2. **d, e** Application of DCM to high-resolution community inter-species and intra-species abundance data during gut colonization. A population of ~$10^8$ *E. coli* cells with ~$5 \times 10^5$ unique chromosomal barcodes is introduced into mice with reduced microbiota by antibiotic pre-treatment (cohort 1, rm) and mice with innate microbiota (cohort 3, im). Community-level and intra-species dynamics were then tracked in fecal samples over a 2-week period. As comparison, samples were also collected in mice with only the colonizing *E. coli* (germ-free or gf, cohort 2) and in mice with only the resident bacteria and no colonization (cohort 4, nc). Mouse schema is *"Created in BioRender. Serohijos, A. (2025)* https://BioRender.com/8gmahgd*"*. **e** *E. coli* bacterial load measured as colony-forming units (CFU) per gram of sampled feces for the colonized mice cohorts with innate microbiota (im), reduced microbiota (rm), and germ-free (gf). Dashed line represents the resolution limit.

positive real parts for some eigenvalues and the presence of oscillatory behavior corresponds to the imaginary component of complex eigenvalues (Methods).

If the community abundance is recorded at time $t_1, \ldots, t_k$ within a $\tau$-width time window $(t_* - \tau/2, t_* + \tau/2)$ centered at $t_*$, we can denote $\mathbf{z_j}(t_*) = \left[z_j(t_1), \ldots, z_j(t_k)\right]^T$ the abundance time-series for species $j$ and $\boldsymbol{\phi_i} = \left[\phi_i(t_1), \ldots, \phi_i(t_k)\right]^T$ the computed vector of per-capita population growth rate of species $i$. Note that the per-capita interaction strength in Eq. 2 $a_{ij}(\mathbf{z_*})$ can be viewed as the slope of a linear regression between $\phi_i$ and $z_j$, and can be estimated using ordinary least squares as the covariance between the two variables (Methods)[50,51]. Thus, the covariance can be used as a metric for interaction strength, with $a_{ij}(\mathbf{z_*}) = \text{Cov}(\boldsymbol{\phi_i}, \mathbf{z_j})$ representing the effect of $\mathbf{z_j}$ on $\boldsymbol{\phi_i}$ measured in units scaled by the standard deviation of $\mathbf{z_j}$ its sign indicates the direction of the linear relationship. Similarly, the overall interaction strength $J_{ij}(t_*)$ can be estimated as $J_{ij}(t_*) = \text{Cov}(f_i, z_j)$. The dynamic covariances $\text{Cov}(\boldsymbol{\phi_i}, \mathbf{z_j})$ and $\text{Cov}(f_i, z_j)$ provide us estimates of the per-capita and overall interaction strengths

of member $j$ on $i$ near $t_*$.[11] DCM analysis thus maps the interaction network of the microbiota as estimated matrices $A_*$ and $J_*$, while the system stability is captured by their eigenvalues.

**Implementation.** To implement DCM, we calculate the interaction strength $J_{ij}(z_*) = \text{Cov}(f_i, z_j)$ over an expanding time window $\tau$ (Fig. 1a, b). Alternatively, the Jacobian could be calculated over a sliding time window[52], but it is more sensitive to noise in real time series data compared to an expanding time window[52,53]. In each window, we calculate the $n$ eigenvalues of the Jacobian ($n$ is size of the abundance time-series vector) to determine if the community is stable/unstable and exhibits oscillatory behaviors (Fig. 1c, left panel)[54–58]. These analyses lead to a time-dependent estimate of the community interaction matrix and the stability of the community. Finally, to determine phases due to shifts in the interaction matrix and community stability, we perform principal component analysis (PCA) on the $n$ eigenvalues, specifically tracking PC1 and PC2 (Fig. 1c, right panel). Boundaries between the phases are identified using change point analysis, which is a statistical technique to detect changes within time-series data[59–64] (Methods).

As an illustrative example, we applied DCM to a 5-species gLV system with an interaction strength matrix A1 that shifts to matrix A2 at time t = 20 (Supplementary Fig. 1a, f and see also Supplementary Movie 1). DCM's time dependent estimate of the community interaction matrix captures the gLV interaction matrices before and after the shift (Supplementary Fig. 1b). Five distinct phases are identified based on the eigenvalues and their dynamic interpretations (stable/unstable or oscillatory) (Supplementary Fig. 1c, d). Phase I corresponds to the equilibration of the abundances of 5 species (due to their initial values set at 15). The second phase corresponds shift in the interaction matrix A1 and A2. Phase III is the sudden change in abundances due to shift in A1 to A2. Phase IV is the drop in abundances of most species. Phase V is marked by the increase of species 3. Phase VI is the return to equilibrium. Notably, one of the major dynamic temporal boundaries corresponds to the shift in ecological matrix A1 and A2 at t = 20. These results show how DCM captures the dynamical changes within a constant gLV matrix (e.g., equilibration in abundances in the time intervals where A1 and A2 are constant) as well as changes in the interaction matrix itself (e.g., shift from A1 to A2). Additional illustrative examples of DCM on gLV systems are described in Supplementary Information section B.2. We also investigated the effect of the abundance sampling time step on DCM's accuracy (Supplementary Fig. 13).

Next, we applied DCM analysis to a mouse gut microbial community perturbed by infection of the pathogen *C. difficile* in a prior study[65] (Supplementary Fig. 2a, see also Supplementary Movie 2). The principal components PC1 and PC2 of the eigenvalues of the Jacobian identify four phases (Supplementary Fig. 2b-c): Phase I is the entry and establishment of a Gnotocomplex microflora, while Phase II reflect the transient instability detected from the rise in *A. mucinphilia* and *B. ovatus*. Phase III is the collapse in abundance upon entry of *C. difficile*. Phase IV is the return to stability accompanied by the increase of *A. mucinphilia* and *B. ovatus* to baseline levels in Phase I.

Altogether, this demonstrates that DCM detects actual changes in species interactions, such as shifts in interaction matrices and invasion by a pathogen. DCM, which is based on analysis of the time-series data, can also capture transient dynamics associated to fixed interaction matrices, such as during equilibration of gLV constant matric models. Importantly, DCM does not need to be informed a priori of the presence of a community shift or when this occurred.

## High-resolution intra-species and inter-species dynamics during colonization of gut bacterial community

To gain insight into community dynamics during gut microbiome colonization, we next aimed to infer the community interaction matrix and assess the system's dynamic stability. Specifically, we were interested in understanding how intra-species variation influences both stability and colonization dynamics. To achieve this, we generated high-resolution abundance time-series data for the colonizing species and the community as a whole. We previously used the Tn7 transposon machinery to introduce ~500,000 distinct chromosomal DNA barcodes into a population of ~$10^8$ *E. coli* cells[43]. Since the barcodes are transmitted from parent to daughter cells, this allowed the tracking of the clonal lineage dynamics of *E. coli* at a resolution of ~$1/10^6$ cells as the population developed antibiotic resistance during in vitro lab evolution[43]. Such high-resolution chromosomal barcoding techniques have been used for single-species analysis[43,45–48,66], but never in a complex and species-rich ecological community. Lineage tracking via barcoded plasmids have also been used to study the colonization dynamics in germ-free mice[67]. Here, we used barcodes to simultaneously track high-resolution clonal lineage dynamics of an *E. coli* population colonizing mouse guts (Fig. 1d).

It is well-established that higher diversity and species richness of an ecological community makes it less susceptible to invasion or perturbation, including the gut microbiota[68,69]. However, this resistance to invasion can be compromised upon environmental perturbations, such as antibiotic treatments, that reduce community diversity, making them susceptible even to non-pathogenic bacteria. Additionally, the gut itself presents a complex "biogeographical" environment, where distinct selective niches arise from heterogeneity in the availability of metabolites, nutrients, and immune effectors, as well as, epithelial topography and mucus architecture[70]. With these considerations in mind, we designed four mice cohorts with different complexities in their gut bacterial microbiomes (Fig. 1d): mice with reduced microbiome due to pre-treatment of antibiotics (cohort 1, "rm"); germ-free mice (cohort 2, "gf"); and mice with innate microbiome (cohort 3, "im"). Lastly, as a control for the community dynamics in the absence of colonization, we had another cohort of mice that received antibiotic treatment to reduce its microbiota like the rm cohort but not colonized by *E. coli* (cohort 4, "nc"). Cohorts 1 and 4 were pre-treated with an antibiotic cocktail (metronidazole, neomycin, ampicillin, and vancomycin) for three weeks followed by three days of no treatment to flush out the antibiotics. On day zero, barcoded *E. coli* populations were introduced in mice of cohorts 1, 2, and 3. Then, for all cohorts (1–4), fecal samples were taken at 3 h, 6 h, 12 h, and 24 h on day one, and then once daily for two weeks. The multiple sampling on day one was required to capture the kinetics of transit through the gut of the colonizing bacteria[71] (Fig. 1d). Extraction of bacterial genomic DNA from the feces, followed by deep sequencing of the *E. coli* barcoded region, afforded high-resolution lineage tracking during gut colonization (Fig. 2a, d, and Supplementary Fig. 3, see Methods for experimental details). We also simultaneously tracked the community dynamics of resident bacteria using 16S rRNA profiling (Fig. 2c, f, g).

To ascertain that the barcode dynamics are not affected by potential dropout due to technical and experimental factors, we simulated the effects of PCR bias due to jackpotting, skewness of the barcode at t = 0, and the number of *E. coli* genomes for PCR amplification (see Supplementary section A). We found that accurate barcode dynamics are guaranteed when the amount of *E. coli* genomic DNA is at least $10^5$ cells, and ideally up to ~$10^6$ (see "Methods"; Supplementary Information section A, Supplementary Fig. 11–12). In our experiments, both the gf and rm cohorts exceeded this cut-off (Supplementary Fig. 4a and b). This is evidenced by the sufficient bacterial load in the fecal samples (Fig. 1e), which allowed for the use of at least ~$10^6$ cells as input for barcode amplification. However, the im cohort, due to unsuccessful gut colonization, falls below this criterion (Fig. 1e, and Supplementary Fig. 4c). Consequently, the barcode dynamics in this cohort are likely affected by an insufficient number of *E. coli* cells and barcode sampling. Nevertheless, the 16S rRNA dynamics, which are not limited by the amount of *E. coli* cells, were accurate for all three cohorts (except gf, which only had the barcoded *E. coli*).

Expectedly, the gut community of the im cohort was resistant to the invasion of the *E. coli*, where after an initial of ~$10^6$ to ~$10^8$ colony-forming units (CFU)/gram of feces within ~3 h, the bacterial load reduced to below ~$10^4$ CFU/gram by day 6 (Fig. 1e). The unsuccessful invasion in the im cohort is also reflected in the diversity of its bacterial community, with dynamics that are largely unperturbed by the entry of *E. coli* into the community (Supplementary Fig. 5e). In contrast, *E. coli* successfully colonized the rm cohort, reaching bacterial loads of ~$10^8$ colony-forming units (CFU)/gram of feces within ~6 h (Fig. 1e), which coincided with the maximal barcode diversity (q = 1 in Supplementary Fig. 5a). The absence of resident bacteria in the gf cohort resulted in *E. coli* reaching a higher bacterial load of ~$10^{10}$ CFU/gram of sample within ~6 h. Interestingly, despite the difference in their CFU levels (Fig. 1e), both the gf and rm cohorts reach highest diversity in ~6 h (q = 1 in Supplementary Fig. 5a and b). Nonetheless, after day 6, the drop in barcode diversity of *E. coli* populations (q = 1) was more precipitous in gf than in rm mice, suggesting a stronger intraspecies selection pressure in the absence of resident bacteria.

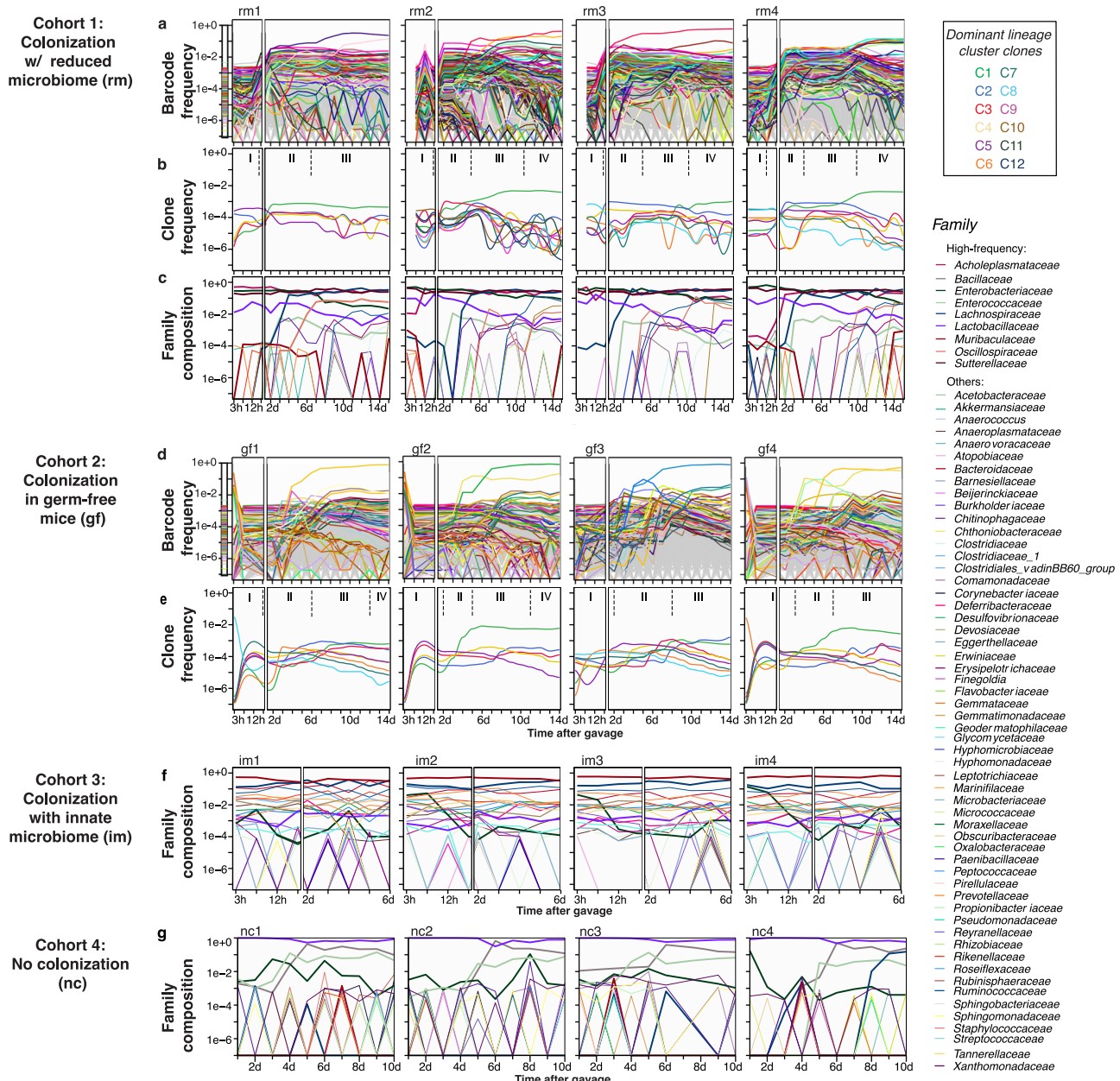

**Fig. 2 | High resolution community and intra-species population dynamics during gut colonization. a** Population of ~10$^8$ *E. coli* cells with ~5 × 10$^5$ unique chromosomal barcodes is introduced into mice with reduced microbiome by antibiotic pre-treatment (cohort 1). Fecal samples were collected over a 2-week period to track both community-level and intra-species dynamics. The leftmost panel shows the barcodes in the input (gavage) sample. The 1000 most frequent barcodes are uniquely colored, whereas the rest are shown in gray. Identical barcodes are colored consistently across mouse replicates and cohorts (see Supplementary Fig. 3a for Muller plot). **b** Dominant clonal clusters were identified by clustering barcodes that persisted for most of the 2-week period. Pairwise Pearson correlation of barcode frequency time series was used as the distance metric. These dominant clusters represent ~5–7% of total unique barcodes. The clusters are ranked by their average frequency (see Supplementary Fig. 6d). Roman numerals indicate distinct phases determined by DCM (Fig. 3). **c** Community dynamics by 16S rRNA profiling

are analyzed at the level of the family. **d** Barcode frequency dynamics in germ-free (gf) mice colonized with the same *E. coli* population. The most frequent 1000 barcodes are colored uniquely, whereas the rest are shown in gray. Identical barcodes are colored similarly across mouse replicates and cohorts (see Supplementary Fig. 3b for the Muller plot). **e**. Dominant clonal clusters are determined by clustering of barcodes that persisted for most of the 2-week period (see Supplementary Fig. 6e). Phases are determined from DCM analysis of the gf cohort (Supplementary Fig. 7: DCM of gf cohort). **f** Community dynamics (16S rRNA profiling) are analyzed at the level of the family. *E. coli* did not successfully colonize the community, as shown by the CFU (Fig. 1e). There are not enough cells to perform high-resolution lineage-tracking with sufficient accuracy (Supplementary Fig. 4). **g** Community dynamics in mice with reduced microbiota, but non-gavage with *E. coli*, showing the recovery of bacterial community from the treatment of antibiotic cocktail. Colors correspond different bacterial families (legend box).

Notably, the CFU counts for mice in the same cohort are broadly indistinguishable (Fig. 1e) despite the complexity of the underlying clonal dynamics viewed at higher resolution (Fig. 2a, d, Supplementary Fig. 3a and b). Although barcodes begin appearing within ~3 h (Fig. 2a, d), they were not observed in the CFU counts for the rm and

gf cohorts (Fig. 1e). Lineages that appeared first were not always the dominant ones at the end (Supplementary Fig. 3a, b). These results highlight the stochasticity of transmission kinetics through the intestinal gut's distinct "island" niches[70], which is not reflected simply by measuring the total bacterial count of the invading species.

## Persistent and dominant clonal lineage clusters during colonization

The ~$10^5$ unique barcodes that we introduced could be used to track the population lineages at high resolution. To simplify the analysis, we identified the most dominant *E. coli* lineages using the chromosomal barcodes (Fig. 2a, d). The barcode lineage dynamics reflects its effective fitness (selection coefficient) over time[45], and thus similarities between individual barcode lineages can be indicative of *E. coli* clones with similar selection coefficients.

We performed a hierarchical clustering analysis of the linage dynamics (Supplementary Fig. 6) using Pearson correlation as the similarity measure computed for pairs of barcodes with mean frequency greater than $5 \times 10^{-5}$ and persisted for at least 12 of the 18 timepoints for rm and 17 timepoints for gf (Supplementary Fig. 6). These persistent barcodes represent ~5-10% of the total barcoded *E. coli* observed in the rm and gf mice (Methods). Clusters of persistent barcodes were defined as putative lineage clones and ranked based on average frequency (Supplementary Fig. 6d & e). The running average (locally estimated scatterplot smoothing (LOESS)) of each cluster is hereafter referred to as a "clonal cluster" (Fig. 2b for the rm cohort and Fig. 2e for the gf cohort). Interestingly, the clonal cluster C1 always contained the dominant barcode lineages that exhibited the sweeps even if C1 cluster itself did not have the largest number of barcodes (Supplementary Fig. 6d & e). Altogether, the high-density barcoding and clustering identified the persistence of very low-frequency *E. coli* lineage clonal clusters during gut colonization (Fig. 2b & e).

## Dynamic covariance mapping defines distinct temporal phases of colonization

Next, we applied DCM on the rm cohort using the combined relative abundance time-series of the community from 16S rRNA dynamics and the colonizing *E. coli* from clonal lineage clusters (Fig. 3). We calculated the community matrix using the covariance estimate of the Jacobian over a progressively increasing time intervals and determined its eigenvalues. Since, for each time interval, there are ~12–19 eigenvalues (*E. coli* clonal cluster lineages number plus ~7 community species from 16S rRNA profiling, see Methods), we performed kernel principal component analysis (PCA) to reduce its dimensionality. The principal components (PCA 1 and 2) of the eigenvalue matrix of $J_\tau$ over the time interval $\tau$ for mouse rm1 are shown in Fig. 3a (see also Supplementary Movie 3). A change in the direction of either PC1 or PC2 is indicative of changes in the eigenvalues and hence of dynamical shifts, such as changes in stable/unstable or manifestation of oscillatory behavior, in the community. PC1 and PC2 capture 54 to 20% of the variance in the eigenvalue matrix (Supplementary Fig. 9). Based on this criterion, we identified distinct invasion phases using changepoint analysis (Methods). In rm1, we observed three phases, whereas rm2, rm3, and rm4 exhibited four phases. Notably, Phase 1 was consistent across all mice. Phase I reflects the transient instability in day 1 (positive real eigenvalues) corresponding to the entry of *E. coli* and the collapse of resident bacteria (Fig. 3b, left). This phase is accompanied by an increase in CFU (Fig. 1e, orange lines for rm) and barcode diversity (Supplementary Fig. 5a). The collapse in resident community in Phase I is manifested in the low bacterial community diversity (Supplementary Fig. 5c). Phase II is the return to a stable regime (negative real eigenvalues) (Fig. 3b, middle) and the re-emergence of the community species (Fig. 2c, also Supplementary Fig. 5c). Phase III in rm 1 (Fig. 3b, right; Fig. 2b,c) and Phase IV in rm2, rm3, and rm4 (Fig. 3d,f,h; Fig. 2b,c) is quasi-dynamic equilibrium with both oscillations in the clonal and community dynamics. The oscillatory behavior of the barcode dynamics that affects the community dynamics is unlikely due to technical and experimental factors, such as PCR bias, efficiency in genome extraction, or number of *E. coli* genomes for PCR amplification (see Supplementary Information section A). The dynamical transition from stability, to instability, and return to stability during colonization

is broadly reproducible across the 4 mice of the rm cohort (Fig. 3c–h; and Supplementary Movies 3–6).

We applied DCM to the gf cohort's clonal cluster dynamics and identified 4 phases for gf1 and gf2 whereas 3 phase for gf3 and gf4 (Supplementary Fig. 7; Supplementary Movies 7–10). For gf1, gf2, and gf4, Phase I corresponds to the entry of the colonizing *E. coli* to the gut (until ~2 d) (Fig. 2e). This phase is also manifested in the rapid increase of the barcode diversity (Supplementary Fig. 5b), which interestingly lags the peak in bacterial load occurring at ~6 h (Fig. 1e). Phase II of gf1, gf2 and gf4 is the rise of the dominant clonal clusters C1 or the start of the clonal sweep and the relative stasis/flatness of the other clonal clusters (Fig. 2e). This also corresponds to the "shoulder" in the frequency-weighted barcode diversity (diversity index q = 1 in Supplementary Fig. 5b). Phase III and phase IV is the establishment of the sweep by C1 accompanied by the decrease in low-frequency clonal clusters (Fig. 2e). For gf3, our first sample at 3 h already contained a large fraction of the barcode diversity (Fig. 2e third panel; Supplementary Fig. 5b third panel). Indeed, this leads to distinct dynamical features from the DCM analysis (Supplementary Fig. 7e). We also identified 3 phases in gf3, where Phase I is characterized by the relative stasis in the barcode dynamics, a feature seen in the Phase II of gf1, gf2, and gf4. Phase II of gf3 also exhibit the rise of clonal cluster lineages and Phase III is the establishment of the dominant clonal cluster (Fig. 2e third panel). Overall, the DCM analysis of the gf cohort shows that its community (strictly on intra-species interaction) matrix has fewer eigenvalues in the unstable regime compared to the rm cohort's community interaction matrix.

DCM applied to the im cohort revealed a lack of distinct phase separation that is reproducible across its 4 mice (Supplementary Fig. 8a–d, Supplementary Movies 11–14), contrasting with what we observed in the gf and rm cohorts. This suggests that the complex structure of the innate microbiome makes it less susceptible to colonization, as indicated by the two principal components (PCA1 and PCA2) (Supplementary Fig. 8a–d, and Supplementary Movies 11–14). Particularly in im3 and im4, we noticed that *E. coli* invasion drove an early shift from a stable to an unstable microbial state—a pattern less evident in im1 and im2 (Supplementary Movies 11–14). This variation could imply different levels of susceptibility to microbial disturbances within the im cohort. Moreover, we observed that changes in the stability of the microbial community correlated with increases or decreases in the abundance of some families in the innate bacteria. For instance, on day 4, im1 exhibited instability when the frequency of a species decreased by one-fold (Supplementary Movie 11). Additionally, removing *Enterobacteriaceae*, which includes the introduced *E. coli*, led to more consistent eigenavalue changes across im mice, as well as enhanced community stability as estimated by DCM (Supplementary Fig. 8e–h, and Supplementary Movies 15–18). This finding confirms the inherent resilience of the im cohort since it still retains its innate microbiome and is more diverse compared to other cohorts.

In the nc cohort, we noted a two-phase dynamic in the microbial communities around the 5- or 6-day mark, where the variance in community stability remained constant, indicating that a dynamic equilibrium was reached (Supplementary Fig. 8i–l, and Supplementary Movies 19–22). This stability represents the recovery from the community perturbation caused by a month-long antibiotic treatment in mice prior to gavage, as *Lactobacillaceae* initially dominated the environment post-antibiotic treatment, while around days 3-4, other species began to increase.

## Specific interactions between *E. coli* clonal clusters and other bacterial families

The DCM analysis showed strong coupling of the intra-species clonal cluster dynamics and inter-species interactions in the rest of microbiome community of the rm cohort. The introduction of *E. coli* in the gut microbiome led to a reduction in the abundance of some resident

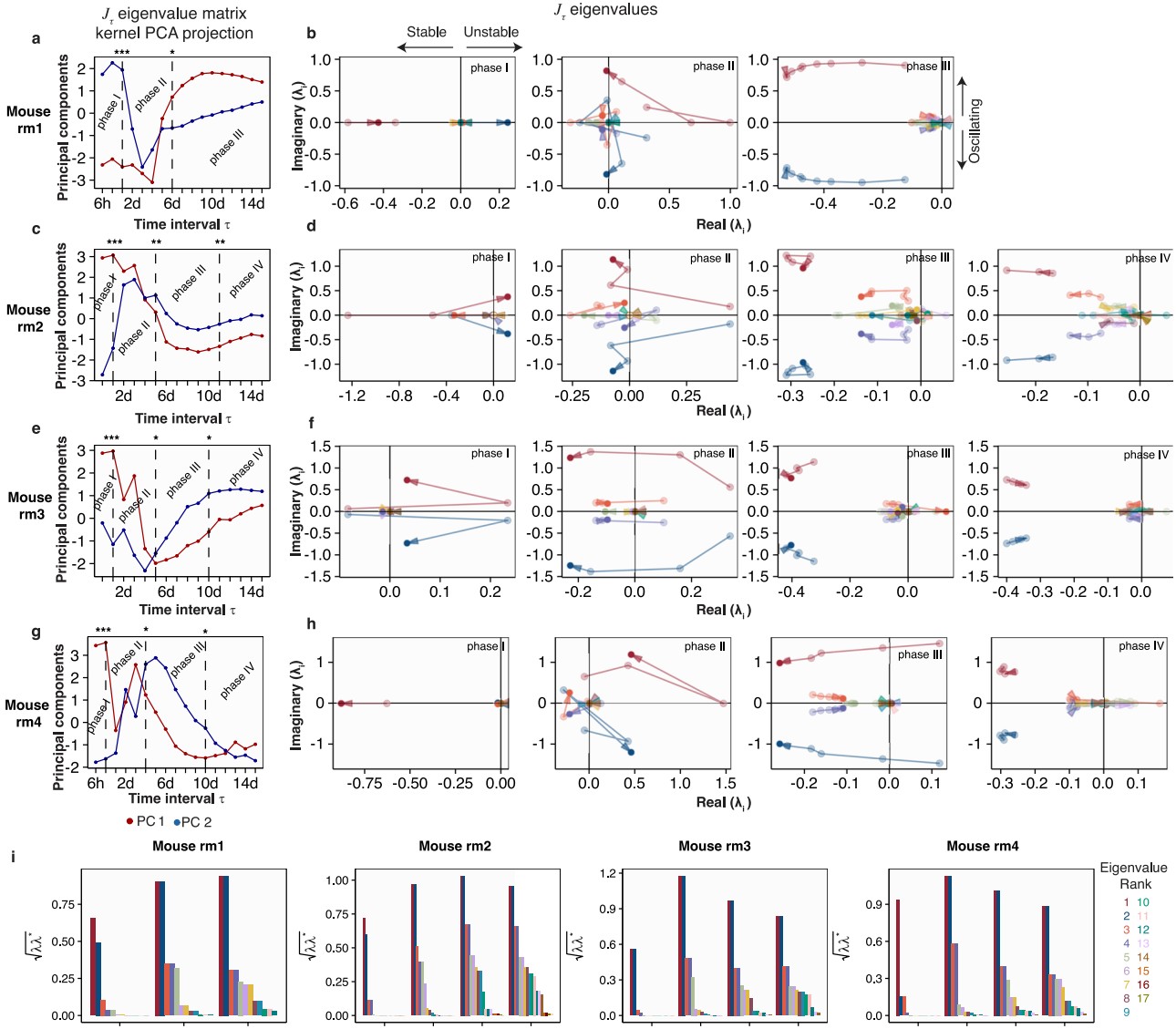

**Fig. 3 | Dynamic covariance mapping on the rm cohort defines distinct phases of colonization. a**, **b** Principal components (PC1 and PC2) of the eigenvalue matrix of $J_\tau$ over the time interval $\tau$ for mouse rm1. A change in the direction of either PC1 or PC2 is indicative of a dynamical change in the community. Using this criterion, three distinct phases are identified based on the eigenvalues of $J_\tau$ and their dynamic interpretation stable/unstable or oscillatory (**b**). The eigenvalue colors correspond to the rank of their magnitude (panel **i**). Phase I is transient instability (positive real eigenvalues) corresponding to the entry of *E. coli* and the collapse of resident

bacteria. Phase II is the return to a stable regime (negative real eigenvalues) and the re-emergence of the community species. Phase III is quasi-dynamic equilibrium with both oscillations in the clonal and community dynamics. Dynamic sub-feature in phase I (*) correspond to the first 6 h of entry of the colonizing *E. coli*. See Supplementary Movie 3. **c–h**, Stability analyses and phases for rm2, rm3, and rm 4. In these mice, Phases IV is the quasi-dynamic equilibrium with co-existence between *E. coli* and the rest of the community. **i** The magnitude of the eigenvalues.

bacterial communities in the rm cohort (16S rRNA in Fig. 2c). This quick initial collapse happens within the first ~3 h and is most clearly manifested in rm2 (16S rRNA in Fig. 2c). The eventual establishment of *E. coli* was accompanied by the resurgence of the bacterial community around day 4, followed by the coexistence of *E. coli* and the resident bacterial community. Notably, *Sutterellaceae* was unperturbed by the introduction of *E. coli* in all four mice (Fig. 2c). *Acholeplasmataceae* was unperturbed in rm 1 and rm 3, and it was the first bacterial family to rebound in rm 2 and rm 4 (Fig. 2c). The canonical member of a gut microbiome, *Lactobacillaceae*, had an intermediate abundance when *E. coli* was introduced, but it declined after *E. coli* established (Fig. 2c). In the no-colonization cohort (nc), the community also exhibited a rebound due to the release from the antibiotic treatment (Fig. 1g). However, the resident community of the nc cohort was dominated by *Lactobacillaceae* and was counterintuitively less diverse than the

resident community of the rm cohort. These differences in the *nc* and *rm* cohorts suggest of the potential impact of *E. coli* introduction on bacterial community composition. The resurgence of the resident community has also impacted the *E. coli* intra-species dynamics characterized by a stabilization of clonal diversity after day 4 (Supplementary Fig. 5a).

To determine whether an individual clonal cluster was associated and potentially interacting with specific bacterial families, we performed co-clustering of the dynamics of clonal clusters of *E. coli* and those of the 16S rRNA bacterial community profiles using the shape-based algorithm[72] (Fig. 4a, and Supplementary Information C Supplementary Fig. 14). These analyses revealed a consistent picture across the rm cohort. The dominant cluster, C1, was always grouped with *Lachnospiraceae or Enterococcaceae*, whereas other low-frequency clusters grouped with *Lactobacillaceae*, the canonical member of gut

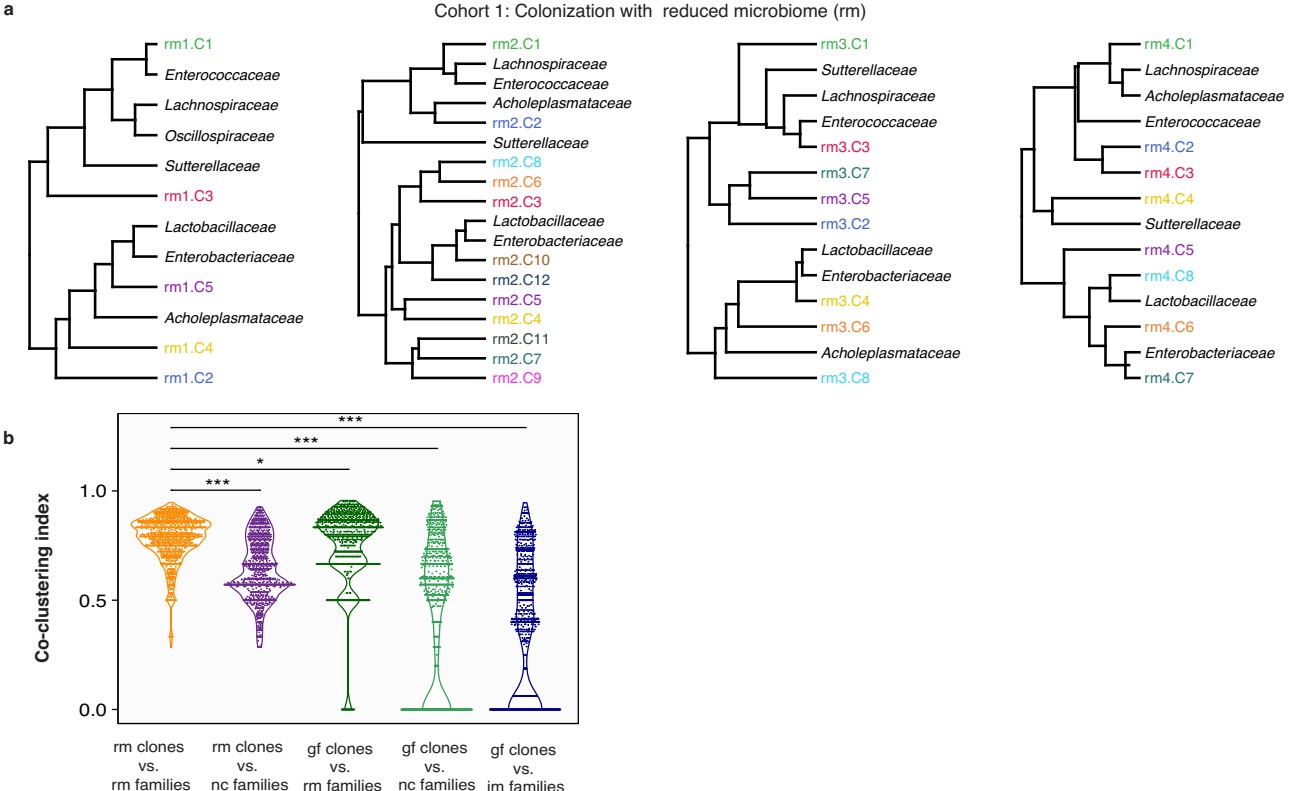

**Fig. 4 | Lineage-specific community interactions in the gut for the rm cohort.** **a** Co-clustering *E. coli* cluster clones with the different bacterial families suggest clone-specific community interactions. The clonal lineage-to-species interactions are broadly reproducible across different mice, whereby the dynamics of C1 is related to *Lachnospiraceae* in rm1-4, and C8 of rm1-2 and C10 of rm3-4 are related to *Lactobacillaceae*. **b** *E. coli* clone and bacterial community interactions are strongest when coming from the same cohort. Co-clustering is measured by a mixing coefficient that compares the distances in the hierarchical tree among families with distances between families and clonal clusters (Methods, Supplementary Information section C). Statistical significance was assessed using a two-sided Wilcoxon rank-sum test ($n = 1276$). Significance is indicated as follows: $p \leq 0.05$ (*), $p \leq 0.001$ (***).

microbiota (Fig. 4a). Interestingly, it was previously shown in colonization studies of pathogenic strains of *E. coli* and *Lachnospiraceae* that these bacteria utilize similar sugars and thrive in the same environment[73]. *E. coli* has also been observed to exhibit co-colonization patterns with *Enterococcaceae*[74]. Additionally, *E. faecalis* can promote the growth and survival of *E. coli* both in vitro and within mouse gut through the production of L-ornithine[75].

To demonstrate that the degree of co-clustering of *E. coli* clonal and bacterial families time-series was specific to the rm cohort, we applied the co-clustering analysis to other pairs of clonal clusters and family time-series from different cohorts of mice (Fig. 4b). The extent of co-clustering was measured using the mixing index, $D_{c,m} = 1 - \left(\max|F(c) - F(m)|\right)$ (Fig. 4b and Supplementary Data Fig. S4), which compares the clustering distance from clonal lineages to bacterial families $F(m)$ with the distance between clonal lineages $F(c)$ (Fig. 4b). Indeed, the co-clustering between clone and bacterial family time-series was strongest in the rm cohort (Fig. 4b). Expectedly, co-clustering was weakest when the 16S rRNA community dynamics of im and rm were paired with the gf cohort clonal lineages (Fig. 4b). Altogether, the co-clustering between *E. coli* clonal lineages and bacterial families is strongest when they come from the same biological cohort (Fig. 4b), suggesting intra- and inter-species interactions.

**The dynamic similarity is driven by similar chromosomal barcodes**

How similar are the clonal clusters across different mice, and are they driven by the same barcodes? To address this question, we performed pairwise clustering using the Pearson correlation to measure the

similarity of the clonal cluster time-series. In both rm and gf cohorts, we observed that the dominant clonal clusters (C1 and, to a lesser extent, C2) have similar time-series (Fig. 5a, d). By calculating the overlap coefficient between barcodes in each clonal clusters, we found that the dominant clonal lineages are more likely to be comprised of the same barcodes, an observation that is notably stronger in the gf than rm mice (Fig. 5b, c for gf and Fig. 5e, f for rm). The reproducibility of dominant barcode dynamics could be driven by standing genetic variation in the colonizing population, potentially followed by the selection of similar de novo mutations. Additionally, the similarity in barcode dynamics and composition is understandably weaker in the rm than in the gf mice, where the effects of standing genetic variation and/or de novo mutations in rm are modulated by ecological interactions with a changing bacterial community. These observations are consistent with the findings of the DCM analysis, where the dynamical features of stable/unstable, including the presence of distinct temporal domains are largely consistent among mice of different cohorts.

**Whole-genome sequencing reveal mutations related to motility loss, carbon metabolism and the TCA cycle**

To uncover the genetic underpinnings behind the dynamics of the clonal clusters, we conducted whole-genome sequencing (WGS) (Fig. 6). For the initial population (gavage sample), metagenomic sequencing was utilized to assess the breadth of population genetic diversity, achieving a resolution of at least ~1% (Fig. 6). In the case of the experimental endpoints, our objective was to associate genomic mutations and the barcodes of predominant clonal lineages; thus, WGS was executed on individually picked colonies. We cultured the homogenized fecal samples from gf and rm cohort mice on plates

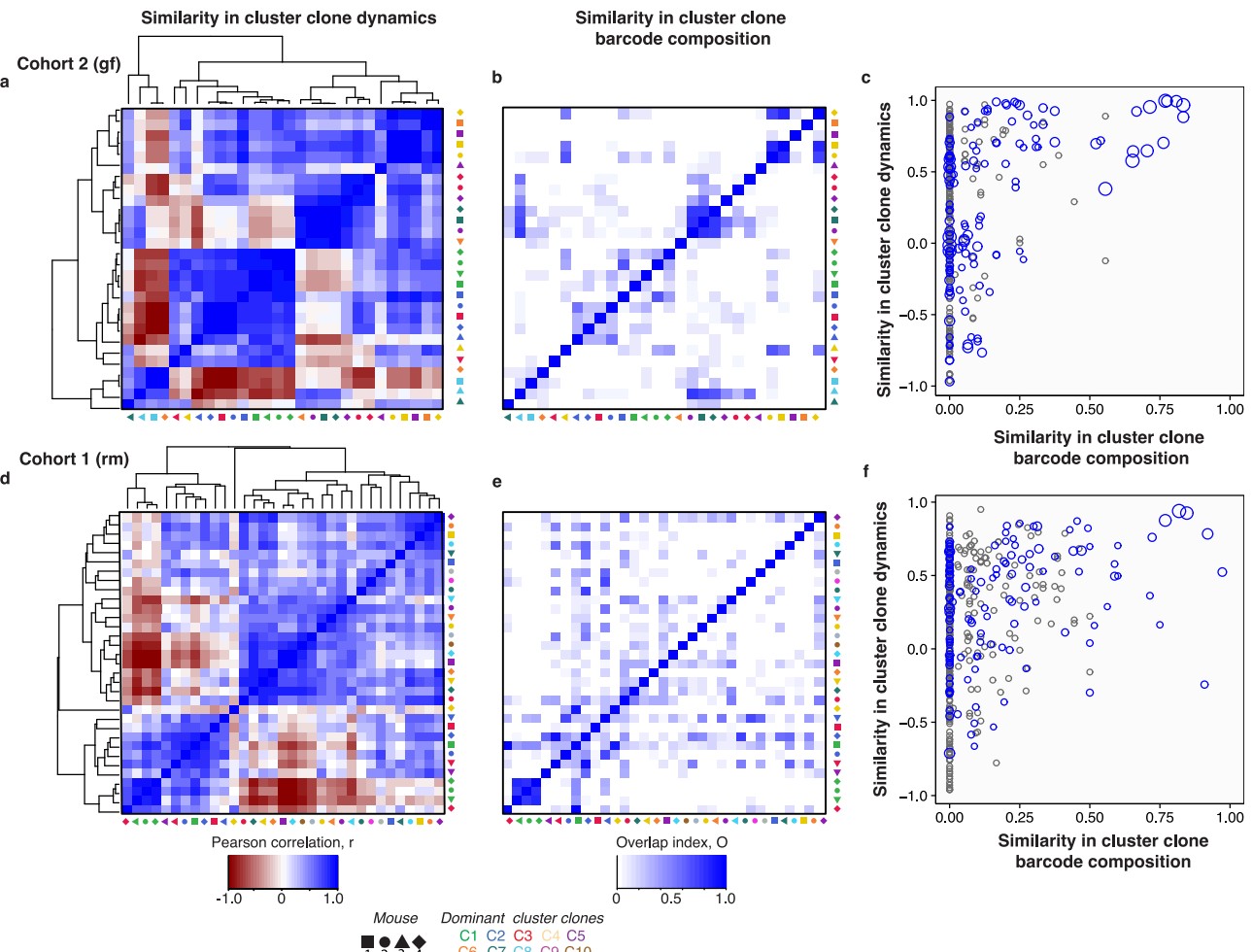

**Fig. 5 | The similarity of clonal lineage dynamics across mice is partly driven by identical barcodes. a** Similarity between the time series of clonal clusters across all 4 gf mice quantified by Pearson correlation. Matrix elements are clustered based on hierarchy (dendrograms indicated). Colors indicate the clonal cluster's identity, while the shape indicates the mouse of origin. **b** The similarity in barcode identity between the different clonal cluster is quantified by the overlap coefficient, $OC(A, B) = |A \cap B|/\min(|A|, |B|)$, where A and B are the sets of unique raw DNA barcodes that belong to two dominant clonal clusters. Identities of matrix elements are similar to panel (**a**). **c** Scatter plot of the similarity in dynamics between two clonal clusters by Pearson correlation vs. similarity in their barcode identity by overlap coefficient. Overlap coefficients satisfying two-sided $P$ value <0.05, calculated from z-scores derived via bootstrap resampling, are shown in blue; otherwise, they are shown in gray (Methods). The size of the circle is proportional to the significance of the overlap coefficient. **d**–**f** Similarity in dynamics and barcode identity for the colonization in mice with the resident microbiome.

containing spectinomycin, the barcode selection marker. Subsequently, we randomly selected 4 colonies from each mouse for Sanger sequencing of the barcode regions (see Supplementary Data 1). Due to the clonal sweeps, the randomly screened barcodes corresponded to the dominant lineage cluster C1 for all mice, except for gf2, which belonged to the second largest lineage cluster C2. For each mouse, we performed WGS on two clones, each possessing different barcodes, except for gf3, where only one clone was sent since all four screened colonies shared the same barcode (Supplementary Data 1, Methods). We also measured the fitness of these clones in M9 minimal media (Supplementary Fig. 10a, b), with the caveat that M9 does not fully capture the nutrient composition, host factors, and dynamic interactions occurring in the gut environment. Based on cumulative biomass (Supplementary Fig. 10c), the colonies from clonal cluster C2 have expectedly lower fitness than all those from C1.

From the WGS, overall, we identified mutations following colonization that were common to both rm and gf cohorts (Group I), or exclusive only to either gf (Group II) or rm (Group III)((Supplementary Data 2, Methods). These de novo mutations, consistently identified across different mice and individual colonies, hint at their adaptive significance. Several of these mutations, linked to motility, biofilm

production, and fundamental metabolic functions (carbon metabolisms and TCA cycle regulation), were also reported in earlier investigations of *E. coli* colonization in germ-free and reduced microbiota environments[37,67,76].

The Group I mutations are those shared among multiple replicates of both rm and gf cohorts, potentially driving adaptive responses to the gut milieu, independent of the resident microbiota or species-specific interactions. Notably, an IS1 deletion spanning ~15–16 kb from *flhE* to *flhD*, encompassing 17 or 18 genes related to bacterial motility, was identified in clones from gf3-4 and rm1-4. In the rm cohort, one barcode appeared in multiple mice (TCGTAACTAAGGCTT in Supplementary Data 1 and, in Fig. 6, rm2.C1. b1, rm3.C1.b1, and rm4.C1.b1 (notation indicates the isolated genome's barcode (b1), clonal cluster lineage (C1) and mouse replicate)). This barcode exhibited the exact same ~16.5 kb deletion, suggesting that the deletion could have been present in this specific barcode prior to gavage, although not seen in the metagenomics because of resolution. On the other hand, in the gf cohort, another barcode (ATACAACGTGGTAGC in Supplementary Data 1 and gf1.C1. b1 and gf4.C1.b1/b2 in Fig. 6) also appeared in multiple mice. However, only gf4.C1.b1/b4 exhibited a deletion at this locus (~6.2 kb), while gf1.C1.b1 did not, suggesting that

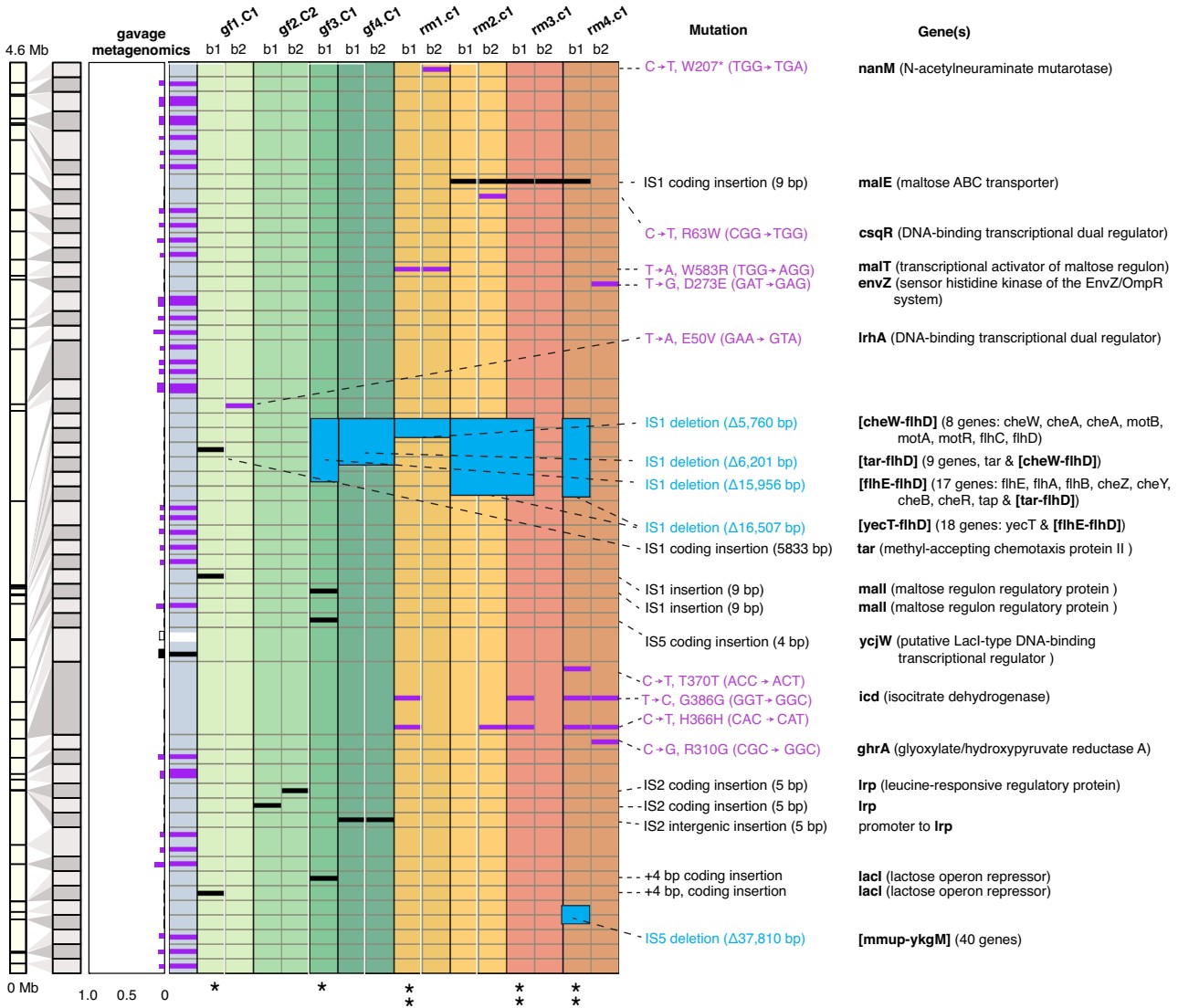

**Fig. 6 | Whole genome sequencing (WGS) of dominant clonal lineages reveals mutations leading to motility loss and in genes involved in sugar and carbohydrate metabolism and biofilm formation.** Population metagenomic WGS was performed on the gavage sample, while WGS was performed on single colonies of the fecal samples from gf and rm mice at the end of the colonization experiment. 32 colonies were initially randomly picked (4 per mouse), then their chromosomal barcode was identified to associate them with a clonal lineage (Supplementary Data 1). Almost all clones belonged to the dominant lineage C1, except gf2 with

clones belonging to the second dominant lineage C2. Unique chromosomal barcodes (labeled as b1 and b2) of the 2 screened per mouse were sent for WGS. Only high-confidence mutations are shown insertions (black), deletions (blue), and point mutations (purple). We found mutations that are present only in the gf cohort, rm cohort, or both. Some of these mutations, related to motility, biofilm formation, and core metabolic processes, were also observed by previous studies of *E. coli* colonization in germ-free and reduced microbiota[37,67]. See main text for detailed description.

the mutation is likely be de novo. Indeed, another barcode found exclusively in gf3 displayed a different form of deletion (~15.9 kb) at this locus, further supporting its de novo origin. These type of expansive motility gene deletions within the *flhE-flhD* region has been documented in prior studies of *E. coli* colonization of gf mice[67,76] and mice with "stripped" microbiota[77], and was shown to result in ~90% reduction in *E. coli* motility following about two weeks of gut colonization. This deletion, and consequent loss of motility, might allow *E. coli* to reallocate energy resources previously used for flagella synthesis and function towards growth. Furthermore, *flhD* and *flhC* proteins, both transcriptional factors, play roles in glucose and fructose uptake and metabolism, as well as other carbon metabolic pathways[78]. A knockout of these genes indeed led to growth retardation, coupled with increased ATP and NADPH production, shifting carbon flux towards the TCA cycle and pentose phosphate pathways[79].

The Group II mutations are exclusive to the gf cohort, including a 4-base pair insertion within the *lacI* gene (a DNA-binding transcriptional repressor), observed in sequenced clones of gf1.C1 and gf3.C1 (Fig. 6). This frameshift inactivation of *lacI*, noted in previous *E. coli* gut colonization studies[67,80], might activate the Lac operon, facilitating lactose breakdown into glucose and galactose. Vasquez et al. also repeatedly identified a 4-base pair insertion in the *lacI* gene during gf mice colonization experiments, albeit only 16 bp upstream the gene locus. A related mutation, *ycjW*, a *lacI*-type transcriptional repressor, affects carbohydrate metabolic gene expression and thiosulfate sulfurtransferase PspE[81], influencing hydrogen sulfide (H₂S) production, which could act as a defensive mechanism against host immune responses[82]. Additional mutations include changes in the *lrp*, a DNA-binding transcriptional dual regulator, with 5-bp deletions observed in intergenic regions (2 clones of gf4) and coding sequences (2 clones of

gf3). Lrp proteins, recognized for their feast/famine regulatory functions, respond to environmental nutritional cues, and regulate metabolism, virulence, motility, and stress tolerance genes, essential for niche adaptation[83,84]. These mutations were also observed in earlier studies. For instance, Barroso-Batista et al. found that such genetic alterations improve *E. coli*'s amino acid competitive uptake, with serine and threonine identified as preferred metabolites in a mono-colonized mouse gut[37]. Growth assays in vitro and in vivo confirmed this serine preference and the diminished advantage of *lrp* mutants impaired in serine metabolism[37]. Mutations were also observed in the global regulator *lrHA*, controlling several genes associated with motility, chemotaxis, and flagellum assembly by directly regulating the master regulator *flhDC*[85]. Mutations were also discovered in the maltose regulon regulatory protein MalI, previously reported with established adaptive significance[76]. We hypothesize that inactivating this repressor gene could enhance *malT* (maltose transcriptional activator) activation, serving as a mutational target for gut adaptation.

The final set of mutations (Group III) were uniquely identified in the rm cohort relates to disruptions within the maltose regulon: *malT*, the DNA-binding transcriptional activator of the maltose regulon, and *malE*, a periplasmic maltose-binding ABC transporter protein. De Paepe et al. found that bacteria that lost the ability to use maltose are often selected following the colonization by *E. coli* pathogenic strain 536 in antibiotic-treated mice[76]. This observation suggests that inhibiting the maltose regulon may confer adaptive advantages under certain intestinal conditions. Additional Group III mutations are in the *CsqR* gene (also known as *YihW*), a DNA-binding transcriptional dual regulator[86] that suppresses catabolism of sulfoquinovose (SQ), a derivative of sulfoquinovosyl diacylglycerol (SQDG) breakdown[87]. *CsqR* likely participates in regulating lactose metabolism, operating either in synergy with or contrary to cAMP-CRP, a master regulator of carbohydrate metabolism, serving as a sugar-responsive dual regulator[88]. Sulfoquinovose (SQ) is a sulfonated monosaccharide integral to the plant sulfolipid sulfoquinovosyl diacylglycerol (SQDG). As a constituent of plant chloroplast membranes, SQ is prevalent in the environment, attributed to the massive decomposition of plant matter. Bacteria, notably those within the gut microbiome, contribute to SQ degradation as part of the broader process of decomposing plant substances. Certain bacteria are capable of metabolizing SQ to derive sulfur. These bacteria possess specialized enzymes that decompose sulfoquinovose, leading to the release of sulfite, a byproduct of their metabolic activities. This capacity of specific intestinal bacteria to generate $H_2S$ from substrates like SQ may impact *E. coli* interactions with other microbial populations.

Most intriguingly, we identified three synonymous mutations within isocitrate dehydrogenase *icd*: one found in all four rm mice (H366H), another in three mice (G386G), and another in a single mouse (T370T) (Fig. 6). None of these were present in the gavage sample. Isocitrate dehydrogenase is key enzyme of the TCA cycle, a crucial process in bacteria for energy production and biosynthesis. Traditionally, synonymous mutations have been viewed as neutral. Nonetheless, mounting evidence suggests this may not be the case, as demonstrated in recent studies. Marx and colleagues demonstrated that high-impact beneficial synonymous mutations in an essential metabolic enzyme (Fae, which encodes the formaldehyde-activating enzyme) prompted rapid, parallel adaptation in *M. extorquens*[89]. In a parallel vein, repeated synonymous mutations occurred in the permease subunit of a glucose-inducible ATP-binding cassette (ABC) transporter, *gtsB*, a vital metabolic enzyme responsible for glucose uptake[90,91]. Given these observations, we speculate that mutations in *icd* could influence the regulation of this essential metabolic pathway, potentially leading to changes in energy production and resource distribution.

We hypothesize in the Discussion how such mutations may lead to the highly reproducible interaction of the C1 clonal cluster with *Lachnospiraceae*. One potential molecular mechanism by which synonymous mutations in the *icd* gene could be adaptive is the modulation of ICD abundance via co-translational folding. Jacobs et al[92]. have analyzed the association between the intermediate folding conformations of the nascent polypeptide chains in the *E. coli* proteome and the enrichment of rare codons as a sign of translation pauses to assist with co-translational folding. They detected that such an association exists around the amino acid position ~350 to ~390 of the ICD protein[92], the same region that were mutated in clones from the rm mice colonization (Fig. 6). The mutations in the *icd* gene could potentially be responsible for the pervasive C1-*Lachnospiraceae* association found in the rm mice. *Lachnospiraceae* is the main producer of short-chain fatty acids (SCFA) (propionate and butyrate) in the gut microbiome[93]. SCFA, which are utilized via the TCA cycle, were shown to be important for *E. coli*'s invasiveness of the gut microbiota[94]. It is therefore plausible that *icd* mutations that are specific to the rm mice underly the similarity between C1 and *Lachnospiraceae* dynamics.

Altogether, the presence of mutations that could be part of standing genetic variation (Group I, rm cohort's *flh* loci deletions) and the repeatability of selected de novo mutations between cohorts and within cohorts (Groups II and III) corroborate the observed repeatability of barcode dynamics (Fig. 5) and, importantly, of the community interaction's stability analysis by DCM for both the gf and rm cohorts (Fig. 3, Supplementary Fig. 7).

## Discussion

Our experimental and computational framework offers a generalized approach to quantify microbial community interaction matrix and its consequences on dynamics and stability, particularly following perturbations triggered by invading species. With our experimental barcoding protocol, we demonstrate that clonal dynamics from intraspecies variation can be used to estimate time-dependent interactions, even during the very early stages of community colonization. Although the dynamics are complex, the global colonization dynamics are surprisingly replicable and can be defined by 3 or 4 phases that arise from the coupling of ecological and evolutionary dynamics. This seems contradictory to the reported lack of reproducibility and replicability of microbiome composition across mice replicates[95]. However, we note that despite only after two weeks of colonization, we already observed diverging mutations across the cohorts. We cannot yet comment on the long-term implication of intra-species variation at the resolution afforded by this experiment since our barcode diversity is exhausted after a clonal sweep. This would require a "renewal" or regeneration of new DNA barcodes, as recently done in yeast[96].

Previous studies have also employed DNA barcoding to investigate the dynamics of gut invasion, but typically with a lower barcode diversity than our method offers. For instance, Vasquez et al. utilized ~200 barcodes in plasmids to explore colonization lineage dynamics in germ-free mice[67]. In a similar vein, STAMP (sequence tag-based analysis of microbial populations) was used to demonstrate the complex spatio-temporal dynamics of *Vibrio cholerae* infection along the rabbit gut, as well as to quantify the bottlenecks in *Citrobacter rodentium* in mice[97,98]. Additionally, Barroso-Batista et al. used an isogenic *E. coli* population, differentiated only by chromosomally encoded YFP/CFP fluorescence markers, to track adaptive mutations and their fitness effects in mice[71]. Similarly, Grant et al. explored the spatiotemporal dynamics of invasive bacterial disease using 8 barcoded strains of *S. typhimurium*, though the low barcode count limited their ability to quantify very low-frequency lineages and their correlation with the microbial community[99]. Our approach contrasts with these methods by providing a much higher DNA barcode diversity. More importantly, the use of high-density chromosomal barcoding to investigate the inter- and intra-species ecological dynamics of species-rich communities in natural environments, such as the gut, remains unexplored. Altogether, despite varying details in the techniques, we anticipate that

DCM could be useful in the analysis of abundance time-series arising from these other lineage-tracking approaches.

Our chromosomal barcoding approach could be extended to species that are innate to the gut microbiota or more pathogenic bacteria. The high-resolution colonization dynamics could also be extended by barcoding pathogenic species, such as *P. aeruginosa* and *S. enterica*, which are more aggressive colonizers than *E. coli*. Recently, the successful barcoding of Pseudomonas fluorescens SBW25[100], demonstrates the potential applicability of our approach for generating a library of more than $10^5$ barcodes. Additionally, our use of the Tn7 integration site, conserved among Gram-negative bacteria, suggests that our method could be easily extended to other bacteria in this group. The transformation step of the two plasmids used for barcoding often presents a bottleneck, thus, enhancing the competency of cells and the transformation procedure could be necessary for each new bacterial species. Additionally, the methodical and computational insights from chromosomal barcoding could be extended to related approaches in microbiology, such as the more robust quantification of fitness changes in colonization of mutational libraries in Tn-seq experiments[101,102].

In the current work, we use the term "interaction" in the ecological sense, particularly in the context of the community matrix, which is the correlation between the abundances of community members. We then aimed to elucidate, with the WGS data, what molecular/genetic mechanisms might mediate such interactions, at least for the dominant lineage clusters. Determining the molecular/genetic mechanisms for the low-frequency lineage clusters is currently experimentally challenging, as metagenomic WGS is shallow, and screening individual colonies followed by WGS is biased by the high-frequency clones. To overcome this challenge in the microbial eco-evolutionary scientific community, bacterial chromosomal barcoding needs to be coupled with techniques for enrichment and isolation of in live cells with rare barcodes, potentially, by utilizing CRISPR/Cas9, similar to methods used in mammalian cancer cells[103–106].

We also note that the goal of this study is to correlate intra-species barcode dynamics with the bacterial community, thus, the barcode cluster clone analysis focused on persistent barcodes, defined as those present in 75% of the timepoints. Consequently, typically only the most frequent ~5% of the barcodes survive. A significant number of barcodes that become extinct in the initial days of colonization are not included in the analysis. Additionally, the goal is to simplify the representation of intra-strain diversity to model lineage dynamics more effectively. Treating each barcode as a separate sub-lineage would either be computationally impractical or introduce excessive multiple hypothesis testing challenges.

In this work, the gut microbiome is treated as an ecological system, such that all the approaches presented here could be broadly applicable to most microbial ecological networks. However, the gut microbiome has particularities. More specifically, the gut microbiota itself is shaped by the genetics and phenotypes of the mice, which we do not explore. Indeed, the mice themselves, in general, are not homogenous and could have an impact on the gut composition. In human microbiomes, it was shown that genetic variation in humans could itself impact the diversity of the microbiomes[107]. In the future, the impact of host diversity could be explored by performing colonization experiments in mice with diverse genetic backgrounds.

Broadly, the DCM that we developed here represents a model- and parameter-free approach to analyzing the stability and distinct temporal phases of a microbial system, starting simply from high-resolution time-series abundance data. In the Supplementary Information, we provide illustrative examples of how DCM captures the interaction strength matrix of general Lotka-Volterra (gLV) models. However, since the gLV model assumes no mutation, no intra-species heterogeneity, no migration (colonization), and a constant environment, it cannot capture the complexities of coupled ecological-

evolutionary dynamics, such as those occurring during gut microbiome colonization. In gLV, the interaction strength matrix is constant, but this is not necessarily the case when there is a supply of mutations or when environmental conditions fluctuate. The conceptual essence of gLV is that it links the per capita growth rate of a species (or community member) to the abundance of other community members. This concept is also central to DCM's Jacobian matrix analysis, which is the covariance between the derivative of a community member's abundance time-series (i.e., its growth rate) and the time-series abundance of another community member. However, unlike gLV, DCM does not assume that the interaction strength matrix within the community is constant. Therefore, DCM could serve as a general framework for analyzing coupled ecological-evolutionary dynamics, extending even beyond gut microbiomes. As such, the DCM is also available to the general microbial community via *Github* (Methods).

The strength of DCM is that it non-parametrically estimates the community matrix solely from abundance time-series data. However, this also presents its potential weakness, since DCM strongly depends on the quality of the time-series. In particular, the sampling frequency of the abundance needs to sufficiently capture the richness in community dynamics (see Supplementary Note section B (Fig. S3) on the effect of sampling rate on LV model communities). Nonetheless, DCM provides a complementary quantitative analysis to vast microbiome time-series data, whose resolution, quality, and mode of collection we anticipate improving in the future.

Finally, our results also showed that these phases of invasion and the intra- and inter-species interactions are highly reproducible across mouse replicates. This is rather unexpected, considering the variability in microbiome compositions, which is the norm in the microbiome field[108]. We argue that although specific compositions may be highly variable across mice, the overall tempo of ecological and evolutionary dynamics, as manifested by the DCM analysis, are more reproducible features of the microbiota. To this end, the DCM and its future incarnations could provide a framework for predicting the microbiota's response to perturbations, especially in the context of the invasion of pathogenic species[109] and fecal transplant to treat human disorders[110].

## Methods

### Dynamic covariance mapping: extended description

**Eigenvalue decomposition and stability analysis.** Since $J = \text{diag}(\mathbf{z})\mathbf{A}$, where $\text{diag}(z)$ is a diagonal matrix with $z_i$ as the diagonal elements, Eq. 2 can be rewritten as $\dot{\mathbf{z}}(t) = J\mathbf{z}(t)$, after a change of variable where the abundance $\mathbf{z}$ is now $(\mathbf{z} - \mathbf{z}(t_*)$[11]. Mathematically, the matrix $J$ transforms the abundance vector $\bar{\mathbf{z}}$ into a new vector space. By definition, $JV = J\Lambda$, where $\mathbf{V} = [\mathbf{v}_1, \ldots, \mathbf{v}_i, \ldots, \mathbf{v}_n]$ is the eigenvector matrix with the eigenvector $\bar{v}_i$ (in the ith column). The eigenvector $\boldsymbol{v}_i$ corresponds to the eigenvalue $\lambda_j$, such that $J\mathbf{v}_i = \lambda_j\mathbf{v}_i$. $\Lambda$ is the eigenvalue matrix with diagonal elements $\lambda_j$ and zero elsewhere. The solution to $\dot{\mathbf{z}}(t) = J\mathbf{z}(t)$ is $\mathbf{z}(t) = \mathbf{z}(t_*)e^{Jt} = \mathbf{z}(t_*)\mathbf{V}e^{\Lambda t}\mathbf{V}^{-1}$, which can be written as $z_i(t) = \sum_{j=1}^{n} z_j(t_*)v_{ji}e^{\lambda_j t}$. These eigenvalues are, in general, complex numbers $\lambda_j = \theta + i\omega$. Using Euler's formula and taking only the real components $z_i(t) \propto \sum_{j=1}^{n} v_{ji}e^{\text{Re}(\lambda_j)t}\{\text{Cos}(\text{Im}(\lambda_j)t)\}$. Thus, the abundance time-series of the community members can be expressed as a linear combination of periodic functions with frequency $\omega = \text{Im}(\lambda_k)$ and amplitude $e^{\text{Re}(\lambda_j)t}$. Notably, when the real part of the eigenvalue is positive, the amplitude increases exponentially, indicative of instability, while if negative, the amplitude is bounded, indicative of stability. Additionally, the magnitude of the eigenvalues' imaginary component is indicative of characteristic oscillations in the community.

**Estimating the community interaction strength using covariance.** When the community abundance is sampled at time $t_1, \ldots, t_k$ within a time window $\tau$-width $(t_* - \tau/2, t_* + \tau/2)$ centered at $t_*$, we can have

$\mathbf{z_j}(t_*) = \left[z_j(t_1), \ldots, z_j(t_k)\right]^T$ as the abundance time-series for species $j$ and $\boldsymbol{\phi_i}(t_*) = \left[\phi_i(t_1), \ldots, \phi_i(t_k)\right]^T$ as vector of per-capita population growth rate of species $i$. According to the Frisch-Waugh-Lovell (FWL) theorem for solving a multiple linear regression using ordinary least squares, per-capita interaction strength $a_{ij}(t_*)$ can be estimated as $a_{ij}(t_*) = \mathrm{Cov}(\boldsymbol{\phi_i}, \mathbf{z_j})/\mathrm{Var}(\mathbf{z_j})$ if abundance fluctuations near $t_*$ are independent from each other[50,51]. Other regression methods lead to an altered estimate[51]; for instance, $a_{ij}(t_*) = \mathrm{Cov}(\boldsymbol{\phi_i}, \mathbf{z_j})/(\mathrm{SD}(\mathbf{z_j})\mathrm{SD}(\boldsymbol{\phi_i}))$ when the reduced major axis is used, while $a_{ij}(t_*) = \mathrm{Cov}(\boldsymbol{\phi_i}, \mathbf{z_j})$ if $\mathbf{z_j}$ is standardized. SD is standard deviation. In general, the estimate contains the covariance in the numerator that captures the strength and direction of the linear relationship between $\boldsymbol{\phi_i}$ and $\mathbf{z_j}$, while the denominator represents the unit for interpreting and comparing the covariance (e.g., the reduced major axis produces a unitless estimate which is Pearson's correlation). To ease comparison between species[11,52], we directly chose the covariance as the metric for interaction strength, $a_{ij}(t_*) = \mathrm{Cov}(\boldsymbol{\phi_i}, \mathbf{z_j})$. Similarly, the overall interaction strength $J_{ij}(t_*)$ can be estimated as $J_{ij}(t_*) = \mathrm{Cov}(f_i, z_j)$. Altogether, the dynamic covariances of $\mathrm{Cov}(\boldsymbol{\phi_i}, \mathbf{z_j})$ and $\mathrm{Cov}(f_i, z_j)$ provide us estimates of per-capita and overall interaction strengths of member $j$ on $i$ near $t_*$.

**Principal component analyses and determining dynamical boundaries by change-point analysis.** Dynamical analysis of eigenvalues typically focuses on the largest eigenvalues[54,55]. However, all eigenvalues can contribute to a system's behavior, resilience, and resistance[56–58]; thus, we instead analyze all eigenvalues. Since there are $n$ eigenvalues (corresponding to $n$ community members) for each time interval, we sought a simpler representation of the community's dynamic behavior. To this end, we used kernel PCA to reduce the dimensionality of the eigenspace as a function of time. The input to this kernel PCA dimensionality reduction is a $(2N) \times (s)$, where the $2N$ columns correspond to the real and imaginary components of the $i$th eigenvalue $\lambda_i$, while the $s$ rows is the number of $\tau$ time intervals. The first two principal components (PC1 and PC2) as a function of time intervals are then tracked to determine the dynamical shifts in the community (Fig. 1c,). The code and data are available on GitHub: https://github.com/melisgncl/Intra--and-inter-species-interactions-drive-phases-of-invasion-in-gut-microbiota-.

To explicitly identify shifts in PC1 and PC2, we employed change-point analysis, a statistical technique to detect points when the distributional properties (mean and variance) of the timeseries changes[59–64]. Specifically, we used the package *geomcp R package*[111] that implements change-point analysis on multiple timeseries. It also uses maximum likelihood estimation (MLE) to estimate the timeseries distribution parameters before and after a changepoint and determine the number and location of the changepoints. We use the following parameters: geo_result <- geomcp(*data*, penalty = *penalty_type*, test.-stat = *"Empirical"*, nquantiles = *nq*), where *data* is the combined timeseries PC1 and PC2. The underlying distribution of the timeseries is derived empirically (test.stat = *"Empirical"*) based on a specified number of quantiles (*nq*).

We performed several independent changepoint analyses, varying the parameter *nq* and, importantly, using different types of MLE penalties (*penalty_type*) available in *geomcp*: Modified Bayesian Information Criterion (MBIC), Bayesian Information Criterion (BIC), Schwarz Information Criterion (SIC), and Hannan-Quinn Criterion. These penalties help reduce the likelihood of overfitting, where too many irrelevant changepoints are detected, and underfitting, where critical transitions may be missed[111]. Changepoints that are consistently identified as significant by multiple MLE penalty criteria are denoted as (***), those identified in 90% of the independent changepoint analyses are denoted by (**), and those identified in 70% of the analyses are

denoted by (*). We then inspected whether the identified changepoints made intuitive sense. Details of the changepoint analyses are available in the GitHub link above.

## Dynamic covariance mapping: applied to intra-species and community abundance time-series

Depending on the cohort, our abundance vector $\mathbf{z}(t) = \left[z_1(t), \ldots, z_i(t), \ldots, z_n(t)\right]$ is composed of only community abundances from 16S rRNA (im and nc cohort), only E. coli intra-species (gf cohort), or both (rm cohort). Then, we calculate the community matrix as $J_{ij,\tau} = \mathrm{Cov}(\frac{dz_i}{dt}, z_j)_\tau$ progressively increasing time intervals $\tau$ (3h-6h, 3h–12h, …, and 3h-15 days), with altogether a total of 16 or 17-time intervals for the rm and gf cohort. Additionally, as the movies illustrate, the eigenvalues show distinct "jumps" on the PC1 and PC2, indicating distinct temporal phases. The codes and data on the application of DCM to gut invasion dynamics is: https://github.com/melisgncl/Intra--and-inter-species-interactions-drive-phases-of-invasion-in-gut-microbiota-. The code for application of DCM on a generic community time-series (user-generated) data is also provided in the same *Github* link.

## Experimental procedures

***E. coli* barcoded population generation.** Barcoded *E. coli* populations were generated as previously described[43] using the Tn7 transposon library. The first step is transforming the recipient *E. coli K12 strain MG1655* cells with the Tn7 helper plasmid and induction of the transposase integration machinery. The second step is the transformation of the Tn7 integration plasmid library, which integrates the barcodes into the chromosome of the bacteria. The Tn7 integration plasmids with barcode and spectinomycin cassette were extracted from Trans-forMax EC100D pir + cells (Lucigen) with a Qiagen midi kit. Then *E. coli* cells were transformed with the Tn7 helper plasmid to induce the transposase integration machinery. Transformed cells with Tn7 helper plasmid were grown overnight in LB supplemented with 100 µg/ml ampicillin at 30 °C. In these cells, transposon machinery was induced with arabinose to transform with Tn7 integration plasmids. After overnight incubation on the bench, they were plated on LB agar plates containing 100 µg/ml spectinomycin. Randomly picked colonies were checked for chromosomal incorporation of barcode cassettes by targeting the Tn7 integration site. We scraped all the colonies from the plates, then pooled, thoroughly mixed, and aliquoted them with 15% glycerol. These stocks were stored at −80 °C pending the mice colonization experiments.

**Mice evolution experiments.** We used several cohorts of mice to determine colonization dynamics in their gut: Cohort 1 (im) mice with innate microbiota followed by *E. coli* colonization (4 replicates); Cohort 2 (rm) or mice with reduced microbiota and pre-treated with an antibiotic cocktail followed by *E. coli* colonization (4 replicates); Cohort 3 (gf) or mice that were initially germ-free and colonized with barcoded *E. coli* barcode (4 replicates); and Cohort 4 (nc) or mice with microbiota and pre-treated with an antibiotic cocktail but not colonized by *E. coli* (4 replicates). Cohorts 2 and 4 (rm and nc) were administered an antibiotic cocktail (metronidazole 1 g/L, neomycin 1 g/L, ampicillin 1 g/L, and vancomycin 0.5 g/L) for four weeks to reduce the complexity of the gut microbiota. Under these conditions, 99.5% of the cecal bacteria are eliminated at the end of treatment[112,113]. Then, we let them recover for three days *without antibiotics* before introducing the barcoded population, which we set as our day zero. After gavage of the barcoded population, fecal samples were taken at 3, 6, 12, and 24 h and once daily until day 14 for rm and day 15 for gf. The nc cohort fecal samples were collected for ten days. During the day of fecal collection, we split the sample, one for bacterial load measurements (see below) and another for storage at −80 °C until subsequent genomic analysis. 80 µl of the feces homogenate was placed with 20 µl of 100% glycerol

to make 20% glycerol stocks for later recovery of live bacteria. Fecal samples are obtained directly from the mouse anus and placed into a pre-weighed, sterile 2 ml Eppendorf tube.

Over the course of the colonization experiments, individual mice were housed in different cages (1 mouse/cage and each mouse in the same cage). All cohort mice were C57BL/6 N (Taconic Biosciences), and only 12-week-old females were used. Animals were kept in filter-covered ventilated cages, maintained at 23 °C with 40% humidity, under a 12-h light/dark cycle. Cohorts were fed *ad libitum* with the Teklad 2018SX (Sterilizable) chow, an 18% protein diet that was sterilized by autoclave. All mice experiments were performed at the mouse facility of Université de Sherbrooke, except for the gf cohort which were performed at the gnotobiotic mouse facility of University of Toronto. All experimental protocols on mice were approved by Université de Sherbrooke Ethics Committee for Animal Care and by University of Toronto's Animal Care Committee, both in accordance with guidelines established by the Canadian Council on Animal Care.

**Bacterial load measurement.** To measure the bacterial load in the fecal samples, we spread them with increasing dilutions on LB plates with spectinomycin 50 µg/ml to select for the colonizing *E. coli*. The chromosomal barcode contains the spectinomycin resistance cassette (spR)[43]. Measurements of bacterial loads were done in 3 independent replicates.

**Genomic DNA extraction in fecal samples, chromosomal barcode amplification, and next-generation sequencing.** Genomic DNA (gDNA) was extracted from whole fecal pellets using the QIAamp Fast DNA Stool Mini kit (Cat: 51604). Low yield from gDNA extraction of samples from 3 h of im2 and rm 2,3 means that we could not continue with their downstream analysis. A two-step PCR was used to amplify the chromosomal barcodes and then append the Illumina adapter sequences. For the first PCR, 200 ng of template per sample was used with PrimeSTAR GXL DNA Polymerase from TAKARA (Cat: R050B). The parameters for this 1st reaction were as follows: 94 °C for 5 min, 30X (95 °C for 10 s, 53 °C for 15 s, 68 °C for 45 s), 68 °C for 5 min, hold at 4 °C. The Primers for this PCR are the following: 5'-TCGTCGGCAGCGTCAGATGTGTATAAGAGACAG-3', 5'-GTCTCGTGGGCTCGGAGATGTGTATAAGAGACAG-3'. The resulting amplicon sequence from this PCR is the following: 5'-gatatcggatcctagtaagccacgttttaattaatcagatccctcaatagccacaacaactggcgggcaaacagtcgttgctgattggtcgtcggcagcgtcagatgtgtataagagacagtcgcgccggNNNNNNNNNNNNNNNtatctcggtagtgggatacgacgataccgaagacagctcatgttatatcccgccgttaaccaccatcaaacaggattttcgcctgctggggcaaaccagcgtggaccgcttgctgcaactctctcagggccaggcggtgaagggcaatcagctgttgcccgtctcactggtgaaaagaaaaccaccctggcgcccaatacgcaaaccgcctctccccgcgcgttggccgattcattaatgcagctggcacgacaggtttcccctgtctcttatacacatctccgagcccacgagacgccactcgagttattgccgactaccttggtgatctcgcctttcacgtag-3'. The contiguous 15 *N*s in this amplicon sequence corresponds to the random nucleotides that serve as our chromosomal barcodes[43]. The product from this PCR was purified and cleaned with NucleoSpin Gel and PCR clean-up kit from TAKARA. A 2nd PCR was performed with high-fidelity PrimeSTAR GXL DNA Polymerase (Takara Cat: R050B) to add the Nextera indices (Nextera XT primers Set A 96 Indexes, 384 Samples, Cat# FC-131-2001). We followed the suggested cycling conditions, which are as follows: 94 °C for 5 min, 12X (95 °C for 10 s, 55 °C for 15 s, 68 °C for 45 s), 68 °C for 5 min, hold at 4 °C. The primers for this 2nd reaction were the following: 5'CAAGCAGAAGACGGCATACGAGAT[I7]GTCTCGTGGGCTCGG-3' and 5'-AATGATACGGCGACCACCGAGATCTACAC[I5]TCGTCGGCAGCGTC-3'. PCR products from all reaction tubes were purified with magnetic beads (Beckman Colter) and pooled together, spiked with 15% of PhiX DNA, and sequenced using either Miseq or Nextseq Illumina chips at Université of Montréal's IRIC Genomic Platform. Bioinformatic analyses are described in the Analysis section below.

**16S profiling.** Similar to the chromosomal barcode amplification, we used a two-step PCR to amplify the genomic region of interest and prepare the library for Illumina sequencing. The 16S rRNA V4 region was PCR-amplified with buffer and polymerase PrimeSTAR GXL DNA Polymerase (Takara, Cat: R050B). The cycling conditions for the PCR are as follows: 98 °C for 3 min, 35X (95 °C for 10 s, 60 °C for 15 s, 68 °C for 35 s), 68 °C for 5 min, hold at 4 °C. The primers for the reaction are the following: 5'-TCGTCGGCAGCGTCAGATGTGTATAAGAGACAGYRYRGTGCCAGCMGCCGCGGTAA-3' and 5'-GTCTCGTGGGCTCGGAGATGTGTATAAGAGACAGGGGACTACHVGGGTWTCTAAT-3'. PCR products were purified with Nucleospin Gel and a PCR purification kit from TAKARA (Cat: 740609). Illumina sequencing adapters were added to respective samples with PCR using the same primers and protocols similar to the barcode amplification. The PCR amplicons of the samples were then pooled after a purification and concentration equalization process with the AMPureXP Kit (Beckman Colter). The libraries were processed in an Illumina MiSeq v2 (500 cycles and paired-end).

**Clone isolation and growth rate measurements.** Glycerol stocks of fecal samples from the last time point of the gm and rm cohorts were resuspended in PBS. Subsequently, dilutions of these suspensions were plated on LB agar containing 100 µg/mL spectinomycin and incubated at 37 °C for 18 h. From each mouse's agar plate, we randomly selected four different colonies. The DNA barcodes from these colonies were then amplified using PCR and sent for Sanger sequencing to confirm their identity and to determine whether the barcodes belonged to the C1 clonal cluster (Supplementary Data 1). Except for gf3, from which we obtained two different barcodes across both cohorts, all identified barcodes were included in the C1 cluster. In the case of gf2, we isolated the C2 cluster.

Next, the clonal bacterial cultures were grown from glycerol stocks for 12 h in LB medium supplemented with spectinomycin (100 µg/mL) at 37 °C with constant shaking. These cultures were then washed and diluted into M9 minimal medium supplemented with D-glucose (0.2%, wt/wt). Each well was filled with a final volume of 200 µL, resulting in an inoculum concentration of ~10^5 cells. The plates were grown in a BioScreenC MBR and incubated at 37 °C with continuous shaking, while the OD600 was measured every 15 minutes for a duration of 25 h. All BioScreen growth experiments were conducted in biological triplicates.

**Whole genome sequencing.** To identify mutations segregating in the *E. coli* C1 clonal cluster during colonization of gf and rm cohorts, we conducted whole-genome sequencing (WGS) analysis. Specifically, we performed WGS on screened colonies from fecal samples collected at the endpoint of our experiments. Additionally, we conducted metagenomics analysis on the initial library that was introduced to mice. The screened colonies were grown overnight at 37 °C with spectinomycin. Subsequently, we isolated bacterial genomic DNA from bacterial suspensions using Takara Microbial DNA extraction kit (Cat = 740235). The DNA amount was quantified using a Nanodrop spectrophotometer. Whole-genome DNA libraries were prepared using the Kapa Hyperprep DNA library kit from Roche. Sequencing was performed on a NextSeq500 MidOutput 150-cycles flowcell, yielding ~10 million paired-end reads per sample (PE75). Sequencing were carried out at the Université de Montréal's IRIC Genomic platform. Mutations were identified using the BRESEQ pipeline version 0.23[114], with the polymorphism option enabled. For the analysis of samples from single colonies, we considered mutations with a frequency of 100%. In contrast, for the analysis of the initial library, we focused on consensus mutations between two technical replicates (two separate WGS preparations of the gavage sample).

## Analysis

### Barcode extraction from the FASTQ file and determining putative "true" lineages.

To understand how clonal populations of cells change over time, we first identified and extracted the barcode sequences from our raw sequencing data. To extract barcode sequences from raw sequencing data, we first applied a quality filter, discarding any reads with an average Phred score below 30 to minimize sequencing errors. Barcode sequences were identified and extracted using BarcodeCounter2[39], a Python pipeline that utilizes BLASTn+ v2.6.0[115,116] to detect barcodes based on a predefined sequence template. The pipeline relied on two short, constant flanking sequences located on either side of the barcode within each read. These sequences served as reference points for barcode identification and reads lacking these flanking sequences were excluded to prevent misidentification. BLASTn+ was run with the following parameters: *-word_size 8 -outfmt 6 -evalue 1E0 -maxhsps 1*. The word size was set to 8 to ensure short sequences could be accurately identified. The e-value threshold was set to 1E0, allowing more flexibility in detecting short barcode sequences. The output format was set to 6, and the maximum number of high-scoring segment pairs (maxhsps) was limited to 1 to simplify output processing and reduce redundant matches.

Extracted barcode sequences can vary in length due to sequencing artifacts, synthesis errors, and library preparation inconsistencies. While the expected barcode length was 15 nucleotides (15 N), insertions or deletions can lead to slight deviations. To account for this, only barcodes between 13 and 17 nucleotides were retained, and sequences outside this range were excluded. To correct for sequencing errors, including insertions, deletions, and substitutions, we applied the Deletion-Correct method, a modified version of the approach from Johnson et al. (2019)[117]. This method improved barcode accuracy by clustering similar sequences into deletion neighborhoods, where barcodes within three edits of each other were grouped together. Within each neighborhood, the most abundant sequence was selected as the true barcode, while less frequent variants were corrected to match it. This ensured that minor sequencing errors did not artificially increase barcode diversity. The parameters used for Deletion-Correct were the following: *min_counts_for_centroid = 2, max_edits = 3, poisson_error_rate = 0.1*. With these settings, barcodes with at least two reads were considered for correction, only sequences within three edits of a more common barcode were corrected, and a Poisson error rate of 0.1 was used to model sequencing error rates.

### Visualizing barcode dynamics.

To compare barcode trajectories within and between mice cohorts, we aimed to use consistent color coding for barcode lineages. First, we assigned a unique color to all lineages that reached a relative frequency of 5e-05 in their respective mouse. The frequency $f_i(k)$ of barcode lineage $k$ in condition $i$ is:

$$f_i(k) = \frac{x_i(k)}{\sum_j x_i(j)} \qquad (3)$$

where $x_i(k)$ is the barcode read count. This operation was applied to each mouse, such that the color scheme was consistent when the dynamics were compared (Fig. 2a, d and Supplementary Fig. 3). For example, a barcode lineage that was assigned the color "magenta [#c20078]" will always have this color in all the figures. Conversely, no other barcode was assigned the same color. To create the Muller-type plots for each mouse (Supplementary Fig. 3a, b and d), the barcode frequencies at every time point were represented in linear scale. In each mouse, the barcodes were sorted by the maximum frequency they attained over the time-series. This produced a stacked area plot where dominant barcodes were shown starting from the bottom of the panel and progressively lower-frequency barcodes were shown at the top. The same data was used to plot the frequency trajectories in log10-

transformation (Fig. 2a & d and Supplementary Fig. 3c). Barcodes that reached a minimum frequency of 1e-05 throughout its time-series were shown in color, whereas the remaining barcodes were shown in gray for clarity.

### Quantification of barcode diversity.

The simplest way to quantify the diversity of barcoded lineages in a population is to count the number of unique barcodes observed at a particular time point (Fig. 2a, d and Supplementary Fig. 5c). However, if lineages differ widely in frequency, then this measure may not be very informative and will suffer from substantial sampling bias (since very low-frequency barcodes will be under-sampled). A more general approach is to quantify the diversity of barcodes using the effective diversity index[118]

$$^qD = \left( \sum_k f_k^q \right)^{1/(1-q)} \qquad (4)$$

where $f_k$ is the frequency of the $k$th barcode lineage, and $q$ is the "order" of the diversity index.

When $q = 0$, the index simply counts the absolute diversity in the sample, i.e., the total number of unique barcode lineage. This measure is equivalent to the species richness used in ecological studies[119]. When $q = 1$, the index weights each barcode lineage by its frequency. This measure is equivalent to the exponential of the Shannon entropy $H = -\sum_k f_k \log(f_k)$. When $q \to \infty$, the index is equal to the reciprocal of the proportional abundance of the most common barcode lineages. Thus, only the higher-frequency lineages contribute to the value of this index. By comparing the diversity index across these three orders for $q$, we could describe the complex dynamics of the barcode composition over the course of the experiment. In the trivial case when all barcode frequencies were equal, the effective diversity index would be equal to the absolute number of barcodes regardless of the order of $q$. We should expect absolute diversity ($q = 0$) to be no greater than the maximum theoretical diversity of the barcode library. Additionally, we should also expect this measure to decrease over time as barcodes are lost from the population since diversity is exhausted (no new barcodes are generated).

### Barcode lineage clustering.

To identify the clonal lineages, we clustered the barcode lineages for each mouse based on the similarity of their time-series behavior. To maximize the accuracy of this clustering, we excluded barcodes with insufficient time points. Specifically, for each mouse, we retained only the lineages that i) exhibited non-zero frequency over at least 12 out of 18 time points for the rm cohort and ii) the mean frequency over the entire time-series is ≥5e-5. Similarly, for the gf cohort which had 1 time-point less, we retained barcodes with i) non-zero frequency for at least 11 out of 17 time-points and ii) the mean frequency over the entire time-series is ≥5e-5. This ensured that all barcode lineages included in the clustering had a sufficient number of points for pairwise comparison. This procedure meant that the lineage clustering focused on dominant and persistent clones; barcodes that immediately went to extinction were excluded. Altogether, this procedure was performed on a subset of ~300 to ~1300 lineages for each mouse, representing ~5% to ~10% of total unique barcodes. These dominant and persistent lineages represent ~7% to ~50% of the total number of *E. coli* cells at the end of the colonization experiment. The distance $\Delta F_{ij}$ between two frequency trajectories $f_i$ an $f_j$ was calculated as

$$\Delta F_{ij} = 1 - \rho(\log f_i, \log f_j) \qquad (5)$$

where $\rho(\log f_i, \log f_j)$ is the Pearson correlation coefficient between the trajectories. A distance close to 0 indicated a strong positive

correlation between the lineages, whereas a distance close to 2 indicated a strong negative correlation. From the resulting pairwise distance matrix, we applied hierarchical clustering using the "linkage" method from the *scipy.cluster.hierarchy* module in SciPy. We used the "average" agglomerative clustering method, which implements the algorithm unweighted pair group method with arithmetic mean (UPGMA)[120]. This method computes the distance between two clusters as the arithmetic mean of the distances between all lineages in both clusters. Then, for each cluster, we fitted a consensus trajectory using the local regression (LOESS). LOESS is a form of moving average where a line is fit locally using neighboring points weighted by their distance from the current point. These moving averages were referred in the text as "clonal clusters".

To determine the optimal clustering threshold, we note two general trends (Supplementary Fig. 6a–c). First, the LOESS of clusters with very few lineages will be sensitive to sequencing error. Thus, we include only clusters with at least 8 barcodes for the rm and gf cohorts. Second, when the threshold is too small, there are many clusters, but multiple clusters are similar to each other. This is manifested by the value of the smallest distance between the LOESS average of any cluster pair (black dots). Third, when the threshold is too large, there are very few clusters where barcodes with distinct dynamics are grouped together. In clustering, the practice was to find the cross-over between the smallest distance between cluster centroids (LOESS) and the number of clusters. This cut-off was indicated as the red curve in Supplementary Fig. 6b and c). Based on these cut-offs, 10 clusters for the rm cohort, and 6 or 7 for gf (Supplementary Fig. 6d and e).

**Quantification of community dynamics by 16S profiling.** The paired-end MiSeq Illumina reads resulting from sequencing of the 16S rRNA V4 region were processed using the *dada2* v1.22 pipeline[121]. Primer sequences were removed using *cutadapt* v2.8[122] before amplicon sequence variant (ASV) inference. Forward and reverse read pairs were trimmed to a run-specific length defined by a minimum quality score (Phred score ≥25) using the *filterAndTrim* function of the *dada2* R package[121]. Error rates were estimated from sequence composition and quality by applying a core denoising algorithm for each sequencing run. Then pairs were merged if they overlapped using the *mergePairs* function. Bimeras, which were chimeric sequences, were removed with the *removeBimeraDenovo*. Taxonomy was assigned using the *assignTaxonomy* function that maps reads onto the *SILVA* (v. 138) reference database[123]. We excluded sequences that matched mitochondrial or chloroplast DNAs. In each mouse, the relative abundance of a taxonomic unit $i$ at time $t$ is given by:

$$a_i(t) = \frac{r_i(t)}{\sum_j r_j(t)} \tag{6}$$

where $r(t)$ is the absolute abundance (number of reads) for the unit. Similar to the barcode dynamics, we calculated the community's effective diversity index but at the level of the family (see ***Quantification of barcode diversity***). For further analyses, families with frequency lower than 1e-03 were grouped as "Other", while the rest of the groups were clustered under their bacterial family classification.

**Co-clustering of *E. coli* clonal lineages and community dynamics from 16S.** To detect the potential interactions between the bacterial community and *E. coli* clonal clusters, as might be manifested in the correlation between their time-series, we recognized that the interactions could introduce local and transient stretching or lags. Thus, a straightforward Pearson correlation is ill-suited to detect such

interactions. Therefore, we calculated the pairwise distances using the shape-based metric (SBD)[72]. Briefly, the SBD is an iterative algorithm that detects the shape similarity of two time series, regardless of amplitude or phase differences (Supplementary Data Fig. S4). For the community dynamics, we used the log-transformed relative abundances of taxa at the family level with a minimum of 7 non-zero time points. For the clonal dynamics, we used LOESS smoothing arising from the clustering of *E. coli* barcodes. We z-normalized the time-series vectors to remove the amplitude effect and then calculated the shape-based distance (SBD)[72] implemented in the *tsclust* package[124] to calculate our distance matrix. Lastly, tree linkage was performed using the "average" (UPGMA) method to generate dendrograms (Fig. 4b and Supplementary Data Fig. S4).

**Replicability of clonal lineages in different mice from the same cohort.** To determine the replicability of clonal lineage dynamics across different mice, we applied hierarchical clustering using distance matrices derived from pairwise Pearson correlation followed by UPGMA linkage (Fig. 5a, d). The input to these analyses was the LOESS of the clonal lineages from each mouse (section iv. Barcode lineage clustering).

**Quantification of barcode similarity between mice from the same cohort.** To determine if the similarity in clonal lineage dynamics in different mice is driven by the same barcodes, we evaluated the overlap index in raw barcode identity for each cluster. In general, the overlap coefficient quantifies the Simpson similarity between two sets $A$ and $B$ that are not necessarily of the same size:

$$OC(A, B) = \frac{|A \cap B|}{\min(|A|, |B|)} \tag{7}$$

A value close to 1 indicates a high number of common elements, whereas a value near 0 indicates little overlap. We calculated the overlap index for all pairs of clonal lineage clusters in mice from the same cohort (see Fig. 5b, e). To determine that the overlap index did not arise by chance, we generated different compositions of sets $A$ and $B$ drawn randomly from our total pool of barcodes. For each composition, we calculated the overlap index (Eq. 7). This was performed 1000 times to arrive at a distribution of $OC(A, B)$ values. The significance of the observed overlap index $x$ between the real clusters $A$ and $B$ was expressed as a z-score on the simulated distribution of overlap indices:

$$Z = \frac{x - \mu}{\sigma} \tag{8}$$

where $\mu$ is the mean and $\sigma$ the standard deviation of the sample distribution. Lastly, significant overlap coefficient values with $|Z| > 1.96$ or $P$ value 0.05 are shown in blue in Fig. 5c, g, and their size is scaled proportionally to their $P$ value.

**Reporting summary**
Further information on research design is available in the Nature Portfolio Reporting Summary linked to this article.

## Data availability
Raw barcode sequencing data from this study have been deposited in the National Center for Biotechnology Information Sequence Read Archive. This includes BioProject accession number PRJNA1113167 for all 16S data, PRJNA1113343 for high-resolution barcode data from germ-free, reduced microbiota, and innate microbiota cohorts across different time points, and PRJNA1113345 for whole genome sequencing results of dominant clonal clusters lineages. Source data are provided with this paper.

## Code availability

The code used for data analysis and figures is available at https://github.com/melisgncl/Intra-and-inter-species-interactions-drive-phases-of-invasion-in-gut-microbiota-.

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

## Acknowledgements

The authors thank Stephen Michnick and James Omichinski for discussions and the Jesse Shapiro lab for protocols on 16S rRNA profiling. We thank the mice facilities at Université Sherbrooke and the University of Toronto. This work was supported by the following grants: Canadian Institute for Health Research PG-408523 and PG-197783 (A.S.), Natural Sciences and Engineering Research Council of Canada RGPIN-2016-06566 (AS), Canada Research Chairs (A.S.), and National Research Foundation (South Africa) grant 89967 (C.H.). We also acknowledge a fellowship from Fonds de recherche du Québec – Nature et technologies (L.G.).

## Author contributions

A.S., M.G., and S.B. conceptualized this study. G.M.C., A.F., and S.R. performed the *E. coli* colonization experiments in mice with innate and antibiotic-perturbed microbiota. D.T. and D.P. performed the colonization experiments in germ-free mice. MG performed the genomic extraction, barcode amplification, 16S rRNA profiling with the help of Z.S. and C.M. M.G. implemented all the bioinformatic analysis and simulations with the help of L.G. and D.G.L. A.S., M.G., and C.H. developed the dynamic covariance mapping approach. A.S. and M.G. wrote the manuscript with feedback and editorial support from all authors.

## Competing interests

The authors declare no competing interests.

## Additional information

[1]Department of Biochemistry, Université de Montréal, 2900 Édouard-Montpetit, Montréal, Quebec, Canada. [2]Robert-Cedergren Center for Bioinformatics and Genomics, Université de Montréal, 2900 Édouard-Montpetit, Montréal, Quebec, Canada. [3]Département de microbiologie et d'infectiologie, Université de Sherbrooke, 12e avenue Nord, Sherbrooke, Québec, Canada. [4]Centre for Invasion Biology, Department of Mathematical Sciences, Stellenbosch University, Stellenbosch, South Africa. [5]National Institute for Theoretical and Computational Sciences, African Institute for Mathematical Sciences, Cape Town, South Africa. [6]Department of Immunology, University of Toronto, 1 King's College Circle, Toronto, Ontario, Canada. [7]Département d'immunologie et biologie cellulaire, Université de Sherbrooke, 12e avenue Nord, Sherbrooke, Québec, Canada. [8]Department of Life Sciences, Ben-Gurion University of the Negev, Be'er Sheva, Israel. ✉e-mail: adrian.serohijos@umontreal.ca

