## [Transparent Peer Review file · Nature Communications]

Quantifying the intra- and inter-species community interactions in microbiomes by dynamic covariance mapping

Corresponding Author: Professor Adrian Serohijos

Version 0:

Reviewer comments:

Reviewer #1

(Remarks to the Author)

In this work, Gencel et al. develop a mathematical approach to estimate the interaction matrix of microbial communities from abundance time-series data. They apply this new approach called Dynamic Covariance Mapping (DCM) to study *E. coli* colonization of the mouse gut. They use a chromosomal barcoding technique to track intra-species interactions with high-resolution in addition to 16S profiling and study this colonization under different levels of native microbiome complexity: germ-free, antibiotic-perturbed and innate-microbiota. Using DCM, they identify distinct temporal phases in the community dynamics following *E. coli* inoculation. The innate microbiota condition was resistant to *E. coli* colonization and according to the authors showed only one identifiable phase of community dynamics while the antibiotic-perturbed condition showed three distinct phases: instability, recovery of resident microbiota, and equilibrium. The germ-free mice similarly showed three phases of community dynamics with clonal sweeps in place of resident microbiota recovery. Building on the DCM analysis, the authors profit from the high-resolution of their chromosomal barcoding technique to identify specific interactions between resident microbiota species and *E. coli* barcode clusters: one between the dominant cluster and Lachnospiraceae, and other between low-frequency clusters and Lactobacillaceae, showing that subpopulations of *E. coli* can interact differently with other species in the community. Following isolation and whole genome sequencing of clones belonging to the dominant cluster, the authors identified mutations in many genes that had been previously connected to higher fitness in the mouse gut and others that could be associated with this cluster's interaction with Lachnospiraceae.

Current methods to quantify species interactions from time-series data assume that the interaction matrix is fixed over time, so DCM could potentially be a useful tool to detect time-dependent species interactions (e.g. species changing from a commensal to a competitive interaction). That said, it is unclear whether the temporal dynamics that the authors identify in the mice data results from changes in the matrix of species interactions itself or instead if they are just quantifying temporal variation resulting from a fixed interaction matrix. Even in the simple 3-species example that the authors present in the supplementary material (Extended Fig 1 and Supp Note B) the species interaction matrix is assumed to be fixed so it is unclear to what extent DCM is able to detect changes in the interaction matrix relative to existing approaches based on fitting gLV models to the data. Moreover, the authors claim that their method is particularly useful to account for intraspecific variation but the way this is incorporated into the model is by adding additional equations to track individual strains within a species. Again this is something that could be done with standard gLV models so unless the authors benchmark their results using existing approaches (especially in scenarios where the matrix of species interactions changes over time) it is hard to determine how useful their method is. Other major points below:

* It is unclear how the different phases are defined from the variation in the PCA components. The authors said that this is done by looking at changes in the direction of PC1/PC2 (page 11, line 29) but based on this criteria one could also identify several phases in the dynamics of mice in the intact microbiome cohort (where the authors claim that there are no phases - Extended Fig 7a-d). It is also unclear based on this definition why the authors claim there is a phase in rm1 starting on day 8 (Fig 3a) while on day 11 for rm4 (where the PC value also goes from increasing to stabilizing) the authors don't identify any additional phases (Fig 3g). Given that identifying different temporal phases from PC dynamics is a key part of what this method could offer, the authors should formulate a much more precise definition of what a phase is and how to identify this from the PC data.

* Supplementary Note B.1 and B.2: Why is the J_{ij} compared to A_{ij} in the example 3-species gLV model when the authors have shown in the methods section of the main paper that in that model they expect $J_{ij}=a_{ij}z_i^*$. Could they show the comparison between their estimated $J_{ij\tau}$ and the analytical J_{ij} ? Can they also clarify this point in Eq. 3 and 5 in the main text?

* Page 6, line 1-3: "In practice, the Jacobian J can be estimated by the covariance of a member i 's time derivative and species j 's abundance $z(t)$ over a time interval τ (Fig. 2b; Methods)". Can the authors provide a mathematical motivation for why this covariance should estimate the jacobian? Their methods sections DCM(i) and DCM(ii) don't mention this covariance and methods section "DCM applied to intra-species..." already assumes the relationship.

*Page 27, line 18: Why just extend the temporal window used to compute the interaction matrix instead of using a sliding window?

Minor Comments:

* Page 4, line 31: repeated phrase.

* Page 3, line 15-16: unclear - please rephrase.

* Page 10, line 20: how do the authors define optimal diversity?

* Page 14, line 6: "the first bacterial family to rebound in rm 2 and rm 3". Should it be rm 2 and rm 4?

* Page 23, line 23: "are not homogenous and could has an impact on the gut composition" to "are not homogenous and could have an impact on the gut composition"

* Page 26, line 5 and page 26, line 28: Eq S1 is not there, do you mean Eq 1?

* Page 26, line 7: The use of Φ here is different from the main text and may cause confusion since it is also incompatible with the use in Eq. S4. Consider using a different variable.

Consider explicitly detailing how the derivatives used for the Covariance estimation were numerically calculated from the abundance data.

* Extended Data Figure 1b: Consider changing the color of the lines since it is very hard to distinguish between the black and dark blue currently used.

(Remarks on code availability)

Reviewer #2

(Remarks to the Author)

My review is attached as a pdf.

(Remarks on code availability)

I have not tried running the code myself or reviewed it, but have taken a look at the posted data, which I comment on in my review. The code that is missing is the initial barcode extraction and error correction work (which may just be a list of command line commands).

Reviewer #3

(Remarks to the Author)

(Remarks on code availability)

Version 1:

Reviewer comments:

Reviewer #1

(Remarks to the Author)

We believe the clarifications and new benchmarks the authors presented make it more clear that the DCM approach can identify temporal shifts in community interactions, a powerful advantage over standard gLV models. The new descriptions and structure of the methods also make the model assumptions and derivation more understandable, facilitating its replication by members of the community.

We would suggest that the figures used to respond to our comments (Reviewer Figures 1 and 2) are included as extended data figures in the paper. In particular we find Reviewer Fig 1 important because this data shows that gLV models aren't great when there are shifts in the interaction matrix and would even consider adding the fits for the DCM estimates of A1 and A2 to this figure and presenting both panels in Fig 1 in the main text as a way to motivate the DCM approach. Also, note that the legend for panel f in the Extended Fig 1 is missing.

(Remarks on code availability)

Reviewer #2

(Remarks to the Author)
Review attached in pdf

(Remarks on code availability)

Reviewer #3

(Remarks to the Author)
I co-reviewed this manuscript with one of the reviewers who provided the listed reports. This is part of the Nature Communications initiative to facilitate training in peer review and to provide appropriate recognition for Early Career Researchers who co-review manuscripts.

(Remarks on code availability)

Reply to reviews

NCOMMS-24-28412-T: “Quantifying the intra- and inter-species community interaction in a microbiome by dynamic covariance mapping” by Gencel *et al.*

Black: Reviewer comments

Blue: Reply to reviews

Red: Quotes from the Main text and accompanying files.

REVIEWER 1

Comment 1: *Current methods to quantify species interactions from time-series data assume that the interaction matrix is fixed over time, so DCM could potentially be a useful tool to detect time-dependent species interactions (e.g. species changing from a commensal to a competitive interaction). That said, it is unclear whether the temporal dynamics that the authors identify in the mice data results from changes in the matrix of species interactions itself or instead if they are just quantifying temporal variation resulting from a fixed interaction matrix. Even in the simple 3-species example that the authors present in the supplementary material (Extended Fig 1 and Supp Note B) the species interaction matrix is assumed to be fixed so it is unclear to what extent DCM can detect changes in the interaction matrix relative to existing approaches based on fitting gLV [generalized Lotka-Volterra] models to the data. Moreover, the authors claim that their method is particularly useful to account for intraspecific variation but the way this is incorporated into the model is by adding additional equations to track individual strains within a species. Again, this is something that could be done with standard gLV models so unless the authors benchmark their results using existing approaches (especially in scenarios where the matrix of species interactions changes over time) it is hard to determine how useful their method is.*

Reply: The question of whether DCM detects actual changes in species interactions over time, rather than merely reflecting temporal abundance variations within a fixed matrix, is valid. Therefore, as suggested by the reviewer, we use additional simulations and published 16S rRNA data to demonstrate that DCM captures changes in the interaction matrix. We also benchmark DCM against fixed-interaction generalized Lotka-Volterra (gLV) models.

- a) **DCM and simulation of a community with a variable interaction matrix:** In the new Extended Data Fig. S1a, we present the composition of a five-species community initially governed by a random interaction matrix, A_1 , simulated using the gLV model. At time $t = 20$, the community undergoes a perturbation by changing the interaction matrix to A_2 . The pre- and post-perturbation interaction matrices are still correlated ($r = 0.6$) (Extended Data Fig. S1e), but sufficiently different to change the abundance time series of the community. We then apply DCM to the full abundance time series and without prior input of when or whether a perturbation occurred. As shown in

Extended Data Fig. S1b, DCM correlates with the interaction matrix A1 and then adjusts to eventually correlate with the interaction matrix A2 after the perturbation. This result demonstrates DCM's sensitivity to changes in the interaction matrix, even without prior knowledge of when they occur. The community matrix shifts from A1 to A2 is also detected as boundary between dynamical phases from eigenvalue analysis (Extended Data Fig. S1c,d).

Extended Data Figure 1. DCM captures the interaction strength matrix and defines dynamical domains in Lotka-Volterra systems.

a, An illustrative example of DCM analysis on a 5-species gLV system with an interaction strength matrix A1 that shifts to matrix A2 at time $t = 20$. The correlation between elements of matrices A1 and A2 is in panel f. The species abundance time series are the solution to gLV and when all species abundances are initialized to 15 at $t = 0$. **b**, Correlation between the Community matrix estimated by DCM's J_{ij} and A1 matrix in the interval $t = [0,20]$ and between J_{ij} and A2 matrix in the interval $t = [20,50]$. The time-dependent J_{ij} reflects the shift in the 5-species interaction of the Lotka-Volterra matrix. **c**, Principal components (PC1 and PC2) of the eigenvalue matrix over the time interval for the 5-species community shown in panel a. A change in the direction of either PC1 or PC2 is indicative of a dynamical shift in the community. This shift is detected using change-

point analysis applied to the PC1 and PC2 timeseries (main Text). Five distinct phases are identified based on the matrix eigenvalues and their dynamic interpretations (stable/unstable or oscillatory (panel d)). The stars denote confidence level of identifying a dynamical shift: *** is highly confident phase boundary and ** is intermediate confidence (Methods). One of the major dynamic temporal boundaries corresponds to the shift in ecological matrix A1 and A2 (between phases II and III). Other phase boundaries correspond to approach to equilibration of the community from an initial set of abundances (panel d). **d-e**, The eigenvalue colors correspond to the rank of their magnitude (panel e) (See also Supplementary movie 1). Phase I is the transient approach to equilibrium of the abundances initially set to 15, followed by their equilibrium as dictated by matrix A1 (phase II). Phase III is the sudden change in abundances, and destabilization (real primary eigenvalues), due to ecological shift from A1 to A2. Phase IV is the approach to new equilibrium abundances dictated by A2, and Phase V is the steady-state equilibrium defined by the new interaction matrix.

We benchmarked our results against standard gLV models with constant interaction matrices. Specifically, we used web-gLV (Kuntal et al., *Frontiers in Microbiology*, 2019¹) that estimates the community matrix from time-series data. Similar to DCM, we did not inform web-gLV that there is a community shift at $t=20$. Reviewer Figure 1 below shows that the estimated community matrix by web-gLV over the entire period does not correlate with either A1 nor with A2, as expected since the community shift is not in the framework of constant interaction gLV. Of course, gLV could be forced to model a shift in the community interaction, but this information will need to be known up front to partition the abundance time-series into segments where constant community matrix applies.

These results show DCM captures the shift dynamics within a constant gLV matrix (such as approach to equilibrium), as well as shifts in the community matrix. Importantly, the DCM does not need to be informed *a priori* of this community shift, as one would in a gLV constant matrix model.

Reviewer Figure 1: Comparison of interaction matrices from GLV simulation under perturbation. This figure presents a comparison of interaction matrices from a Generalized Lotka-Volterra (GLV) simulation under perturbation, assessing how well the predicted interaction matrix corresponds to both the pre-perturbation matrix (A1) and the

post-perturbation matrix (A2) when the perturbation time is unknown. In the first panel, the predicted interaction matrix is compared to A1, yielding a Pearson correlation coefficient of $r = 0.39$ (P-value=0.05). The second panel compares the predicted matrix to A2, where the correlation is slightly lower ($r = 0.34$, P-value = 0.1).

b) DCM and experimental data with variable interaction matrix due to pathogen invasion: To evaluate DCM's performance on published data, we analyzed time-series microbiome data from a study on the invasion of the pathogen *Clostridium difficile* (Bucci et al., Genome Biology 17: 121 (2016)). The dataset includes 28 days of microbial abundance from five mice before and after infection with *C. difficile*.

Extended Data Figure 2. Application of DCM to a community perturbed by pathogenic bacteria.

a, Bucci et al.¹ used gnotobiotic mice pre-colonized with the Gnotocomplex microflora for 28 days, infected the mice with *C. difficile* at day 28, and then monitored the microbiome for another 28 days. Abundance timeseries shows the microbiome's most abundant five species during the 56-day experiment. **b & c**, Principal components PC1 and PC2 of the eigenvalues of the Jacobian. Change-point analysis identifies four phases. Phase I reflects the entry and establishment of the Gnotocomplex microflora, while Phase II reflect the transient instability detected from the rise in *A. muciniphila* and *B. ovatus*. Phase III is the collapse in abundance upon entry of *C. difficile*. Phase IV is the return to stability accompanied by the increase of *A. muciniphila* and *B. ovatus* to baseline levels in Phase I. (See also Supplementary movie 2).

Altogether, the key advance of DCM lies in its ability to capture time-varying interactions that standard gLV models—with their static interaction matrices—cannot accurately represent. This feature is particularly crucial in ecological systems where environmental factors or species behaviors shift dynamically, such as during infection, invasion, or when species members themselves evolve. Although various methods, including constant matrix gLV, can be applied in cases where community matrices experienced a shift (i.e., infection), the timing of this shift need to be known upfront to partition the abundance timeseries into intervals where the assumption of constant community matrix applies.

Comment 2: *It is unclear how the different phases are defined from the variation in the PCA components. The authors said that this is done by looking at changes in the direction of PC1/PC2 (page 11, line 29) but based on this criteria one could also identify several phases in the dynamics of mice in the intact microbiome cohort (where the authors claim that there are no phases - Extended Fig 7a-d). It is also unclear based on this definition why the authors claim there is a phase in rm1 starting on day 8 (Fig 3a) while on day 11 for rm4 (where the PC value also goes from increasing to stabilizing) the authors don't identify any additional phases (Fig 3g). Given that identifying different temporal phases from PC dynamics is a key part of what this method could offer, the authors should formulate a much more precise definition of what a phase is and how to identify this from the PC data.*

Reply: We agree and now provide a precise definition of the phases based on well-established change-point analysis of timeseries data (mentioned in the text (page 8, line 27) and in greater detailed in the Methods (page 32)):

(iii) Principal component analyses and determining dynamical boundaries by change-point analysis

Dynamical analysis of eigenvalues typically focuses on the largest eigenvalues.^{2, 3} However, all eigenvalues can contribute to a system's behavior, resilience, and resistance;⁴⁻⁶ thus, we instead analyze all eigenvalues. Since there are n eigenvalues (corresponding to n community members) for each time interval, we sought a simpler representation of the community's dynamic behavior. To this end, we used kernel PCA to reduce the dimensionality of the eigenspace as a function of time. The input to this kernel PCA dimensionality reduction is a $(2N) \times (s)$, where the $2N$ columns correspond to the real and imaginary components of the i th eigenvalue λ_i , while the s rows is the number of τ time intervals. The first two principal components (PC1 and PC2) as a function of time intervals are then tracked to determine the dynamical shifts in the community (Fig. 1c).

The code and data are available on GitHub: <https://github.com/melisgncl/Intra--and-inter-species-interactions-drive-phases-of-invasion-in-gut-microbiota->.

To explicitly identify shifts in PC1 and PC2, we employed change-point analysis, a statistical technique to detect points when the distributional properties (mean and variance) of the timeseries changes.⁷⁻¹² Specifically, we used the package *geomcp R package*¹³ that implements change-point analysis on multiple timeseries. It also uses maximum likelihood estimation (MLE) to estimate the timeseries distribution parameters before and after a changepoint and determine the number and location of the changepoints. We use the following parameters: `geo_result <- geomcp(data, penalty = penalty_type, test.stat = "Empirical", nquantiles = nq)`, where *data* is the combined timeseries PC1 and PC2. The underlying distribution of the timeseries is derived empirically (`test.stat = "Empirical"`) based on a specified number of quantiles (*nq*).

We performed several independent changepoint analyses, varying the parameter *nq* and, importantly, using different types of MLE penalties (*penalty_type*) available in *geomcp*: Modified Bayesian Information Criterion (MBIC), Bayesian Information Criterion (BIC), Schwarz Information Criterion (SIC), and Hannan-Quinn Criterion. These penalties help reduce the likelihood of overfitting, where too many irrelevant changepoints are detected, and underfitting, where critical transitions may be missed.¹³ Changepoints that are consistently identified as significant by multiple MLE penalty criteria are denoted as (***), those identified in 90% of the independent changepoint analyses are denoted by (**), and those identified in 70% of the analyses are denoted by (*). We then inspected whether the identified changepoints made intuitive sense. Details of the changepoint analyses are available in the GitHub link above.

We clarify that we do not “claim that there are no phases” for the intact microbiota (im), because there are. But what we claim instead is that the phases are not reproducible across all mice, likely due to the technicality of not being able to sequence of the barcodes at sufficient resolution (to acquire intra-species tracking) (Extended Data Figure 4c).

Comment 3: Supplementary Note B.1 and B.2: Why is the J_{ij} compared to A_{ij} in the example 3-species gLV model when the authors have shown in the methods section of the main paper that in that model, they expect $J_{ij}=a_{ij}z_i^*$? Could they show the comparison between their estimated $J_{ij\tau}$ and the analytical J_{ij} ? Can they also clarify this point in Eq. 3 and 5 in the main text?

Reply: We compare J_{ij} with A_{ij} in the example of a three-species gLV model because it is a non-trivial result from the theory. We also wanted an illustrative example to demonstrate that the gLV matrix is captured by the Jacobian.

As suggested by the reviewer, we show below the comparison between $J_{ij\tau}$ and the analytical J_{ij} , which are strongly correlated:

Reviewer Figure 2: Comparison of J_{ij} to $\langle J_{ij} \rangle_\tau$ for three species gLV system in Supplementary Note B.1. The black line represents the Pearson correlation between J_{ij} and $\langle J_{ij} \rangle_\tau$ the values (left y-axis). The blue line indicates the p-value (right y-axis). (The peaks in the p-value correspond to roughly half a period of the 3 abundance sinusoids (Supplementary Note Fig. S3)).

We also rewrote the description of the DCM theory to clarify the relationship between estimated and theoretical Jacobian (page 5 and reply to Comment 4).

Comment 4: Page 6, line 1-3: “In practice, the Jacobian J can be estimated by the covariance of a member i ’s time derivative and species j ’s abundance $z(t)$ over a time interval τ (Fig. 2b; Methods)”. Can the authors provide a mathematical motivation for why this covariance should estimate the Jacobian? Their methods sections DCM(i) and DCM(ii) don’t mention this covariance and methods section “DCM applied to intra-species...” already assumes the relationship.

Reply: We rewrote and expanded the section “Dynamic Covariance Mapping (DCM): Theoretical Framework” in Results on pages 5-7. In particular, we provided the necessary details of using the covariance matrix to approximate the Jacobian. Specifically on page 5 and 6 of the main text, we first Taylor expanded the generic dynamical system in Eqn. (1):

Let the abundance vector and the population growth rate vector at time t_* be denoted by $z_* = [z_1(t_*), \dots, z_n(t_*)]^T$ and $f_* = [f_1(z_*), \dots, f_n(z_*)]^T$ respectively, the community dynamics can be approximated by Taylor expansion at time t_* as $\dot{z} \approx f_* + J_*(z - z_*)$, where J_* is the system Jacobian matrix evaluated at z_* , with its non-diagonal entry $J_{ij}(z_*) = \partial f_i / \partial z_j \Big|_{z=z_*} = z_i(t_*) \partial \phi_i / \partial z_j \Big|_{z=z_*}$ for $i \neq j$. The non-diagonal entry $J_{ij}(z_*)$ of the Jacobian matrix can be used to indicate the overall impact of the variation of member j 's abundance on the population growth rate of member i at point z_* , while the per-capita impact of j on i at point z_* is thus $J_{ij}(z_*) / z_i(t_*) = \partial \phi_i / \partial z_j \Big|_{z=z_*} \equiv a_{ij}(z_*)$. This allows us to approximate Eq.1 as the following:

$$\dot{z}_i / z_i = \phi_i \approx \phi_i(z_*) + \sum_j a_{ij}(z_*) (z_j(t) - z_j(t_*)). \quad (2)$$

Both J_{ij} and a_{ij} have been used to indicate the interaction strength of member j on i ,⁴⁹ and they are not necessarily time invariant. Only in some simple models of second-order differential equations (e.g. the generalized Lotka-Volterra model (gLV)), a_{ij} is assumed to be time-invariant parameters. The correspondence between Eq. 2 and gLV model with constant interaction matrix is shown in the Supporting Information section B.1.

Following this paragraph, we provide the mathematical motivation of using covariance for interaction strength estimation:

If the community abundance is recorded at time t_1, \dots, t_k within a τ -width time window $(t_* - \tau/2, t_* + \tau/2)$ centered at t_* , we can denote $z_j = [z_j(t_1), \dots, z_j(t_k)]^T$ the abundance time-series for species j and $\phi_i = [\phi_i(t_1), \dots, \phi_i(t_k)]^T$ the computed vector of per-capita population growth rate of species i . Note that the per-capita interaction strength in Eq. 2 $a_{ij}(z_*)$ can be viewed as the slope of a linear regression between ϕ_i and z_j , and can be estimated using ordinary least squares as the covariance between the two variables (Methods, DCM Part ii).^{51, 52} Thus, the covariance can be chosen as a metric for interaction strength, $a_{ij}(z_*) = \text{Cov}(\phi_i, z_j)$, which quantifies the direction of the linear relationship

between ϕ_i and z_j . Similarly, the overall interaction strength $J_{ij}(z_*)$ can be estimated as $J_{ij}(z_*) = \text{Cov}(f_i, z_j)$. The dynamic covariances of $\text{Cov}(\phi_i, z_j)$ and $\text{Cov}(f_i, z_j)$ provide us estimates of the per-capita and overall interaction strengths of member j on i near z_* .⁹ This DCM thus maps the interaction network of the microbiota as estimated matrices A_* and J_* , while the system stability is captured by their eigenvalues.

Further explanation is also provided in the Method section DCM part (ii) “Estimating the community interaction strength using covariance”.

Comment 5: Why just extend the temporal window used to compute the interaction matrix instead of using a sliding window?

Reply: Both sliding and expanding window approaches are widely used to analyze abundance-based time-series data, particularly in ecology, to detect transitions or tipping points^{14, 15}. Sliding windows calculate indicators over a fixed-size window that moves forward in time, making them effective for capturing short-term fluctuations but potentially sensitive to noise^{16, 17}. In contrast, expanding windows progressively increase the size of the dataset, enabling cumulative data analysis over time, which helps smooth out noise and provides more robust insights into long-term trends. Expanding windows often offer more reliable early warning signals in ecological systems, particularly when long-term stability and gradual transitions are of interest, especially in systems prone to noise^{15, 16}. Natural systems and experimental designs often carry noise that can be amplified by sliding windows without further manipulation. For this reason, we chose the expanding window approach, as it mitigates the impact of noise and offers a clearer understanding of systemic changes.

We also separated the theoretical motivation to DCM from its implementation to show that there are choices that need to be made (such as the expanding window) in the signal analysis due to particularities of the data. When the quality of the time-series improves in the future, finer time-sampling, which could reduce the noise in estimating changes in temporal dynamics.

In the manuscript, we added the following text (page 8):

Dynamic Covariance Mapping (DCM): Implementation

To implement DCM, we calculate the interaction strength $J_{ij}(z_*) = \text{Cov}(f_i, z_j)$ over an expanding time window τ (Fig. 1a, b). Alternatively, the Jacobian could be calculated over a sliding time window⁵³, but is more sensitive to noise in real time series data compared to an expanding time window^{53, 54}.

Rev 1-Comment 6: List of minor comments

* Page 4, line 31: repeated phrase.

Reply: Done

* Page 3, line 15-16: unclear - please rephrase.

Reply: Done

* Page 10, line 20: how do the authors define optimal diversity?

Reply: We changed the wording to “highest diversity” which is shown in Extended Data Fig. 5b. (Page 13 line 9-11)

* Page 14, line 6: “the first bacterial family to rebound in rm 2 and rm 3”. Should it be rm 2 and rm 4?

Reply: Done (Page 18 line 6)

* Page 23, line 23: “are not homogenous and could has an impact on the gut composition” to “are not homogenous and could have an impact on the gut composition”

Reply: Done (Page 29 line 18)

* Page 26, line 5 and page 26, line 28: Eq S1 is not there, do you mean Eq 1?

Reply: Done

* Page 26, line 7: The use of Φ here is different from the main text and may cause confusion since it is also incompatible with the use in Eq. S4. Consider using a different variable.

Reply: This section is cut and now placed in the main text, including updates on equation numbering.

* Consider explicitly detailing how the derivatives used for the Covariance estimation were numerically calculated from the abundance data.

Reply: We now provide a more detailed explanation of DCM, separated into Theory and Implementation.

* Extended Data Figure 1b: Consider changing the color of the lines since it is very hard to distinguish between the black and dark blue currently used.

Reply: Original Extended Data Figure 1 is now removed.

REVIEWER 2

Comment 1. *In my check of the data analysis pipeline, I found some issues with the barcode extraction that have an impact on downstream visualization and analysis. For most barcodes, my quick pass at barcode extraction / correction agrees with the provided data, but I was surprised to see that the most frequent barcodes in the processed RM1 3h data were not present in my data. Looking closer I could see that these barcodes do not look right in the reads, and generally represent some kind of off-target PCR product that should be discarded during analysis.*

Reply: We sincerely thank the reviewer for the valuable input and thorough inspection of the results. Indeed, some artifactual barcodes arose in our pipeline due to specific characteristics of the original tool we used, *Bartender*. Based on recent analysis of barcode extraction and counting approaches (Johnson et al., *J Mol Evol* 2023), we have updated our pipeline to use *BarcodeCounter2* and *Deletion-Correct*. This successfully removed the spurious barcodes that notably occurred only in the rm cohort, but not the gf.

The removal of these spurious barcodes did not substantially alter our results, but we have updated all figures and the text accordingly. The new barcode extraction approach is described in the Methods section (page 38):

(i) Barcode extraction from the FASTQ file and determining putative “true” lineages

To understand how clonal populations of cells change over time, we first identified and extracted the barcode sequences from our raw sequencing data. To extract barcode sequences from raw sequencing data, we first applied a quality filter, discarding any reads with an average Phred score below 30 to minimize sequencing errors. Barcode sequences were identified and extracted using *BarcodeCounter2*¹¹⁶, a Python pipeline that utilizes *BLASTn+* v2.6.0^{117, 118} to detect barcodes based on a predefined sequence template. The pipeline relied on two short, constant flanking sequences located on either side of the barcode within each read. These sequences served as reference points for barcode identification and reads lacking these flanking sequences were excluded to prevent misidentification. *BLASTn+* was run with the following parameters: `-word_size 8 -outfmt 6 -evalue 1E0 -maxhsp 1`. The word size was set to 8 to ensure short sequences could be accurately identified. The e-value threshold was set to 1E0, allowing more flexibility in

detecting short barcode sequences. The output format was set to 6, and the maximum number of high-scoring segment pairs (maxhsps) was limited to 1 to simplify output processing and reduce redundant matches.

Extracted barcode sequences can vary in length due to sequencing artifacts, synthesis errors, and library preparation inconsistencies. While the expected barcode length was 15 nucleotides (15N), insertions or deletions can lead to slight deviations. To account for this, only barcodes between 13 and 17 nucleotides were retained, and sequences outside this range were excluded. To correct for sequencing errors, including insertions, deletions, and substitutions, we applied the Deletion-Correct method, a modified version of the approach from Johnson et al. (2019)¹¹⁹. This method improved barcode accuracy by clustering similar sequences into deletion neighborhoods, where barcodes within three edits of each other were grouped together. Within each neighborhood, the most abundant sequence was selected as the true barcode, while less frequent variants were corrected to match it. This ensured that minor sequencing errors did not artificially increase barcode diversity. The parameters used for Deletion-Correct were the following: *min_counts_for_centroid=2*, *max_edits=3*, *poisson_error_rate=0.1*. With these settings, barcodes with at least two reads were considered for correction, only sequences within three edits of a more common barcode were corrected, and a Poisson error rate of 0.1 was used to model sequencing error rates.

Comment 2: *The code used to do the barcode extraction and error correction should be posted, or the commands to bartender should be included in the methods.*

Reply: In Github, we removed the old script and replaced it with the new, improved version. The updated pipeline addresses the concerns above, ensuring more accurate extraction and analysis.

Comment 3: *I'm not sure how much real biology is captured by the clustering of barcodes, which is best seen in Extended Data Fig. 5. The purpose of this is to coarse grain the intra-strain diversity in order to model the dynamics of lineages, and I assume the idea is that modeling each barcode as a sub lineage is just not computationally feasible or presents too much of a multiple hypothesis testing issue. I am curious how many of these clusters*

will disappear when the barcode extraction is more strict - for example clonal cluster 3 in RM1 is certainly an artifact, and I suspect some of the other clusters are also the result of artifacts, not any differences in lineage genotypes. After filtering out these artifacts, I would strongly encourage you to do this clustering across all of your rm and gf data, since it appears that the bulk of the variation you are capturing here predates barcoding. In this way you would end up with <10 clusters that consist of the same barcodes across mice (Figure 5 shows that this is already kind of happening). This should make comparisons with family level dynamics and comparisons between replicates simpler and will probably better capture the biological variation in the *E. coli* population.

Reply: The reviewer is correct that the clustering of barcodes is to simplify the description and modeling of intra-strain diversity. Following the stricter barcode extraction pipeline, we performed the clustering again across the gf and rm data as suggested (revised Extended Data Fig. 5). In general, we observed less barcode cluster compared to our original results (e.g., 4 clusters in rm1 instead of 10, 8 clusters for rm3 and rm4 instead of 10, etc.).

Extended Data Figure 6. Determining the number of dominant clonal lineages. (panels d-e only) d, Dominant clusters for the mice with reduced microbiota (rm). The colors correspond to Figure 2b. Colored lines correspond to unique chromosomal barcodes in the cluster. Black lines correspond to the LOESS average. The number of unique raw barcodes that belong to the cluster is indicated. The clonal lineage clusters (or simply “clonal clusters”) are ordered, starting from the left, based on their average

barcode frequency on the last day. **e**, Dominant clusters for the germ-free mice (gf). The colors correspond to Figure 2e.

Comment 4. *Group 1 mutations are present in the gavage at 100% frequency and are also found in all the isolated clones. These mutations must be present in the ancestor of the barcoding reaction and differ from the reference genome, but do not reflect genetic variation or any kind of adaptation in this population. They should be excluded from analysis and not discussed, or only discussed in description of the strain background. Other mutations are categorized into groups 2-4 based on whether they are shared between mice in different treatments or the same treatment. I'm not sure this is the most useful way to partition and present this mutation data. We can see in the data that some sets of mutations occur together and are shared across multiple barcodes, which likely means that these were haplotypes existing in the population before barcoding. For example, this appears to be the case for the deletions near the flh genes. Seeing the same barcodes rise to high frequency in replicate mice also suggests that most of the variation we are seeing is from mutations that occurred before gavage. To be sure of this, we need to be sure that there was no migration of fit barcodes between mice. Were these mice co-housed? The methods say "Individual mice were housed in different cages over the course of the colonization experiments," but I'm not completely sure what this means- did each mouse have its own cage? Were they switched into different cages during the experiment so they could have been coprophagous?*

Reply:

- a) As suggested, we removed in Fig. 6 the mutations (originally Group1) present in the gavage sample at 100% and found in all clones. The reviewer is correct that these mutations are part of the ancestor immediately prior to the barcoding experiment.
- b) We maintained the way we partitioned the mutations (Groups 2-4, now relabeled as Groups 1-3).
 - (i) "some sets of mutations occur together and are shared across multiple barcodes, which likely means that these were haplotypes existing in the population **before barcoding**." These mutations are indeed the one in the original Group 1, now removed.
 - (ii) "same barcodes rise to high frequency in replicate mice also suggests that most of the variation we are seeing is from mutations that occurred **before gavage**". There are indeed barcodes that repeat in different mice, but whether they have the same mutations depends on the cohort. We clarify this now by adding this text (Page 24, lines 5-14):

In the rm cohort, one barcode appeared in multiple mice (TCGTAACCTAAGGCTT in Supplementary Table 1 and rm2.C1.b1, rm3.C1.b1, and rm4.C1.b1 in Fig. 6) and exhibited the exact same ~16.5 kb deletion. This suggests that the deletion may have been present in this specific barcode prior to gavage. On the other hand, in the gf cohort, another barcode (ATACAACGTGGTAGC in Supplementary Table 1 and gf1.C1.b1 and gf4.C1.b1/b2

in Fig. 6) also appeared in multiple mice. However, only gf4.C1.b1/b4 exhibited a deletion at this locus (~6.2 kb), while gf1.C1.b1 did not, suggesting that the mutation may be de novo. Indeed, another barcode found exclusively in gf3 displayed a different form of deletion (~15.9 kb) at this locus.

c) We now clarify that each mouse was housed in its own cage throughout the course of the experiment (no switching), see also reply to Comment 5.

Comment 5: *The same mutations were present in both the rm and gf cohorts, but different haplotypes were selected in each case, which is a cool result. To be sure this is due to the community, it would be good to clearly rule out other options. The gf cohort experiment was performed at another university - was the diet identical between the two experiments? Are the mice genotypes different? It would also be nice in this part of the methods to detail how fecal samples are collected.*

Reply: We now added details in the Methods (Page 35 lines 1-4):

The genotype for both gf and rm cohort mice is C57BL6/N (Taconic Biosciences). Both mice cohorts were fed ad libitum with the Teklad 2018SX (Sterilizable) chow, an 18% protein diet that was sterilized by autoclave.

Additionally, for the sample collection (Page 34 lines 30-31):

Fecal samples are obtained directly from the mouse anus and placed into a pre-weighed, sterile 2 ml Eppendorf tube.

Comment 6: *Figures 2 and 4 represent the most interesting part of the paper - the basic dynamics and how the dynamics within the E. coli population might be related to the dynamics of other species within the community. In the rm and nc mice, we see a fairly repeatable process at the family level of families coming back to reasonable abundance throughout the experiment, but this varies a bit mouse to mouse. I am tempted to connect this variation to variation in the timing of the sweeps between replicates - for example there is a late bout of selection (barcodes rising in frequency) in rm4, along with later arrivals of Ruminococcaceae and Oscillospiraceae in the community dynamics. In Figure 4 shows the results of shape-based metric clustering of clonal clusters and families. The main result here, that the sweeping barcode cluster has high shape similarity to the quickly rebounding Lachnospiraceae, makes sense but is hard for me to interpret. I'm not sure that a shape-based metric is best to account for lags. Instead, I wonder if you would find tighter associations between barcodes or barcode clusters and families if you looked for correlations across all replicates in the log-fold changes at each time interval. You could even include a free time-lag parameter for each association. For example, it would be very convincing if the same lineages rise in frequency at the same time as Ruminococcaceae in each rm mouse, but that time is different between mice. The power here actually comes from the natural variation in the timing of the community recovery in*

each mouse. Of course, this still doesn't tell us what is causing these changes in frequency, but it tells us that dynamics within the population are mirroring dynamics between community members, which I think would be a very interesting observation. This is something that I think might be interesting, of course I would understand if you looked and it's not there. I do think the current association between C1 and *Lachnospiraceae* doesn't feel convincingly representative of a biological association to me, just because they both happen to be the first thing sweeping in the barcode and 16S data, respectively.

Reply: We thank the reviewer for the constructive feedback.

a. Alternative to shape-based metric clustering in Figure 4a to account for lags.

The reviewer questions whether a shape-based metric is the best approach to account for lags and suggests alternative methods for associating barcode clusters with families. To address this concern, we conducted additional analyses to explore the associations between barcode clusters and microbial families using correlation-based methods with time-lag parameters. Specifically, we employed **Lag-Penalized Weighted Correlation (LPWC)**; <https://gitter-lab.github.io/LPWC/articles/LPWC.html> and Chandereng & Gitter. BMC Bioinformatics 21, 21 (2020)²²), which groups pairs of time series that exhibit closely related behaviors over time, even if their timing is not perfectly synchronized (presence of lags). LPWC aligns time series profiles to identify common temporal patterns, penalizing the correlations of lagged profiles based on the length of the introduced temporal lags using a Gaussian kernel.

With LPWC, the barcode clusters exhibit strong associations with specific microbial families when a time lag is considered, much like with the shape-based approach. For instance, Barcode Cluster C1 still shows a strong correlation with *Lachnospiraceae*. Following the update on barcode clustering to remove the spurious barcodes (Reviewer 2, Comment 1), we now also observe that C1 also with *Enterococcus* (updated Figure 4a).

We compared the dendrograms generated from shape-based distance (SBD) clustering and LPWC (Supporting Information Figure S5). The dendrograms from both observed that they share similar structures, as quantified by cophenetic correlation (P-values are indicated in Figure S5). This suggests that, even with lag-free methodologies, the dendrograms are closely aligned and shape-based clustering can indeed capture lag-penalized temporal variations effectively. Altogether, by incorporating both shape-based and correlation-based methods with time lags, we provide a more robust representation of how barcode clusters and microbial families interact over time.

Due to space constraints, we add these results in the Supplementary Information section C.2.

Figure S5: Comparison of rm cohort co-clustering using shape-based distance (SBD) with lag-penalized weighted clustering (LPWC). **a-d**, First dendrogram created with shape-based distance; second dendrogram generated using LPWC. To assess the similarity between these clustering methods, we computed the cophenetic correlation coefficient, which quantifies how well each clustering method preserves pairwise dissimilarities between samples. The cophenetic correlation was greater than 0.7, for all samples indicating a significant correlation between the two clustering approaches. Colored lines, generated using the tanglegram function from the dendextend¹⁷ R package, connect common branches between the two dendrograms, visually highlighting their structural similarity.

As suggested by the reviewer, we also performed the analysis on log-fold change coupled with LPWC and compared it with original shape-based clustering analysis (**Reviewer-Figure 3**, below). The underlying intuition is that the log-fold change is proportional to the growth rate of species and clones per unit time. When we compared the correlation of abundances via SBD (**Reviewer-Figure 3**, left dendrogram) with correlation of log-fold change with LPWC, we observed significant differences in the resulting dendrograms. These differences suggest that the two approaches capture distinct aspects of the data.

We did observe “tighter” correlation between barcode cluster C1 and *Lachnospiraceae* in the log-fold change analysis for mouse rm1, however not so for the other rm mice. We surmise that a finer time sampling of the abundance time-series could be needed to detect correlations in the log-fold-change.

Reviewer-Figure 3: Comparison of co-clustering using shape-based distance and lag-penalized weighted clustering (LPWC) applied to log-fold change of the abundance time series. (a, b, c, d) Panels show dendrograms comparing two clustering methods: the first dendrogram was created using shape-based distance, while the second was generated using LPWC, specifically on log fold change data. The cophenetic correlation between the two methods also presented to assess clustering similarity.

b. “C1 and *Lachnospiraceae* ... association ... is just because they both happen to be the first thing sweeping in the barcode and 16S data, respectively.”

To determine the robustness of the C1 and *Lachnospiraceae* across the different mice and to determine if clone-species association occurs for other pairs, we performed co-clustering of all 16S and clone dynamics time-series across all mice in the rm (Reviewer-Figure 4). We found that the association between C1 and *Lachnospiraceae* is consistent across the rm mice cohort, suggesting a non-random association (Group 1 in Reviewer-Figure 4a, b, c, & d). We also observed high dynamic similarity between *Lachnobacillaea* and other low-frequency clones (Group 2 in Reviewer-Figure 4a, b, c, & d), showing that clone-species association extends beyond the C1-*Lachnospiraceae* pair. Moreover, similarity in dynamics within Group 1 and within Group 2 tend to be driven by similar barcodes (Figure 4c, f).

Reviewer-Figure 4: Dynamics and barcode content comparison within the rm cohort.

a,b. Dendrogram derived from co-clustering analysis within entire rm entire cohort, identifying two distinct groups based on correlation of abundance time-series of clones and families (panel b). Group 1 is predominantly a cluster of *Lachnospiraceae* and high-frequency clones (C1 and C2), while Group 2 is a cluster of *Lactobacillaceae* with low-frequency clones. Heatmap is the similarity in barcode and community dynamics, quantified by Shape-Based Distance (SBD). The dendrogram (panel a) is the zoom-in of the hierarchical clustering of the heatmap elements. **c,** Overlap index between barcodes

of clonal clusters. The position of matrix elements and the dendrogram is similar to panel b. **d**, Shape-based distances for clones/families within groups 1 and 2, and between and 2 (lower SBD means higher similarity in dynamics). Clones/families within each group have more similar dynamics than between groups (***) is P-value <0.001 using Wilcoxon test). **e**, Similar to panel d, but only among clones. **f**, Overlap index of clone-clone pairs in groups 1 and 2. Significant pairs with barcode overlap—identified through bootstrapping analysis (similar approach to primary Fig. 5e,f)—are colored in blue.

c. The reviewer suggests connecting the variation in dynamics between mice to the variation in the timing of sweeps between replicates, such as the late bout of selection in rm4 and the later arrivals of Ruminococcaceae and Oscillospiraceae.

Since *Ruminococcaceae* and *Oscillospiraceae* are late arrivals, which means that we do not have an accurate measure of their abundance dynamics in earlier timepoints. In our analysis, we only included species that were present for more than 7 time points, which means they had to be present for at least two-thirds of the experiment to be included in the clustering analysis.

Nonetheless, as suggested by the reviewer, we relaxed our criteria above and performed our co-clustering including *Ruminococcaceae* and *Oscillospiraceae* (**Reviewer Figure 5**). We observe that *Ruminococcaceae* and *Oscillospiraceae* tend to broadly cluster together and with high-frequency clones (rather than with *Lactobacillaceae* and lower frequency clones, group 2 in Reviewer Figure 4 above).

In the paper, we opted to keep our stricter criteria of including only families present for more than 7 time points.

Reviewer Figure 5: Co-clustering of Clones and Species Including Late-Rising Species. The figure illustrates the co-clustering of 16S rRNA data with late-rising species and clonal clusters across four cohorts (panels a-d: rm1, rm2, rm3, rm4). The results demonstrate that late-rising species generally tend to cluster with the C1 clone and *Lachnospiraceae*.

Comment 7-*The DCM approach serves as a descriptive tool, but in my opinion, the descriptive benefit doesn't outweigh the cost here. This is coming from more of an evolutionary biologist than an ecologist, but at the end of the day, I'm not sure what the DCM approach adds to our understanding of this dataset. That's not to say that the approach does not have merit itself. But the sentences that describe the dynamics in reference to the DCM phases or matrix eigenvalues are less clear than those that do not reference DCM. Two examples - on page 15: 'the eigenvalue decomposition analyses ... indicate a quasi-stable oscillator in Phase III,' but I don't see any convincing oscillatory behavior in the community or barcode data (the only thing that could be mistaken for oscillations is the normal noisy variation in low-frequency lineages). On page 12, about RM: 'The collapse of the resident community in Phase I is manifested in the bacterial community diversity (Extended Data Fig. 4c).' I do not see this collapse in Fig. 2c or in ED Fig. 4c, and in ED Fig 4c-d, it looks like the community diversity in RM early in the experiment is higher than in the NC population. Again, I think it is possible that applying some version of DCM to a larger swath of microbial time series data might be useful (e.g., for building predictive models as you mention in the final sentence), but for this dataset, I didn't feel that it added to my understanding."*

Reply:

a) Quantifying community stability is a pressing question in ecology, but the dominant tools (generalized Lotka-Volterra), assumes that community interaction is constant and not influenced by evolution. In many cases, this is not true due to polyclonality of species²³⁻²⁷, including in microbial communities in the gut^{28, 29}, as exemplified by the current paper's WGS data in Fig. 6. We argue that DCM is a way forward to relax this assumption, which is both a methodical and conceptual advance.

There is cost indeed to presenting novel tools/concepts, specifically those in between disciplines (ecology and evolution in this case), but that cost needs to be borne by somebody.

More specifically, we now illustrate the application of DCM and its added value over classical gLV by showing that DCM captures the variation in community matrix in (1) simulation and (2) real-life 16S data.

Simulation variable interaction matrix: In the new Extended Data Fig. S1a, we present the composition of a 5-species community initially governed by a random interaction matrix A1, simulated using the generalized Lotka-Volterra (gLV) model. At time t=20 (arbitrary units), the community undergoes a perturbation by varying specific elements in the interaction matrix (details provided). Importantly, the post-perturbation interaction matrix remains highly correlated with the initial matrix (>0.55). We then applied Dynamic Community Modeling (DCM) to the abundance time series to estimate the interaction matrix. Analysis with DCM reveals that while the correlation between A1 and the post-perturbation matrix A2 remains high initially, it shows a significant shift after the

perturbation, demonstrating DCM's sensitivity to changes in the interaction matrix, even without prior knowledge of when they occur.

Extended Data Figure 1. DCM captures the interaction strength matrix and defines dynamical domains in Lotka-Volterra systems.

a, An illustrative example of DCM analysis on a 5-species gLV system with an interaction strength matrix A_1 that shifts to matrix A_2 at time $t = 20$. The correlation between elements of matrices A_1 and A_2 is in panel f. The species abundance time series are the solution to gLV and initialized to 15 at $t = 0$. **b**, Correlation between the Community matrix estimated by DCM's J_{ij} and A_1 matrix in the interval $t = [0,20]$ and between J_{ij} and A_2 matrix in the interval $t = [0,20]$. The time-dependent J_{ij} reflects the shift in the 5-species interaction of the Lotka-Volterra matrix. **c**, Principal components (PC1 and PC2) of the eigenvalue matrix over the time interval for the 5-species community shown in panel a. A change in the direction of either PC1 or PC2 is indicative of a dynamical shift in the community. This shift is detected using change-point analysis applied to the PC1 and PC2 timeseries (main Text). Five distinct phases are identified based on the matrix eigenvalues and their dynamic interpretations (stable/unstable or oscillatory (panel d)). The stars denote confidence level of identifying a dynamical shift: *** is highly confident phase boundary and ** is intermediate confidence (Methods). One of the major dynamic

temporal boundaries corresponds to the shift in ecological matrix A1 and A2. Other phase boundaries correspond to approach to equilibration of the community from an initial set of abundances (panel d). **d-e**, The eigenvalue colors correspond to the rank of their magnitude (panel **e**). Phase I is the equilibration of the abundances, followed by the equilibrium dictated by matrix A1 (phase II). Phase III is the sudden change in abundances, and destabilization (real primary eigenvalues), due to ecological shift from A1 to A2. Phase IV is the approach to new equilibrium abundances dictated by A2, and Phase V is the steady-state equilibrium defined by the new interaction matrix.

16S data with variable interaction matrix due to infection: To evaluate DCM's performance on real-world data, we analyzed time-series microbiome data from a study on *Clostridium difficile* infection (Bucci et al., *Genome Biology* 17: 121 (2016)³⁰). The dataset includes 28 days of microbial abundance from five mice before and after infection with *C. difficile*. DCM captures the community shift due to infection of *C. difficile* (Extended Data Fig. 2).

Extended Data Figure 2. Application of DCM to a community perturbed by pathogenic bacteria.

a, Bucci et al.¹ used gnotobiotic mice pre-colonized with the Gnotocomplex microflora for 28 days, infected the mice with *C. difficile* at day 28, and then monitored the microbiome for another 28 days. Abundance timeseries shows the microbiome's most abundant five species during the 56-day experiment. **b & c**, Principal components PC1 and PC2 of the eigenvalues of the Jacobian. Change-point analysis identifies four phases. Phase I reflects the entry and establishment of the Gnotocomplex microflora, while Phase II reflect the transient instability detected from the rise in *A. muciniphilia* and *B. ovatus*. Phase III is the collapse in abundance upon entry of *C. difficile*. Phase IV is the return to stability

accompanied by the increase of *A. muciniphilia* and *B. ovatus* to baseline levels in Phase I. (See also Supplementary movie 2).

Altogether, DCM does not require a priori assumption of the constant community matrix. gLV can be used a time-series that has a community shift, but gLV would need this information as an input, but not in DCM. In real communities where communities are changing due to overlapping timescales of ecology and evolution, this information of when communities shift is not necessarily known.

b) Clarify sentences pertaining to DCM results.

Originally on page 15: “the eigenvalue decomposition analyses ... indicate a quasi-stable oscillator in Phase III”. **Reply:** The original sentence and the section on fitness estimates have been removed. In the new page 15, we add the following phrase after our description of Phase III oscillations (Page 15, lines 2):

The oscillatory behavior of the barcode dynamics is unlikely due to technical and experimental factors, such as PCR bias, efficiency in genome extraction, or number of *E. coli* genomes for PCR amplification (see Supplementary Notes section A).

On page 12, about RM: “The collapse of the resident community in Phase I is manifested in the bacterial community diversity (Extended Data Fig. 4c).” I do not see this collapse in Fig. 2c or in ED Fig. 4c, and in ED Fig 4c-d, it looks like the community diversity in RM early in the experiment is higher than in the NC population.

Reply: We appreciate the reviewer’s comment. To clarify, when referring to a “collapse,” we mean that both species richness and diversity in rm become notably lower compared to other phases. This collapse results primarily from antibiotic treatment combined with *E. coli* gavage. During the initial phase, *E. coli* dominates community dynamics, competitively displacing resident bacteria, thus reducing overall diversity. However, in later phases, resident community diversity does recover and increase again.

We also share the reviewer’s curiosity about why nc exhibits lower diversity compared to rm at early time points. One canonical ecological explanation is that certain resident species can rapidly become dominant by competitively excluding others, especially under perturbed conditions such as antibiotic treatments. For instance, previous studies have demonstrated that canonical gut residents, like *Lactobacillaceae* species, often become dominant following antibiotic exposure^{31, 32}.

Comment 8-On Page 27, line 18: “progressively increasing time intervals [τ] (3h-6h, 3h-12h, ..., and 3h-15 days)”. Are these the time intervals used for the DCM to create all the

figures etc.? The text implies that the DCM results represent these “phases” of dynamics over the course of the experiment, which I assume meant it was applied as a sliding window along the time course. But from this description it sounds like it is an “expanding window”, so that the late phase is actually data from the entire experiment. In general, I would expect a covariance matrix to change as the amount of data gets larger in a dynamic population, so now I’m just generally confused about the time scale of the model and these phases. Again, I find that DCM contributes more to confusion than understanding.

Reply: We clarify throughout the text that the time intervals is indeed an expanding window. Importantly, we now divide the description of DCM into Theory and Implementation. Specifically on page 8:

Dynamic Covariance Mapping (DCM): Implementation

To implement DCM, we calculate the interaction strength $J_{ij}(z_*) = \text{Cov}(f_i, z_j)$ over an expanding time window τ (Fig. 1a, b). Alternatively, the Jacobian could be calculated over a sliding time window⁵³, but is more sensitive to noise in real time series data compared to an expanding time window^{53,54}. In each time window, we calculate the n eigenvalues of the Jacobian (n is size of the abundance time-series vector) to determine if the community is stable/unstable and exhibits oscillatory behaviours (Fig. 1c, left panel)⁵⁵⁻⁵⁹. These analyses lead to a time-dependent estimate of the community interaction matrix and the stability of the community

Comment 9: According to the methods, before the gavage of the barcoded population, the rm and nc treatments were the same, but at day one the nc mice have a very different makeup. Page 14: “Sutterellaceae was unperturbed by the introduction of *E. coli* in all four mice (Fig. 2c). Muribaculaceae was unperturbed in rm 1 and rm 3, and it was the first bacterial family to rebound in rm 2 and rm 3 [I think this should say rm4].” When I looked at the 16S data in https://github.com/melisgncl/Intra--and-inter-species-interactions-drive-phases-of-invasion-in-gut-microbiota/tree/main/data/16S/taxa_long_format, I don’t see Muribaculaceae in the rm samples - maybe you mean to refer instead to Acholeplasmataceae here? More importantly, I don’t see Sutterellaceae or Acholeplasmataceae in the nc samples, which are instead dominated by Lactobacillaceae. Do you think there was something that caused one cohort to have such a different microbiota at the start of the experiment? Or do we think the Lactobacillaceae crashed out in the rm mice in 24 hours?

Reply: Thank you for noting the discrepancy regarding *Muribaculaceae* in the rm samples, which should have been labeled as *Acholeplasmataceae* instead. This has been revised in the text (Page 18 line 6).

Regarding the difference between the rm and nc mice microbiota at the start of the experiment, while the treatments were the same, we observed that the nc mice were dominated by *Lactobacillaceae*, which was not as prominent in the rm cohort. This variation could be attributed to natural differences in microbiota between cohorts, as such variability is common even under controlled conditions³³.

Additionally, *Lactobacillaceae* is a dominant family in the gut microbiota of mice, and its dominance in the nc cohort following antibiotic treatment is not unexpected^{31,32}. It is also possible that the introduction of *E. coli* affected the rm mice more rapidly, leading to a reduction in *Lactobacillaceae* within the first 24 hours.

Comment 10-*I couldn't find a description of how exactly fitness was calculated in the methods, but the results plotted in Extended Data Fig. 9 make me assume that mean fitness is not being accounted for here, which is why we are seeing so much "negative relative fitness." As the population is more dominated by the adaptive mutants, the population mean fitness increases so that neutral (ancestral genotype) lineages will decrease in frequency faster. Any measure of fitness here needs to take into account mean fitness (see the Levy and Blundell paper for an example of this). But, for the purposes of this paper, I'm not sure it is necessary to do an in-depth fitness analysis - as you say, the goal here is not to measure the DFE, so I would suggest cutting this section altogether, since it doesn't clearly contribute to the overall story.*

Reply: As suggested, we deleted the Extended Data Fig. 9 and we removed the section on fitness altogether.

Figure comments

1. General comment: The x-axis in all of these figures with time on the x-axis needs to either have 12h scaled as half a day, or have a split x-axis, with the first day shown and then a break in the line representing a change in time scale before the day-by-day samples are shown.

Reply: We appreciate the reviewer's suggestion regarding the x-axis scaling. However, maintaining the hourly resolution is essential for accurately capturing the fine-scale temporal dynamics within the first 24 hours. Many of the key effects occur on a short time scale, and adjusting the x-axis in the suggested manner may obscure important trends and patterns critical to our analysis.

2. Figure1: The graphics in panel d are nice! For 1e, I'm confused-the methods say the CFUs are on spectinomycin and should only be the barcoded population, but the *im* data makes it look like it is the whole gut population? Is the dashed line the resolution limit (this should be explained in the caption)?

Reply: The CFU is indeed just the barcoded population (plated on spectinomycin plate). The dashed line is the resolution limit, now mentioned in the caption.

The *im* data shows only the barcoded population colonizing the mice with innate (unperturbed) microbiota. The drop is due to unsuccessful colonization (depletion of barcoded cells in the fecal homogenates).

3. Figure2: I would go lower on the number of colored barcodes (maybe top250) to make the dynamics easier to see. There are only a handful of families dominating the dynamics here, only those should be shown in color, so the legend is useful (right now I have no idea what anything is and had to look at the data myself).

Reply: We tried this, but it's not dramatically better than the current figure. We maintained the number of colored barcodes to be consistent with our previous paper (Jasinska, et al. Nature Eco & Evo, 2020).

4. Figure3: As discussed above, for me this is more confusing than useful.

Reply: We now provide more explanation for the DCM's motivation and utility over and above standard ecological models. We also split the description of DCM into Theory and implementation.

5. Figure 4: I am hopeful that with new clusters the result here might become more clear. It would be great to see an example of similar trajectories between a species and barcode cluster across the 4 mice (if a good one exists).

Reply: The new clusters indeed show similar trajectories between C1 and *Lachnospiraceae* and *Enterococcaceae* (Region 1 in Reviewer Figure 4b, d, & w) and *Lactobacillaceae* with low-frequency clones (Region 2 in Reviewer Figure 4a, c, & d). This result is also in primary Fig. 5.

6. Figure 5: Clustering across mice would make this figure unneeded

Reply: We believe this figure (together with Reviewer Figure 4) is important to illustrate the similarity in barcode dynamics is accompanied by similarity in barcode composition.

7. Figure 6: Removing the mutations found at 100% in the gavage will simplify this figure a bit. I like the synteny layout of the grid.

Reply: Done.

Detailed comments:

1. Abstract, first sentence “a microbiome’s composition ... is dictated by the community interaction matrix” - this ignores any higher order effects etc. Maybe “is often described using the community interaction matrix, which is commonly assayed...”

Reply: We changed the word “dictated” with “influenced”. (Page 2 line 1)

2. Page 3, lines 3-4: “These characteristics are embedded in the community interaction matrix that quantifies” - same comment as above, I don’t think embedded is the right word, maybe “can often be explained using the community interaction matrix, which quantifies” (I think the “ , which” just reads much better in both cases).

Reply: Done. (Page 3 line 2)

3. A few examples of sentences that break the flow due to sentence structure/grammar issues. Page 3, lines 11-12: “[microbes] have members that are challenging to culture or practically isolate” - the subject of this clause is microbes, which doesn’t make sense.

Reply: Done. (Page 3 line 10)

Page 3, lines 15-16: “Non-parametric models of the community models of the community matrix does not incorporate evolutionary forces” - “does” should be “do” here, but more importantly, the first half of the sentence is just hard to parse. Maybe just “Most non-parametric community models do not incorporate...”?

Reply: Done. (Page 3 line 15-16)

Page 3, line 21: “the community matrix, is in principle, also influenced” - remove the commas.

Reply: Done.

Page 4, lines 4-5: “to understand its stability and dynamics” - maybe there is supposed to be an “and” before this text? Right now, it doesn’t make sense.

Reply: Done. (Page 4 line 4-5)

4. Page 3, line 29: I would say “A typical human gut microbiome” (I assume you’re referring to a human gut, or say mouse if that’s the case).

Reply: Done. (Page 3 line 29)

5. End of page 3, start of page 4: I’m not sure “clones” is the best word to be using here. Maybe “haplotypes” is more direct and clear?

Reply: We think clone is appropriate here as we refer to the genetic identity across genomes as a concept, although the technical measurement of these clones is via haplotype sequencing such as in Illumina.

6. Page 4, line 17: “E. coli population” -> “microbial populations” - most of this work hasn’t been in E. coli.

Reply: We do refer here to a specific work, Jasinska et al, Nat Eco Evo, which was done in E. coli.

7. Page 7, line 6: “enables [us] to”.

Reply: This section is rewritten entirely.

8. Page 10, lines 22-23: “suggesting a strong intraspecies selection pressure in the absence of resident bacteria” - to me, this is one of the coolest results in the paper!

Reply: We think so too!

9. Page 10, line 27: “Barcodes that appeared” - I think “Lineages” is a better term than “Barcodes,” since barcodes refer to the data, but lineages refer to the actual system.

Reply: Done. (Page 13 line 17)

10. Page 12, line 30, last word: “include[s]”

Reply: Done.

11. Page 16, lines 17-19: “the similarity in barcode dynamics and composition is understandably weaker in the rm and gf mice, where the effects of standing genetic variation and/or de novo mutations in rm are modulated by ecological interactions with [a changing] bacterial community” - I think adding that the bacterial community is dynamic is important and should make this more clear.

Reply: Done. (Page 20 line 14-16)

12. Page 20, line 15: cut “below” (redundant)

Reply: Done. (Page 27)

13. Page 20, line 18: “dynamic[s]”

Reply: Done. (Page 27 line 1 -2)

14. Page 20, lines 30-31: “we demonstrate that intra-species variation leads to time-dependent interactions.” Unfortunately, I don’t think this is clearly demonstrated - while the community interaction matrix is trying to quantify interactions, it may just be quantifying correlations between abundances and abundance derivatives that occur by chance or due to outside environmental factors. It’s also not clear to me that the intra-species variation in particular is leading to any time-dependent interactions.

Reply: We revised the sentence. (Page 27 line 12-15)

15. Page 21, line 9: “Other previous studies” - cut “Other” (redundant)

Reply: Done. (Page 27 line 25)

16. Page 21, line 9: “dynamics like gut invasion” -> “dynamics of gut invasion”

Reply: Done. (Page 27 line 25)

17. Page 21, line 20: “providing much higher DNA barcode density” -> “barcode diversity”

Reply: Done. (Page 28 line 5)

18. Page 22, lines 10-26: This paragraph feels like a response to a previous reviewer asking you to compare to plasmid systems. In my opinion, it doesn’t belong in the discussion and should be moved to the supplement, so it doesn’t interrupt the flow of the more important things you’re discussing here.

Reply: Done. We moved this paragraph to the supplementary discussion.

19. Page 23, lines 8-17: I like the first clause here about the goal of this study, but the rest of the paragraph again feels like a response to a previous reviewer and distracts from your main message.

Reply: Done.

20. Page 23, line 23: “could has” -> “could have”

Reply: Done. (Page 29 line 18)

21. Page 24, line 8: “underly the C1 Lachnospiraceae” -> “underly the similarity between C1 and Lachnospiraceae dynamics.” And above on line 4, “C1 Lachnospiraceae interaction” -> “C1-Lachnospiraceae association,” because this is not necessarily an interaction. I also suggest moving this paragraph to the results - having it in the discussion puts too much weight on it and breaks up the flow.

Reply: These sentences have been revised and moved this discussion to the results page.

22. Page 24, line 12: “pedantic” - maybe you meant “didactic”? I would just cut, “illustrative examples” is enough.

Reply: Done. (Page 29 line 25)

23. Page 24, lines 15-16: “is too simplistic and inadequate to” -> simplify to “cannot”

Reply: Done. (Page 29 line 28)

Thoughts for the future:

This is not a suggestion for this paper, but I wanted to include a hypothesis for what could be the most interesting phenomena here. In the *gf* mice, we repeatedly see a kind of secondary sweep around day seven in which lineages are getting up to ~1% frequency. We see similar family-level dynamics in the *rm* data. I wonder whether this is the same successional pattern - for example maybe these *gf* lineages are fulfilling a similar ecological role as one of the families rising to ~0.1-1% in *rm* mice. The C2 clones with *lrp* insertions are examples of these *gf* lineages and look like they have a different metabolic strategy based on ED Fig. 10. Again, not a suggestion to dig into for this paper, just an idea.

Reply: Excellent thoughts. We will do follow-up studies in the future.

1. Kuntal, B.K., Gadgil, C. & Mande, S.S. Web-gLV: a web based platform for Lotka-Volterra based modeling and simulation of microbial populations. *Frontiers in microbiology* **10**, 288 (2019).
2. Grziwotz, F. *et al.* Anticipating the occurrence and type of critical transitions. *Sci Adv* **9**, eabq4558 (2023).
3. Scheffer, M. *et al.* Anticipating critical transitions. *Science* **338**, 344-348 (2012).
4. Barabas, G. & Allesina, S. Predicting global community properties from uncertain estimates of interaction strengths. *J R Soc Interface* **12**, 20150218 (2015).
5. Chen, S., O'Dea, E.B., Drake, J.M. & Epaneanu, B.I. Eigenvalues of the covariance matrix as early warning signals for critical transitions in ecological systems. *Sci Rep* **9**, 2572 (2019).
6. Emary, C. & Malchow, A.K. Stability-instability transition in tripartite merged ecological networks. *J Math Biol* **85**, 20 (2022).
7. Page, E.S. Continuous inspection schemes. *Biometrika* **41**, 100-115 (1954).
8. Brodsky, E. & Darkhovsky, B.S. *Nonparametric methods in change point problems*, Vol. 243. (Springer Science & Business Media, 2013).
9. Kawahara, Y. & Sugiyama, M. Sequential change-point detection based on direct density-ratio estimation *Stat. Anal. Data Mining: ASA Data Sci. J* **5**, 114-127 (2012).
10. Talih, M. & Hengartner, N. Structural learning with time-varying components: tracking the cross-section of financial time series. *Journal of the Royal Statistical Society: Series B (Statistical Methodology)* **67**, 321-341 (2005).
11. Jia, S. & Shi, L. Efficient change-points detection for genomic sequences via cumulative segmented regression. *Bioinformatics* **38**, 311-317 (2022).
12. Bolton, R.J. & Hand, D.J. Unsupervised profiling methods for fraud detection. *Credit scoring and credit control VII*, 235-255 (2001).
13. Grundy, T., Killick, R. & Mihaylov, G. High-dimensional changepoint detection via a geometrically inspired mapping. *Statistics and Computing* **30**, 1155-1166 (2020).
14. Drake, J.M. & Griffen, B.D. Early warning signals of extinction in deteriorating environments. *Nature* **467**, 456-459 (2010).
15. Clements, C.F. & Ozgul, A. Including trait-based early warning signals helps predict population collapse. *Nature communications* **7**, 10984 (2016).
16. O'Brien, D.A. *et al.* EWSmethods: an R package to forecast tipping points at the community level using early warning signals, resilience measures, and machine learning models. *Ecography* **2023**, e06674 (2023).
17. Barter, E., Brechtel, A., Drossel, B. & Gross, T. A closed form for Jacobian reconstruction from time series and its application as an early warning signal in network dynamics. *Proceedings of the Royal Society A* **477**, 20200742 (2021).

18. Venkataram, S., Kuo, H.-Y., Hom, E.F.Y. & Kryazhimskiy, S. Mutualism-enhancing mutations dominate early adaptation in a two-species microbial community. *Nat Ecol Evol* **7**, 143-154 (2023).
19. Altschul, S.F., Gish, W., Miller, W., Myers, E.W. & Lipman, D.J. Basic local alignment search tool. *Journal of molecular biology* **215**, 403-410 (1990).
20. Camacho, C. *et al.* BLAST+: architecture and applications. *BMC bioinformatics* **10**, 1-9 (2009).
21. Johnson, M.S., Martsul, A., Kryazhimskiy, S. & Desai, M.M. Higher-fitness yeast genotypes are less robust to deleterious mutations. *Science* **366**, 490-493 (2019).
22. Chandereeng, T. & Gitter, A. Lag penalized weighted correlation for time series clustering. *BMC Bioinformatics* **21**, 21 (2020).
23. Ferreira, A., Crook, N., Gasparrini, A.J. & Dantas, G. in *Cell*, Vol. 172 1216-1227 (Elsevier Inc., 2018).
24. Carrara, F., Giometto, A., Seymour, M., Rinaldo, A. & Altermatt, F. Inferring species interactions in ecological communities: a comparison of methods at different levels of complexity. *Methods Ecol Evol* **6**, 895-906 (2015).
25. Hromada, S. *et al.* Negative interactions determine *Clostridioides difficile* growth in synthetic human gut communities. *Mol Syst Biol* **17**, e10355 (2021).
26. Wang, X.R., Peron, T., Dubbeldam, J.L.A., Kéefi, S. & Moreno, Y. Interspecific competition shapes the structural stability of mutualistic networks. *Chaos Solitons Fractals* **172** (2021).
27. Stump, S.M., Song, C.L., Saavedra, S., Levine, J.M. & Vasseur, D.A. Synthesizing the effects of individual-level variation on coexistence. *Ecol Monogr* **92** (2022).
28. Poyet, M. *et al.* A library of human gut bacterial isolates paired with longitudinal multiomics data enables mechanistic microbiome research. *Nat Med* **25**, 1442-1452 (2019).
29. Folkesson, A. *et al.* Adaptation of *Pseudomonas aeruginosa* to the cystic fibrosis airway: an evolutionary perspective. *Nat Rev Microbiol* **10**, 841-851 (2012).
30. Bucci, V. *et al.* in *Genome Biology*, Vol. 17 1-17 (Genome Biology, 2016).
31. Liu, P. *et al.* Antibiotic-Induced Dysbiosis of the Gut Microbiota Impairs Gene Expression in Gut-Liver Axis of Mice. *Genes (Basel)* **14** (2023).
32. Guo, J. *et al.* Characteristics of gut microbiota in representative mice strains: Implications for biological research. *Animal Models and Experimental Medicine* **5**, 337-349 (2022).
33. Hoy, Y.E. *et al.* Variation in taxonomic composition of the fecal microbiota in an inbred mouse strain across individuals and time. *PloS one* **10**, e0142825 (2015).

Reply to reviews

NCOMMS-24-28412-T: “Quantifying the intra- and inter-species community interaction in a microbiome by dynamic covariance mapping” by *Gencel et al.*

Black: Reviewer comments

Blue: Reply to reviews

REVIEWER 1:

Comment 1: We believe the clarifications and new benchmarks the authors presented make it more clear that the DCM approach can identify temporal shifts in community interactions, a powerful advantage over standard gLV models. The new descriptions and structure of the methods also make the model assumptions and derivation more understandable, facilitating its replication by members of the community.

We would suggest that the figures used to respond to our comments (Reviewer Figures 1 and 2) are included as extended data figures in the paper. In particular we find Reviewer Fig 1 important because this data shows that gLV models aren't great when there are shifts in the interaction matrix and would even consider adding the fits for the DCM estimates of A1 and A2 to this figure and presenting both panels in Fig 1 in the main text as a way to motivate the DCM approach.

Reply: We thank the reviewer for recognizing the powerful advantages of DCM over standard gLV models. As recommended, we have now included Reviewer Figure 1 as panels G-H in Extended Data Figure 1 (see below). This shows that the estimated community matrix by web-gLV over the entire period does not correlate with either A1 nor with A2, as expected since the community shift is not in the framework of constant interaction gLV.

However, acknowledging the second reviewer's concern that the manuscript is already dense, we decided to keep Reviewer Figure 2 in the rebuttal rather than include it in the main text or extended data. The reviews and reply to comments will be publicly available.

Extended Data Figure 1. DCM captures the interaction strength matrix and defines dynamical domains in Lotka-Volterra systems.

a, An illustrative example of DCM analysis on a 5-species gLV system with an interaction strength matrix A_1 that shifts to matrix A_2 at time $t = 20$. The correlation between elements of matrices A_1 and A_2 is in panel **f**. The species abundance time series are the solution to gLV and when all species abundances are initialized to 15 at $t = 0$. **b**, Correlation between the Community matrix estimated by DCM's J_{ij} and A_1 matrix in the interval $t = [0,20]$ and between J_{ij} and A_2 matrix in the interval $t = [0,20]$. The time-dependent J_{ij} reflects the shift in the 5-species interaction of the Lotka-Volterra matrix. **c**, Principal components (PC1 and PC2) of the eigenvalue matrix over the time interval for the 5-species community shown in panel **a**. A change in the direction of either PC1 or PC2 is indicative of a dynamical shift in the community. This shift is detected using change-point analysis applied to the PC1 and PC2 timeseries (main Text). Five distinct phases are identified based on the matrix eigenvalues and their dynamic interpretations (stable/unstable or oscillatory (panel **d**)). The stars denote confidence level of identifying a dynamical shift: *** is highly confident phase boundary and ** is intermediate confidence (Methods). One of the major dynamic temporal boundaries corresponds to the shift in ecological matrix A_1 and A_2 (between phases II and III). Other phase boundaries correspond to approach to equilibration of the community from an initial set of abundances (panel **d**). **d-e**, The eigenvalue colors correspond to the rank of their magnitude (panel **e**) (See also Supplementary Movie 1). Phase I is the transient

approach to equilibrium of the abundances initially set to 15, followed by their equilibrium as dictated by matrix A1 (phase II). Phase III is the sudden change in abundances, and destabilization (real primary eigenvalues), due to ecological shift from A1 to A2. Phase IV is the approach to new equilibrium abundances dictated by A2, and Phase V is the steady-state equilibrium defined by the new interaction matrix. **f**, Correlation between the pre- and post-perturbation interaction matrices A1 and A2, respectively. **g-h**, Comparison of interaction matrices estimated from the time series data in panel a using a gLV model with a constant interaction matrix (that is, agnostic to the community shift) (Supplementary Information section B2). In the first panel, the gLV predicted interaction matrix is compared to A1, yielding a Pearson correlation coefficient of $r = 0.39$ (P-value=0.054). The second panel compares the predicted matrix to A2, where the correlation is slightly lower ($r = 0.34$, P-value = 0.1)

Comment 2 : Also, note that the legend for panel f in the Extended Fig 1 is missing.

Reply: We have added the legend for Extended Figure 1.

REVIEWER 2:

Comment 1: *I was not clear enough in my Comment 3. When I recommended you cluster barcodes “across all of your rm and gf data”, I was suggesting that you do one clustering step on all the data, such that e.g. C1 is the same set of barcodes across experiments. **Since most of the variation in the E. coli population appears to be from pre-gavage, this should work, and it should drastically simplify your figures and analysis.** This is why I suggested you cut Figure 5 (panels a and d would still be relevant, but could be moved to a supplemental figure). I still think this is a good idea and would A) simplify your analysis, and B) better represent real variation in the E. coli population, making the analysis about associations between clusters and microbial taxa more powerful.*

Reply: We performed the analysis as suggested. In Reviewer Figure 1, we present results obtained by jointly clustering all the *gf* and *rm* samples, rather than clustering the barcodes per mice separately. Specifically, we used the same lineage definitions as before, pooled all clusters from both conditions, and applied our clustering pipeline. This joint analysis resulted in the identification of 28 clusters across all samples (Reviewer Figure 1a–b), compared to 5–12 clusters when *gf* and *rm* were clustered individually.

As a case study, we then examined how the original clonal clusters germ-free *gf1* (Reviewer Figure 1c) were reassigned into the 28 new clusters (Reviewer Figure 1d). In the original analysis, *gf1* contained 8 clusters; in the joint clustering, these were redistributed into 6 clusters. In the accompanying heatmap, every row shows what percentage of the original cluster is present in the new 28 clusters; increased red intensity indicates higher percentage values. As the reviewer suggested, the major clonal dynamics within *gf1* were largely preserved, with the dominant original cluster (C1) remaining prominent in joint Cluster 1.

However, some sample-specific clusters displayed different behaviors. For example, original cluster C8—an extinction cluster that started with high abundance and subsequently disappeared—is now grouped with original cluster C7 in joint Cluster 18. While both clusters exhibit extinction-like dynamics, they differ in their initial abundance levels and in the timing of their decline. In our opinion, the original C7 and C8 clusters should be considered separately, but now indistinguishable as Cluster 18 in the new analysis.

We performed a similar close look for *rm2* sample (Reviewer Figure 1e). Initially, *rm2* consisted of 12 clusters, which were redistributed into 21 clusters in the joint clustering. As with *gf1*, dominant original clusters were preserved; notably, original cluster C1 remained primarily within joint Cluster 1 (Reviewer Figure 1f). In contrast, low-frequency and mouse-specific clusters were more dispersed and separated more distinctly in the joint clustering.

Reviewer Figure 1: Joint clustering across all gf and rm samples. **a**, Clustering was performed jointly across all gf and rm samples. The clustering threshold was chosen based on our criterion balancing cluster size and the smallest inter-cluster distance. **b**, All identified clusters, along with their smoothed (moving average) frequency trajectories, are plotted and ranked according to their average frequency at the final time point. **c**, Original gf1 clustering results are shown for comparison. **d**, Heatmap showing the distribution of original gf1 clonal clusters mapped to the new joint clusters. The color intensity indicates the fraction of cells from each original cluster captured by the new joint clusters. **e**, Original rm2 clustering results are shown. **f**, Heatmap showing the distribution of original rm2 clonal clusters across the new joint clusters, with coverage represented similarly as in panel d.

Moreover, we examined all eight mouse samples and found that, similar to *gf1* and *rm2*, the C1 clusters in each sample were consistently mapped to either joint Cluster 1 or Cluster 2 (Reviewer Figure 2). This information is indeed consistent with the current Fig. 5a,d. Notably, the *rm* cohort, in contrast to *gf*, show more mouse-specific clusters, again consistent with Fig. 5a.

The joint clustering also did not reduce the total number of clusters (as opposed to the original intuition of the reviewer that “this way you would end up with <10 clusters that consist of the same barcodes”); instead, it resulted in greater separation. Notably, joint clustering introduced a substantial number of low-frequency clusters, including lineages that had been excluded during individual clustering such as singletons below the detection threshold of 10^{-3} .

For example, Cluster 3 is largely composed of such rare lineages. When examining this cluster across samples, the corresponding column is entirely grey, indicating that none of the individual mice had detectable representation of these lineages in their respective clustering. This is because, during individual clustering, we discard clusters that either have fewer than 8 members or lack any member that passes through at least detection threshold of 10^{-3} . This highlights that joint clustering increases sensitivity to low-abundance lineages by aggregating across samples.

Overall, this increase in the number of clusters is expected since there could be clusters unique only to *gf* or *im* cohorts or even only to some mice. While a major driver of the dynamics is standing genetic variation present in the gavaged barcoded population, there is also *de novo* genomic mutations that are unique in each cohort or mice (Fig. 6). Altogether, we believe that this analysis does not substantially add new information to what is already contained in the current figures. We have opted the reviews and replies to be public, thus, this analysis will be accessible, if needed.

Reviewer Figure 2: Clustering of merged gf and rm datasets. The distribution of all eight mouse samples was mapped onto the new joint clusters to assess how much of the original clonal clusters is captured by the new clustering. The analysis shows the degree of overlap between original sample-specific clonal clusters and the newly defined joint clusters.

Comment 2: On page 15 you write “Phase III and phase IV is quasi-dynamic equilibrium with both oscillations in the clonal and community dynamics (Fig. 3b, right; Fig.2b,c). The oscillatory behavior of the barcode dynamics that affects the community dynamics is unlikely due to technical and experimental factors, such as PCR bias, efficiency in genome extraction, or number of *E. coli* genomes for PCR amplification (see Supplementary Notes section A).” I think we may just disagree here about what constitutes “oscillations.” In my mind it implies a kind of repeated, predictable back-and-forth in frequencies, which I don’t see here. I prefer your term “quasi-dynamic equilibrium,” but I am ok with whatever you decide.

Reply: We thank the reviewer for the suggestion and have decided to retain the term “oscillations,”

Comment 3: I read Supplementary Notes section A, but I’m confused by the PCR bias analysis. The worst case scenario you model is $\epsilon=0.5$ for some barcodes, which results in a ~7000-fold frequency change, which is an extreme amount of PCR bias (for example we could observe a barcode that has a true frequency of 1 as being ~98.7% of the population)! I don’t quite understand how this doesn’t lead to a large change in the \log_{10} frequencies in Figure S1j, but maybe it’s because the biases are being applied to each barcode across all timepoints? In my experience the issues arise when PCR biases (or index-hopping etc.) occur at one timepoint and cause barcodes to jump up or drop down in frequency. I think it is misleading to suggest that this amount of PCR bias will affect results less than the amount of template DNA. I would actually be in favor of you cutting Supplementary note A1-3 altogether. It is clear that neither PCR bias or a low amount of template DNA is affecting your data at mid to high frequencies - the barcode trajectories look good! (with the exception of the low-frequency noise, which comes from a bunch of unknown sources and is present in practically every dataset of this kind that I’ve seen).

Reply: We thank the reviewer for the detailed comment. As noted, PCR jackpotting can happen any PCR amplification and other sources of amplification bias can indeed affect observed clonal frequencies, particularly at individual time points. In our analysis, we modeled PCR jackpotting, as described in Supplementary Note A. Importantly, the bias was not applied uniformly to all barcodes. Instead, in our second model, we randomly selected a small subset of barcodes (typically 10–25) to experience amplification bias, mimicking PCR jackpotting events.

Despite the severity of the simulated bias (e.g., $\epsilon = 0.5$), our analysis showed that such stochastic jackpotting events had a limited impact on the overall distribution of \log_{10} frequencies particularly when sufficient template DNA was used. This is consistent with previous studies, which have shown that stochasticity during PCR amplification contributes only a small amount of noise (Kinsler, G. *et al.* Journal of Molecular Evolution 91, 293-310 (2023)).

We believe this section is important to retain, as it supports the conclusion that the patterns observed in our data are not primarily driven by technical noise, but rather reflect underlying evolutionary dynamics. While we appreciate the reviewer’s suggestion to consider removing Supplementary Notes A1–A3, we feel this analysis strengthens the robustness of our conclusions. As noted, the clonal trajectories remain consistent and interpretable at mid to high frequencies, and the low-frequency noise observed is likely due to a combination of technical and biological factors, as is common in similar datasets.

We do appreciate the reviewer’s very favorable assessment of our data quality, but PCR jackpotting is a typical concern, and we believe the Supplementary Notes are needed.

3) *This is partly my fault, but the paper has become even more laden by response-to-reviewer-ballast, which I think further distracts from the cool story at the heart of all this. Some ideas:*

a) I understand I won't convince you to minimize the modeling aspect, but the Results section quickly gets so deep into the modeling weeds. Is there a way to move some parts of this to the methods and focus on the big picture more?

Reply: The current version, we feel, strikes the balance between the requirement for detail by Reviewer 1. The more mathematical details of the basis of DCM is already in the Methods section.

b) Related to 2a above - on Page 12, paragraph starting with "To ascertain..." could be shortened drastically to convey the point that the population crashed in im and so the data wasn't usable. I really don't think you need Supplementary note A, just saying there wasn't enough template DNA to get good data is sufficient.

Reply: We do appreciate the reviewer's very favorable assessment of our data quality, but PCR jackpotting is a typical concern, and we believe the Supplementary Notes are needed.

c) As mentioned above, I think you should cluster across all data and then lose figure 5.

Reply: As noted in our response to the first question and illustrated in Reviewer Figure 1, the joint clustering approach, while successfully capturing the dominant clonal structures, also introduces many singleton clusters when individual mouse samples are subset from the joint results. Moreover, rather than simplifying the clustering, it increases the total number of clusters. Nevertheless, we include this version for the sake of transparency and reproducibility, particularly as the data will be made publicly available. This allows others to further explore and interpret the joint clustering output.

4) Page 18: "These results demonstrated the impact of E. coli introduction on bacterial community composition." As discussed in your response to review, it seems likely that Lactobacillaceae was abundant in the nc samples due "to natural differences in microbiota between cohorts", so unfortunately, I don't think you can make this claim.

Reply: We toned down the sentence to reflect that, controlling for other variability as possible between the cohorts, the difference in the community between the two is suggestive of the *E. coli* colonization.

Figures:

1) General: I understand your decision to scale break mark (//) to the x-axis where the scale changes. I think just adding the mark would be sufficient to convey the scale change.

Reply: We have updated the figure 2 to include axis breaks, as suggested by Reviewer 1.

2) Figure 2: I understand your choice to keep the many colors for the barcodes. But for the families I really think just coloring the ones you mention in the text, or ones that reach above some frequency is a good idea. This will make it much easier for readers to connect the figure to the results. Right now it is hard to look through the long legend and find a family's trajectory data, and for most of those families you can't tell which line they are even if you try. Another option would be to make the key families you mention in the text have a thicker line weight and move them to the top of the legend.

Reply: We thank the reviewer for the helpful suggestion. In response, we have updated the figures 2 to emphasize the key families discussed in the text by plotting them with thicker lines and placing them at the top of the legend for improved visibility and interpretability.

Small comments:

1) Page 3 lines 3-4: “can be often explained” -> “can often be explained”

Reply: We changed into “can often be explained”.

2) Page 5 line 19: “other taxonomic unit (OTU).” Do you mean “operational taxonomic unit”? “other taxonomic unit” would also make sense, but then you shouldn’t use the OTU acronym, since that is well established for “operational taxonomic unit.”

Reply: We changed “other taxonomic unit” to “operational taxonomic unit”

3) Page 9 line 11: “We also applied DCM analysis to mice gut microbial community perturbed...” - maybe “a mouse gut microbial community”?

Reply: Corrected.

4) Page 9 line 13 “identifies” -> “identify”

Reply: Corrected.

5) Page 24 line 7 “This suggests that the deletion could have been present in this specific barcode prior to gavage...” - I think you can be stronger here, this is very strong evidence: “the deletion was present in this specific barcode prior to gavage...”

Reply: We keep the current phrasing since we could not ascertain this due to the resolution of metagenomics.

6) Page 24 line 9: “(genomes of a few hundred cells)” - this is picky but the metagenomics might have a depth of ~100X but that DNA likely comes from millions of cells. I would just cut this parenthetical.

Reply: We removed the “(genomes of a few hundred cells)”

7) Page 30 lines 20-22 - this sentence has a grammar issue and is difficult to read.

Reply: We divide the sentences to make more readable. “Our results also showed that these phases of invasion and the intra- and inter-species interactions are highly reproducible across mouse replicates. This is rather unexpected, considering the variability in microbiome compositions, which is the norm in the microbiome field”

Hi - my name is Milo Johnson and I am a postdoc currently working on understanding the evolution of a plant pathogenic bacterium, *Erwinia tracheiphila*, but I also spent part of my PhD doing yeast experimental evolution. I've been analyzing barcode frequency data throughout this work. I like to sign most of my reviews because I believe the review process should be interactive. I think the asynchronous back-and-forth of review is a vestige of pre-internet days and results in a weird power dynamic where a dialogue is impossible, so reviewers are somehow always right. Feel free to email me if you have questions or want to clarify one of my comments (milo.s.johnson.13@gmail.com)! My general philosophy on reviewing papers is:

- A) Papers should be published if they are not misleading
- B) Review should be:
 - a) Thorough (and consider data analysis and the written structure of the work)
 - b) Critical, but not absolute
 - c) Aimed at making the paper better (rather than aimed at justifying an assessment or decision)

Paper summary

In this paper, the authors tracked the dynamics of gut microbial communities in mice that were either treated with an antibiotic, raised germ-free, or neither, and then were inoculated with a population of barcoded *E. coli* cells. By tracking both the 16S composition of the microbiome and the frequencies of barcoded lineages, they could observe both the dynamics of adaptation and of ecological interactions over the course of two weeks. I think this is a very exciting and interesting way to try to get at the interplay between ecology and evolution in a semi-natural environment.

The negative control cohort was treated with antibiotics but not inoculated with the barcoded *E. coli* population, and in the innate microbiota treatment, the barcoded population did not establish in the gut environment, so the most interesting data involves the reduced microbiota (rm) cohort, which was treated with antibiotics prior to inoculation of the barcoded population, and the germ-free (gf) cohort. The dynamics of both the microbiota and the barcoded populations differ between the cohorts in somewhat reproducible / predictable ways. The differences in adaptive dynamics in the barcoded *E. coli* population between the rm and gf treatments demonstrates that the microbial environment plays a major role in shaping the selective environment.

The authors introduce dynamic covariance mapping (DCM) as a method to describe the dynamics of the microbial community. This method builds a community matrix from the covariance between the rate of change of abundance of each community member with the abundance of each community member, based on the assumption that the dynamics of the community can be primarily described by pairwise interactions between community members. They then calculate eigenvectors from this matrix, do PCA on the eigenvalues, and use these values to qualitatively explain phases of community dynamics in the population.

Major notes, moving from data to analysis and interpretation

1. In my check of the data analysis pipeline, I found some issues with the barcode extraction that have an impact on downstream visualization and analysis. For most barcodes, my quick pass at barcode extraction / correction agrees with the provided data, but I was surprised to see that the most frequent barcodes in the processed RM1 3h data were not present in my data. Looking closer I could see that these barcodes do not look right in the reads, and generally represent some kind of off-target PCR product that should be discarded during analysis.

(This is a rough plot, but the x axis here is timepoints in order and the y-axis is log frequency)

In the bottom row I've filtered out the barcodes that end in "CAGGTCTGAAGC" (the last 12 bases are the same in all these barcodes for some reason, they are shown in red in the original data at top). Note that right now, in RM1 these barcodes make up clonal cluster 3. I couldn't find the code / commands used for the Bartender extraction, so I'm not sure how these get extracted, but in my poking around it looked like these barcodes usually

appear ~16 bp into the read and are followed by “TGTCGCACAG”, whereas the normal barcodes appear at the start of the read and are followed by “TATCTCGGTA”. There are some other barcodes that my method does not extract that don’t have the “CAGGTCTGAAGC” sequence, for example “GTAGGTGGCAAGCGT” also appears 17 bp into reads. These reads blast to uncultured bacteria rRNA. Another barcode present in the processed RM1 data is “CCGCGGTCATACGAT”, and the reads with this sequence map to the mouse mitochondria. I don’t know for sure where these sequences are coming from - maybe from the 16S sequencing, or from off-target priming in bacterial genomes? In any case, stricter barcode extraction should get rid of these false barcodes and give a much clearer picture of intra-species dynamics (for example it should get completely rid of the weird 12h spike in rm4).

2. The code used to do the barcode extraction and error correction should be posted, or the commands to bartender should be included in the methods.
3. I’m not sure how much real biology is captured by the clustering of barcodes, which is best seen in Extended Data Fig. 5. The purpose of this is to coarse grain the intra-strain diversity in order to model the dynamics of lineages, and I assume the idea is that modeling each barcode as a sub lineage is just not computationally feasible or presents too much of a multiple hypothesis testing issue. I am curious how many of these clusters will disappear when the barcode extraction is more strict - for example clonal cluster 3 in RM1 is certainly an artifact, and I suspect some of the other clusters are also the result of artifacts, not any differences in lineage genotypes. After filtering out these artifacts, I would strongly encourage you to do this clustering across all of your rm and gf data, since it appears that the bulk of the variation you are capturing here predates barcoding. In this way you would end up with <10 clusters that consist of the same barcodes across mice (Figure 5 shows that this is already kind of happening). This should make comparisons with family level dynamics and comparisons between replicates simpler, and will probably better capture the biological variation in the *E. coli* population.
4. Group 1 mutations are present in the gavage at 100% frequency and are also found in all the isolated clones. These mutations must be present in the ancestor of the barcoding reaction and differ from the reference genome, but do not reflect genetic variation or any kind of adaptation in this population. They should be excluded from analysis and not discussed, or only discussed in

a description of the strain background. Other mutations are categorized into groups 2-4 based on whether they are shared between mice in different treatments or the same treatment. I'm not sure this is the most useful way to partition and present this mutation data. We can see in the data that some sets of mutations occur together and are shared across multiple barcodes, which likely means that these were haplotypes existing in the population before barcoding. For example this appears to be the case for the deletions near the *flh* genes. Seeing the same barcodes rise to high frequency in replicate mice also suggests that most of the variation we are seeing is from mutations that occurred before gavage. To be sure of this, we need to be sure that there was no migration of fit barcodes between mice. Were these mice co-housed? The methods say "Individual mice were housed in different cages over the course of the colonization experiments," but I'm not completely sure what this means - did each mouse have its own cage? Were they switched into different cages during the experiment so they could have been coprophagous?

5. The same mutations were present in both the rm and gf cohorts, but different haplotypes were selected in each case, which is a cool result. To be sure this is due to the community, it would be good to clearly rule out other options. The gf cohort experiment was performed at another university - was the diet identical between the two experiments? Are the mice genotypes different? It would also be nice in this part of the methods to detail how fecal samples are collected.
6. I think Figures 2 and 4 represent the most interesting part of the paper - the basic dynamics and how the dynamics within the *E. coli* population might be related to the dynamics of other species within the community. In the rm and nc mice, we see a fairly repeatable process at the family level of families coming back to reasonable abundance throughout the experiment, but this varies a bit mouse to mouse. I am tempted to connect this variation to variation in the timing of the sweeps between replicates - for example there is a late bout of selection (barcodes rising in frequency) in rm4, along with later arrivals of *Ruminococcaceae* and *Oscillospiraceae* in the community dynamics. In Figure 4 shows the results of shape-based metric clustering of clonal clusters and families. The main result here, that the sweeping barcode cluster has high shape similarity to the quickly rebounding *Lachnospiraceae*, makes sense but is hard for me to interpret. I'm not sure that a shape-based metric is best to account for lags. Instead, I wonder if you would find tighter associations

between barcodes or barcode clusters and families if you looked for correlations across all replicates in the log-fold changes at each time interval. You could even include a free time-lag parameter for each association. For example, it would be very convincing if the same lineages rise in frequency at the same time as *Ruminococcaceae* in each rm mouse, but that time is different between mice. The power here actually comes from the natural variation in the timing of the community recovery in each mouse. Of course this still doesn't tell us what is causing these changes in frequency, but it tells us that dynamics within the population are mirroring dynamics between community members, which I think would be a very interesting observation. This is something that I think might be interesting, of course I would understand if you look and it's not there. I do think the current association between C1 and *Lachnospiraceae* doesn't feel convincingly representative of a biological association to me, just because they both happen to be the first thing sweeping in the barcode and 16S data, respectively.

7. The DCM approach serves as a descriptive tool, but in my opinion the descriptive benefit doesn't outweigh the cost here. This is coming from more of an evolutionary biologist than an ecologist, but at the end of the day, I'm not sure what the DCM approach adds to our understanding of *this dataset*. That's not to say that the approach does not have merit itself. But the sentences that describe the dynamics in reference to the DCM phases or matrix eigenvalues are less clear than those that do not reference DCM. Two examples - on page 15: "the eigenvalue decomposition analyses .. indicates a quasi-stable oscillator in Phase III," but I don't see any convincing oscillatory behavior in the community or barcode data (the only thing that could be mistaken for oscillations is the normal noisy variation in low frequency lineages). On page 12, about rm: "The collapse of the resident community in Phase I is manifested in the bacterial community diversity (Extended Data Fig. 4c)." I do not see this collapse in Fig. 2c or in ED Fig. 4c, and in ED Fig 4c-d it looks like the community diversity in the rm early in the experiment is higher than in the nc population. Again, I think it is possible that applying some version of DCM to a larger swath of microbial time series data might be useful (e.g. for building predictive models as you mention in the final sentence), but for this dataset I didn't feel that it added to my understanding.
8. On Page 27, line 18: "progressively increasing time intervals [τ] (3h-6h, 3h-12h, ..., and 3h-15 days)". Are these the time intervals used for the DCM

to create all the figures etc.? The text implies that the DCM results represent these “phases” of dynamics over the course of the experiment, which I assume meant it was applied as a sliding window along the timecourse. But from this description it sounds like it is an “expanding window”, so that the late phase is actually data from the entire experiment. In general I would expect a covariance matrix to change as the amount of data gets larger in a dynamic population, so now I’m just generally confused about the time scale of the model and these phases. Again, I find that DCM contributes more to confusion than understanding.

9. According to the methods, before the gavage of the barcoded population, the rm and nc treatments were the same, but at day one the nc mice have a very different makeup. Page 14: “Sutterellaceae was unperturbed by the introduction of *E. coli* in all four mice (Fig. 2c). Muribaculaceae was unperturbed in rm 1 and rm 3, and it was the first bacterial family to rebound in rm 2 and rm 3 **[I think this should say rm4].**” When I looked at the 16S data in https://github.com/melisgncl/Intra--and-inter-species-interactions-drive-phases-of-invasion-in-gut-microbiota-/tree/main/data/16S/taxa_long_format, I don’t see Muribaculaceae in the rm samples - maybe you mean to refer instead to Acholeplasmataceae here? More importantly, I don’t see Sutterellaceae or Acholeplasmataceae in the nc samples, which are instead dominated by Lactobacillaceae. Do you think there was something that caused one cohort to have such a different microbiota at the start of the experiment? Or do we think the Lactobacillaceae crashed out in the rm mice in 24 hours?
10. I couldn’t find a description of how exactly fitness was calculated in the methods, but the results plotted in Extended Data Fig. 9 make me assume that mean fitness is not being accounted for here, which is why we are seeing so much “negative relative fitness.” As the population is more dominated by the adaptive mutants, the population mean fitness increases so that neutral (ancestral genotype) lineages will decrease in frequency faster. Any measure of fitness here needs to take into account mean fitness (see the Levy and Blundell paper for an example of this). But, for the purposes of this paper, I’m not sure it is necessary to do an in depth fitness analysis - as you say, the goal here is not to measure the DFE, so I would suggest cutting this section altogether, since it doesn’t clearly contribute to the overall story.

Overall Thoughts

This is a really interesting dataset. Once the issues in data processing are cleaned up, this data will clearly represent both inter and intra species dynamics in a realistic and important microbial environment. Like many observational studies, it is difficult here to say exactly what the key conclusions are, and I don't think that is inherently a bad thing - this project gives us intuition for what these dynamics are like, and this intuition can help guide future work. In the final paragraph, you write, "although specific compositions may be highly variable across mice, the overall tempo of ecological and evolutionary dynamics, as manifested by the DCM analysis, are more reproducible features of the microbiota." I think this is a great summary of the work, but it would stand if the reference to DCM was removed from the sentence. My first suggestion is to minimize or remove the discussion of DCM in this paper in favor of a more simple descriptive approach. My second suggestion is to do the barcode clustering across all of the data, so that it is easier to detect similarities between the inter and intra specific dynamics. This is a paper that already seems a bit laden by a previous review, with paragraphs and figures added to address reviewer concerns (below I'll suggest cutting some of this!), so I want to be clear that what I'm advocating for here is a simplification of the paper, in which the data in Figure 2 and some version of the idea in Figure 4 are blown up and given more weight, so that other researchers can gain intuition about the dynamics of microbes in the gut from this work.

Figure comments

1. General comment: The x-axis in all of these figures with time on the x-axis needs to either have 12h scaled as half a day, or have a split x-axis, with the first day shown and then a break in the line representing a change in time scale before the day-by-day samples are shown.
2. Figure 1: The graphics in panel d are nice! For 1e, I'm confused - the methods says the CFUs are on spectinomycin and should only be the barcoded population, but the im data makes it look like it is the whole gut population? Is the dashed line the resolution limit (this should be explained in the caption)?
3. Figure 2: I would go lower on the number of colored barcodes (maybe top 250) to make the dynamics easier to see. There are only a handful of families dominating the dynamics here, only those should be shown in color so the legend is useful (right now I have no idea what anything is and had to look at the data myself).
4. Figure 3: As discussed above, for me this is more confusing than useful.

5. Figure 4: I am hopeful that with new clusters the result here might become more clear. It would be great to see an example of similar trajectories between a species and barcode cluster across the 4 mice (if a good one exists).
6. Figure 5: Clustering across mice would make this figure unneeded
7. Figure 6: Removing the mutations found at 100% in the gavage will simplify this figure a bit. I like the synteny layout of the grid.

Detailed comments:

1. Abstract, first sentence “a microbiome’s composition...is dictated by the community interaction matrix” - this ignores any higher order effects etc. Maybe “is often described using the community interaction matrix, which is commonly assayed...”
2. Page 3, lines 3-4: “These characteristics are embedded in the community interaction matrix that quantifies” - same comment as above, I don’t think embedded is the right word, maybe “can often be explained using the community interaction matrix, which quantifies” (I think the “, which” just reads much better in both cases).
3. A few examples of sentences that break the flow due to sentence structure / grammar issues. Page 3, lines 11-12: “[microbes] have members that are challenging to culture or practically isolate” - the subject of this clause is microbes, which doesn’t make sense. Page 3, lines 15-16: “Non-parametric models of the community models of the community matrix does not incorporate evolutionary forces” - “does” should be “do” here, but more importantly the first half of the sentence is just hard to parse. Maybe just “Most non-parametric community models do not incorporate...”? Page 3, line 21, “the community matrix, is in principle, also influenced” - remove the commas. Page 4, lines 4-5 “to understand its stability and dynamics” - maybe there is supposed to be an “and” before this text? Right now it doesn’t make sense.
4. Page 3 line 29: I would say “A typical human gut microbiome” (I assume you’re referring to a human gut, or say mouse if that’s the case).
5. End of page 3, start of page 4: I’m not sure “clones” is the best word to be using here. Maybe “haplotypes” is more direct and clear?
6. Page 4, line 17: “*E. coli* population” -> “microbial populations” - most of this work hasn’t been in *E. coli*.
7. Page 7, line 6, “enables [us] to”
8. Page 10, lines 22-23 “suggesting a strong intraspecies selection pressure in the absence of resident bacteria” - to me, this is one of the coolest results in the paper!
9. Page 10, line 27: “Barcodes that appeared” - I think “Lineages” is a better term than “Barcodes”, since barcodes refers to the data, but lineages refers to the actual system.
10. Page 12, line 30, last word: “include[s]”

11. Page 16, lines 17-19: “the similarity in barcode dynamics and composition is understandably weaker in the rm and gf mice, where the effects of standing genetic variation and/or de novo mutations in rm are modulated by ecological interactions with **[a changing]** bacterial community” - I think adding that the bacterial community is dynamic is important and should make this more clear.
12. Page 20, line 15: cut “below” (redundant)
13. Page 20: line 18 “dynamic[s]”
14. Page 20 lines 30-31: “we demonstrate that intra-species variation leads to time-dependent interactions.” Unfortunately I don’t think this is clearly demonstrated - while the community interaction matrix is trying to quantify interactions, it may just be quantifying correlations between abundances and abundance derivatives that occur by chance or due to outside environmental factors. It’s also not clear to me that the intra-species variation in particular is leading to any time-dependent interactions.
15. Page 21, line 9: “Other previous studies” - cut “Other” (redundant)
16. Page 21, line 9: “dynamics like gut invasion” -> “dynamics of gut invasion”
17. Page 21, line 20: “providing much higher DNA barcode density” -> “barcode **diversity**”.
18. Page 22, lines 10-26: This paragraph feels like a response to a previous reviewer asking you to compare to plasmid systems. In my opinion it doesn’t belong in the discussion and should be moved to the supplement so it doesn’t interrupt the flow of the more important things you’re discussing here.
19. Page 23, lines 8-17. I like the first clause here about the goal of this study, but the rest of the paragraph again feels like a response to a previous reviewer and distracts from your main message.
20. Page 23, line 23: “could has” -> “could have”
21. Page 24, line 8: “underly the C1 *Lachnospiraceae*” -> “underly the similarity between C1 and *Lachnospiraceae* dynamics.” And above on line 4 “C1 *Lachnospiraceae* interaction” -> “C1-*Lachnospiraceae* association”, because this is not necessarily an interaction. I also suggest moving this paragraph to the results - having it in the discussion puts too much weight on it and breaks up the flow.
22. Page 24, line 12: “pedantic” - maybe you meant “didactic”? I would just cut, “illustrative examples” is enough.
23. Page 24, lines 15-16 “is too simplistic and inadequate to” -> simplify to “cannot”

Thoughts for the future:

This is not a suggestion for this paper, but I wanted to include a hypothesis for what could be the most interesting phenomena here. In the gf mice, we repeatedly see a kind of

secondary sweep around day seven in which lineages are getting up to $\sim 1\%$ frequency. We see similar family-level dynamics in the rm data. I wonder whether this is the same successional pattern - for example maybe these gf lineages are fulfilling a similar ecological role as one of the families rising to $\sim 0.1-1\%$ in rm mice. The C2 clones with lrp insertions are examples of these gf lineages, and look like they have a different metabolic strategy based on ED Fig. 10. Again, not a suggestion to dig into for this paper, just an idea.

Once again, I think this paper presents an excellent experiment and dataset, which represent a very exciting and interesting way to try to get at the interplay between ecology and evolution in a semi-natural environment. And I thank the authors for their response to my comments and the changes they've made.

As stated in my previous review, I think that papers should be published as long as they are not misleading, and review should be aimed at improving papers. I think this new draft is improved in many ways, but I think there is the possibility to dramatically strengthen it by subtraction (or rather, by moving things to the methods/supplement). In any case, I certainly recommend the paper for publication.

At a broad level, I still feel that the emphasis on DCM distracts and confuses more than it adds, but I'm willing to accept the authors' conviction that I'm wrong on this front. Because the other reviewer seems more well-versed in the details of this modeling field, I will leave the details of DCM, the simulations, etc. to them.

Remaining comments:

- 1) I was not clear enough in my Comment 3. When I recommended you cluster barcodes “*across* all of your rm and gf data”, I was suggesting that you do one clustering step on all the data, such that e.g. C1 is the same set of barcodes across experiments. Since most of the variation in the *E. coli* population appears to be from pre-gavage, this should work, and it should drastically simplify your figures and analysis. This is why I suggested you cut Figure 5 (panels a and d would still be relevant, but could be moved to a supplemental figure). I still think this is a good idea and would A) simplify your analysis, and B) better represent real variation in the *E. coli* population, making the analysis about associations between clusters and microbial taxa more powerful.
- 2) On page 15 you write “*Phase III and phase IV is quasi-dynamic equilibrium with both oscillations in the clonal and community dynamics (Fig. 3b, right; Fig.2b,c). The oscillatory behavior of the barcode dynamics that affects the community dynamics is unlikely due to technical and experimental factors, such as PCR bias, efficiency in genome extraction, or number of E. coli genomes for PCR amplification (see Supplementary Notes section A).*” I think we may just disagree here about what constitutes “oscillations.” In my mind it implies a kind of repeated, predictable back-and-forth in frequencies, which I don't see here. I prefer your term “quasi-dynamic equilibrium,” but I am ok with whatever you decide.
 - a) I read Supplementary Notes section A, but I'm confused by the PCR bias analysis. The worst case scenario you model is $\epsilon=0.5$ for some barcodes, which

results in a ~7000-fold frequency change, which is an extreme amount of PCR bias (for example we could observe a barcode that has a true frequency of 1% as being ~98.7% of the population)! I don't quite understand how this doesn't lead to a large change in the log₁₀ frequencies in Figure S1j, but maybe it's because the biases are being applied to each barcode across all timepoints? In my experience the issues arise when PCR biases (or index-hopping etc.) occur at one timepoint and cause barcodes to jump up or drop down in frequency. I think it is misleading to suggest that this amount of PCR bias will affect results less than the amount of template DNA. I would actually be in favor of you cutting Supplementary note A1-3 altogether. It is clear that neither PCR bias or a low amount of template DNA is affecting your data at mid to high frequencies - the barcode trajectories look good! (with the exception of the low-frequency noise, which comes from a bunch of unknown sources and is present in practically every dataset of this kind that I've seen).

- 3) This is partly my fault, but the paper has become even more laden by response-to-reviewer-ballast, which I think further distracts from the cool story at the heart of all this. Some ideas:
 - a) I understand I won't convince you to minimize the modeling aspect, but the Results section quickly gets so deep into the modeling weeds. Is there a way to move some parts of this to the methods and focus on the big picture more?
 - b) Related to 2a above - on Page 12, paragraph starting with "To ascertain..." could be shortened drastically to convey the point that the population crashed in im and so the data wasn't usable. I really don't think you need Supplementary note A, just saying there wasn't enough template DNA to get good data is sufficient.
 - c) As mentioned above, I think you should cluster across all data and then lose figure 5.
- 4) Page 18: "These results demonstrated the impact of *E. coli* introduction on bacterial community composition." As discussed in your response to review, it seems likely that *Lactobacillaceae* was abundant in the nc samples due "to natural differences in microbiota between cohorts", so unfortunately I don't think you can make this claim.

Figures:

- 1) General: I understand your decision to maintain the hourly resolution on day one in your figures. But since the x-axis represents time, you need to add a scale break mark (/ /) to the x-axis where the scale changes. I think just adding the mark would be sufficient to convey the scale change.

- 2) Figure 2: I understand your choice to keep the many colors for the barcodes. But for the families I really think just coloring the ones you mention in the text, or ones that reach above some frequency is a good idea. This will make it much easier for readers to connect the figure to the results. Right now it is hard to look through the long legend and find a family's trajectory data, and for most of those families you can't tell which line they are even if you try. Another option would be to make the key families you mention in the text have a thicker line weight and move them to the top of the legend.

Small comments:

- 1) Page 3 lines 3-4: "can be often explained" -> "can often be explained"
- 2) Page 5 line 19: "other taxonomic unit (OTU)." Do you mean "operational taxonomic unit"? "other taxonomic unit" would also make sense, but then you shouldn't use the OTU acronym, since that is well established for "operational taxonomic unit."
- 3) Page 9 line 11: "We also applied DCM analysis to mice gut microbial community perturbed..." - maybe "a mouse gut microbial community"?
- 4) Page 9 line 13 "identifies" -> "identify"
- 5) Page 24 line 7 "This suggests that the deletion could have been present in this specific barcode prior to gavage..." - I think you can be stronger here, this is very strong evidence: "the deletion was present in this specific barcode prior to gavage..."
- 6) Page 24 line 9: "(genomes of a few hundred cells)" - this is picky but the metagenomics might have a depth of ~100X but that DNA likely comes from millions of cells. I would just cut this parenthetical.
- 7) Page 30 lines 20-22 - this sentence has a grammar issue and is difficult to read.